# Dynamical feature extraction at the sensory periphery guides chemotaxis

Aljoscha Schulze[1,2†], Alex Gomez-Marin[1,2†‡], Vani G Rajendran[1,2,3§], Gus Lott[3], Marco Musy[1,2], Parvez Ahammad[3¶], Ajinkya Deogade[1,2], James Sharpe[1,2,6], Julia Riedl[1,2#], David Jarriault[1,2∥], Eric T Trautman[3], Christopher Werner[3**], Madhusudhan Venkadesan[4,5], Shaul Druckmann[3], Vivek Jayaraman[3], Matthieu Louis[1,2,3*]

[1]EMBL-CRG Systems Biology Program, Centre for Genomic Regulation, Barcelona, Spain; [2]Universitat Pompeu Fabra, Barcelona, Spain; [3]Janelia Research Campus, Howard Hughes Medical Institute, Ashburn, United States; [4]Department of Mechanical Engineering and Materials Science, Yale University, New Haven, United States; [5]National Centre for Biological Sciences, Tata Institute of Fundamental Research, Bangalore, India; [6]Institucio Catalana de Recerca i Estudis Avancats, Barcelona, Spain

*For correspondence: mlouis@crg.eu

†These authors contributed equally to this work

Present address: ‡Champalimaud Neuroscience Programme, Champalimaud Centre for the Unknown, Lisbon, Portugal; §Department of Physiology, Anatomy and Genetics, University of Oxford, Oxford, United Kingdom; ¶Instart Logic Inc., Palo Alto, United States; #Research Institute of Molecular Pathology, Vienna, Austria; ∥Centre des Sciences du Goût et de l'Alimentation, Université de Bourgogne, Dijon, France; **T2Biosystems Inc., Lexington, United States

Competing interests: The authors declare that no competing interests exist.

**Abstract** Behavioral strategies employed for chemotaxis have been described across phyla, but the sensorimotor basis of this phenomenon has seldom been studied in naturalistic contexts. Here, we examine how signals experienced during free olfactory behaviors are processed by first-order olfactory sensory neurons (OSNs) of the *Drosophila* larva. We find that OSNs can act as differentiators that transiently normalize stimulus intensity—a property potentially derived from a combination of integral feedback and feed-forward regulation of olfactory transduction. In olfactory virtual reality experiments, we report that high activity levels of the OSN suppress turning, whereas low activity levels facilitate turning. Using a generalized linear model, we explain how peripheral encoding of olfactory stimuli modulates the probability of switching from a run to a turn. Our work clarifies the link between computations carried out at the sensory periphery and action selection underlying navigation in odor gradients.

## Introduction

Chemosensation is an evolutionarily ancient sense found in nearly every living organism. In bacteria, chemotaxis allows individual cells to detect the presence of food and to accumulate in its vicinity. Multicellular organisms have evolved complex sensory systems to track temporal changes in the concentration of volatile odorant molecules relevant to their survival—food odors, pheromones associated with the presence of conspecifics and substances signaling danger. In turn, sensory perception drives behavioral strategies to forage, locate a mating partner and actively avoid danger (*Fraenkel and Gunn, 1961*; *Schöne, 1984*). Bacterial chemotaxis represents the archetype of orientation behavior in unicellular organisms: phases of relatively straight motion—called *runs*—alternate with changes in orientation—called *tumbles*—that randomize the direction of the next run (*Berg, 2004*). Accumulation near the source of an attractive chemical results from the elongation of runs in the direction of the gradient. In multicellular organisms, olfactory behaviors have been investigated in detail in the nematode *Caenorhabditis elegans* (*Bargmann, 2006a*), which uses a combination of undirected turns ('pirouettes') and continuous correction of the orientation of individual runs ('weathervaning') (*Iino and Yoshida, 2009*; *Lockery, 2011*). The neural computations enabling animals with a central nervous system to orient in odor gradients, however, remain poorly understood.

**eLife digest** Fruit flies are attracted to the smell of rotting fruit, and use it to guide them to nearby food sources. However, this task is made more challenging by the fact that the distribution of scent or odor molecules in the air is constantly changing. Fruit flies therefore need to cope with, and exploit, this variation if they are to use odors as cues.

Odor molecules bind to receptors on the surface of nerve cells called olfactory sensory neurons, and trigger nerve impulses that travel along these cells. While many studies have investigated how fruit flies can distinguish between different odors, less is known about how animals can use variation in the strength of an odor to guide them towards its source.

Optogenetics is a technique that allows neuroscientists to control the activities of individual nerve cells, simply by shining light on to them. Because fruit fly larvae are almost transparent, optogenetics can be used on freely moving animals. Now, Schulze, Gomez-Marin et al. have used optogenetics in these larvae to trigger patterns of activity in individual olfactory sensory neurons that mimic the activity patterns elicited by real odors. These virtual realities were then used to study, in detail, some of the principles that control the sensory navigation of a larva—as it moves using a series of forward 'runs' and direction-changing 'turns'.

Olfactory sensory neurons responded most strongly whenever light levels changed rapidly in strength (which simulated a rapid change in odor concentration). On the other hand, these neurons showed relatively little response to constant light levels (i.e., constant odors). This indicates that the activity of olfactory sensory neurons typically represents the rate of change in the concentration of an odor. An independent study by Kim et al. found that olfactory sensory neurons in adult fruit flies also respond in a similar way.

Schulze, Gomez-Marin et al. went on to show that the signals processed by a single type of olfactory sensory neuron could be used to predict a larva's behavior. Larvae tended to turn less when their olfactory sensory neurons were highly active. Low levels and inhibition of activity in the olfactory sensory neurons had the opposite effect; this promoted turning. It remains to be determined how this relatively simple control principle is implemented by the neural circuits that connect sensory neurons to the parts of a larva's nervous system that are involved with movement.

The *Drosophila* larva has the smallest known olfactory system analogous to that of vertebrates (*Cobb, 1999*; *Bargmann, 2006b*; *Gerber and Stocker, 2007*; *Vosshall and Stocker, 2007*). The larva achieves robust odor gradient ascents through an alternation of approximately straight runs and turning events (*Gomez-Marin et al., 2011*; *Gershow et al., 2012*). The duration of runs is modulated by the sensory input: runs up the gradient are elongated while runs away from it are shortened (*Gomez-Marin et al., 2011*; *Gershow et al., 2012*). Although published results hint at how larval chemotaxis may be achieved (*Gomez-Marin and Louis, 2012*, *2014*), a quantitative model of the underlying sensorimotor integration is still missing. Here, we focus on the primary task of the orientation algorithm common to bacteria, *C. elegans*, and *Drosophila*: the control of run duration (*Bargmann, 2006a*; *Lockery, 2011*; *Gomez-Marin and Louis, 2012*). It is known that turns are preceded by stereotyped decreases in odor concentration (*Gomez-Marin et al., 2011*; *Gomez-Marin and Louis, 2012*), but the key question of how concentration differences are computed is unresolved.

In both insects and vertebrates, odor concentrations are represented by time-varying patterns of activity distributed across the olfactory sensory neuron (OSN) population (*Wilson and Mainen, 2006*; *Wilson, 2013*; *Masse et al., 2009*; *Mainland et al., 2014*; *Uchida et al., 2014*). Nonetheless, animals with an olfactory system genetically reduced to a single functional OSN are still capable of robust chemotaxis (*Fishilevich et al., 2005*; *Louis et al., 2008*), implying that the mechanisms of odor concentration detection can be understood at the level of single OSNs. Here, we rely on this simplification to develop a novel larval preparation in which the neural computations underlying odor gradient ascent can be understood in unprecedented detail. We used optogenetics in larvae with a single type of functional OSNs to substitute turbulent olfactory signals with well-controlled light stimulations (*Suh et al., 2007*; *Bellmann et al., 2010*; *Smear et al., 2011*; *Gaudry et al., 2013*). This allowed us to characterize the modulatory effects of OSN firing patterns on the probability of switching from a run to a turn. Toward this goal, we developed a novel tracker to create virtual

olfactory realities (*Kocabas et al., 2012*) in which optogenetic stimulations of genetically targeted OSNs are defined based on the behavioral history of the larva. We used this technology to derive a phenomenological model of the OSN transfer function. The model was validated on free behavior in sensory landscapes designed to produce predictable sensorimotor responses, and ultimately, it was found to be applicable to real odor gradients. We found that for positive gradients, the OSN operates as a slope detector: its activity increases with the stimulus derivative, which suppresses the probability of turning. For strongly negative gradients, the OSN acts like an OFF detector: the inhibition of the neural activity facilitates turning in a nearly deterministic manner. Altogether, our results advance our understanding of how peripheral odor encoding guides action selection during chemotaxis.

## Results

### Run-to-turn transitions as a paradigm for action selection

Odors are generally attractive to *Drosophila* larvae (*Cobb, 1999*). Exposure to an odor produces gradient ascent even in larvae with a genetically manipulated olfactory system reduced to a single OSN (*Fishilevich et al., 2005*; *Louis et al., 2008*). We examined the behavior of larvae with a single functional OSN expressing Or42a, an odorant receptor with a well-characterized tuning profile that includes the odorant isoamyl acetate (IAA) (*Fishilevich et al., 2005*; *Kreher et al., 2008*; *Asahina et al., 2009*). Behavior was studied in a closed environment with a single source of IAA suspended from the 'ceiling' of the arena (*Figure 1* and 'Materials and methods'). For large odor droplets, diffusion from the source creates a radially symmetric gradient that can be approximated by a stationary Gaussian distribution (*Louis et al., 2008*). For smaller odor droplets such as those used in the present study, the temporal evolution of the odor gradient cannot be neglected. We therefore combined infrared spectroscopy (IR) and a partial differential equation (PDE) model to experimentally reconstruct the two-dimensional geometry of the odor gradient over time (see *Figure 1B* and 'Materials and methods'). The simulated odor gradient served as a template to reconstruct the average stimulus time course experienced by the larva during real trajectories (*Figure 1C*).

   *Figure 1A* presents a trajectory consisting of approximately straight segments ('runs') punctuated by large changes in orientation ('turns'). Where to turn to is determined through lateral exploratory head movements, 'head casts', during which the larva scans the local odor gradient (*Gomez-Marin et al., 2011*; *Gershow et al., 2012*). On average, larvae terminate their runs when motion is directed down the gradient where the odor concentration is decreasing (*Gomez-Marin et al., 2011*; *Gershow et al., 2012*). In contrast, turns are suppressed when the direction of motion is along the gradient and the odor concentration is increasing. We sought to define the neural computations underlying this behavior by characterizing the neural activity of the *Or42a*-expressing OSN in response to changes in odor concentration experienced during chemotactic behavior (*Figure 1C*).

### Peripheral representation of naturalistic olfactory stimuli in a single larval OSN

To probe the input–output transfer function of the *Or42a* OSN, we devised an extracellular recording technique based on the suction of the antennal nerve into a glass pipette downstream from the dorsal organ (DO) ganglion (*Figure 2A* and 'Materials and methods'). With the use of an optogenetic spike-sorting strategy we identified the spikes originating from the *Or42a* OSN expressing channelrhodopsin (ChR2) (denoted as '*Or42a*>ChR2 OSN') (*Figure 2B,C* and 'Material and methods'). We devised a customized olfactometer to produce odor stimuli with controlled temporal profiles (*Figure 2A*) with which we examined the response of the *Or42a*>ChR2 OSN to a concentration replay defined by the stimulus time course associated with the trajectory shown in *Figure 1A*. Recordings from this 3-min stimulation led to consistent patterns of neural activity in different preparations (*Figure 2D*). Although the OSN activity appeared to follow the envelope of the stimulus time course, a closer examination revealed greater complexity in the neural response. The OSN firing rate displayed a clear amplification of changes in stimulus intensity, as illustrated by the activity associated with the replay of two consecutive runs, R1 and R2 (*Figure 2F*). Run R2 brought the larva close to and then beyond the peak of the gradient. When stimulated by the corresponding time course of the odor concentration, OSN activity peaked several seconds before the stimulus intensity (*Figure 2F*, arrows). Minute fluctuations in odor concentration were strongly amplified in the OSN spiking dynamics (bursts marked by a sharp # symbol in *Figure 2F*).

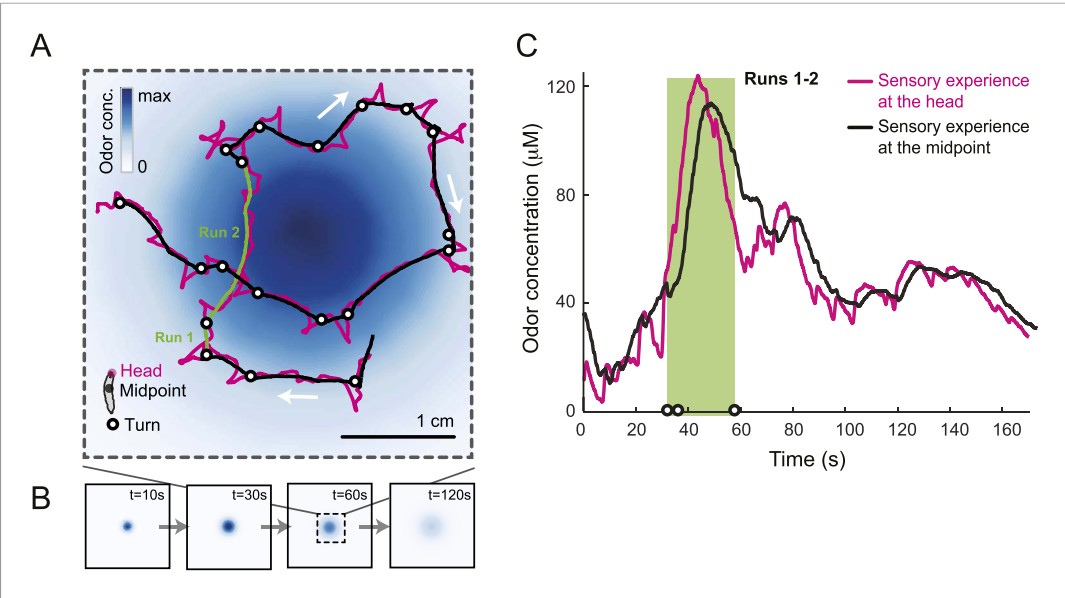

**Figure 1**. Sensory experience corresponding to unconstrained chemotactic behavior. (**A**) Illustrative trajectory of a larva freely moving in an attractive odor gradient (isoamyl acetate, source concentration: 0.25 M). Position of the midpoint shown in black; position of the head shown in magenta. Two run segments, R1 and R2, are underlined in green. Turns are depicted as white disks. White arrows indicate the direction of motion. (**B**) Reconstruction of the odor gradient based on numerical simulations of the odor diffusion process modeled by a partial differential equation (PDE) system with realistic boundary conditions ('Materials and methods'). The gradient shown in panel **A** corresponds to a snapshot obtained 60 s after onset of the odor diffusion. (**C**) Time course of the odor concentration experienced at the head (magenta) and midpoint (black) of the larva during the trajectory shown in panel **A**. The sensory experiences are reconstructed based on mapping the positions of interest on the dynamic odor gradient computed upon integration of the PDE system for the entire duration of the trajectory. The green box outlines the sensory experience corresponding to run segments R1 and R2. Small disks on the abscissa indicate the turns comprised in this behavioral sequence.

## Characterization of the features encoded by a single larval OSN stimulated by controlled olfactory signals

To tease apart the sensory features encoded by the *Or42a*>ChR2 OSN, we examined the OSN response induced by a set of controlled odor ramps with a temporal profile analogous to the run sequence described in *Figure 2F*. As a first approximation, we used linear ramps with symmetrical 8-s rising and falling phases. During the rising phase of the ramp, the neural activity increased in proportion with the derivative of the odor concentration (*Figure 3A*). During the falling phase of the ramp, the firing rate appeared to be driven by the stimulus intensity rather than the stimulus derivative (*Figure 3B*), suggesting that the response properties of the OSN differed for positive and negative gradients.

To assess the slope sensitivity of the OSN, we compared the neural activity elicited by nonlinear ramps in which the first derivative of the stimulus changed over time (*Figure 3C*). In an exponential ramp, the stimulus derivative increased throughout the rising phase of the ramp. This acceleration correlated with a continuous increase in spiking activity. To further test the hypothesis that the OSN encodes features related to the slope of the stimulus, we examined a sigmoid ramp (*Figure 3C*, right panel) for which the first derivative of the stimulus reached its maximum (gray arrow) prior to the stimulus intensity (magenta arrow). Consistent with the slope-sensitivity hypothesis, the OSN spiking activity peaked with the first derivative and not the absolute intensity of the stimulus. During the falling phase of the ramps, the OSN firing rate behaved in a way that could not be predicted from the slope sensitivity observed during the rising phase. At the end of the falling phase, OSN activity decreased below baseline (star signs * in *Figure 3C*), suggesting offset inhibition similar to that observed in OSNs of adult flies (*Hallem et al., 2004*; *Nagel and Wilson, 2011*). Our findings on the

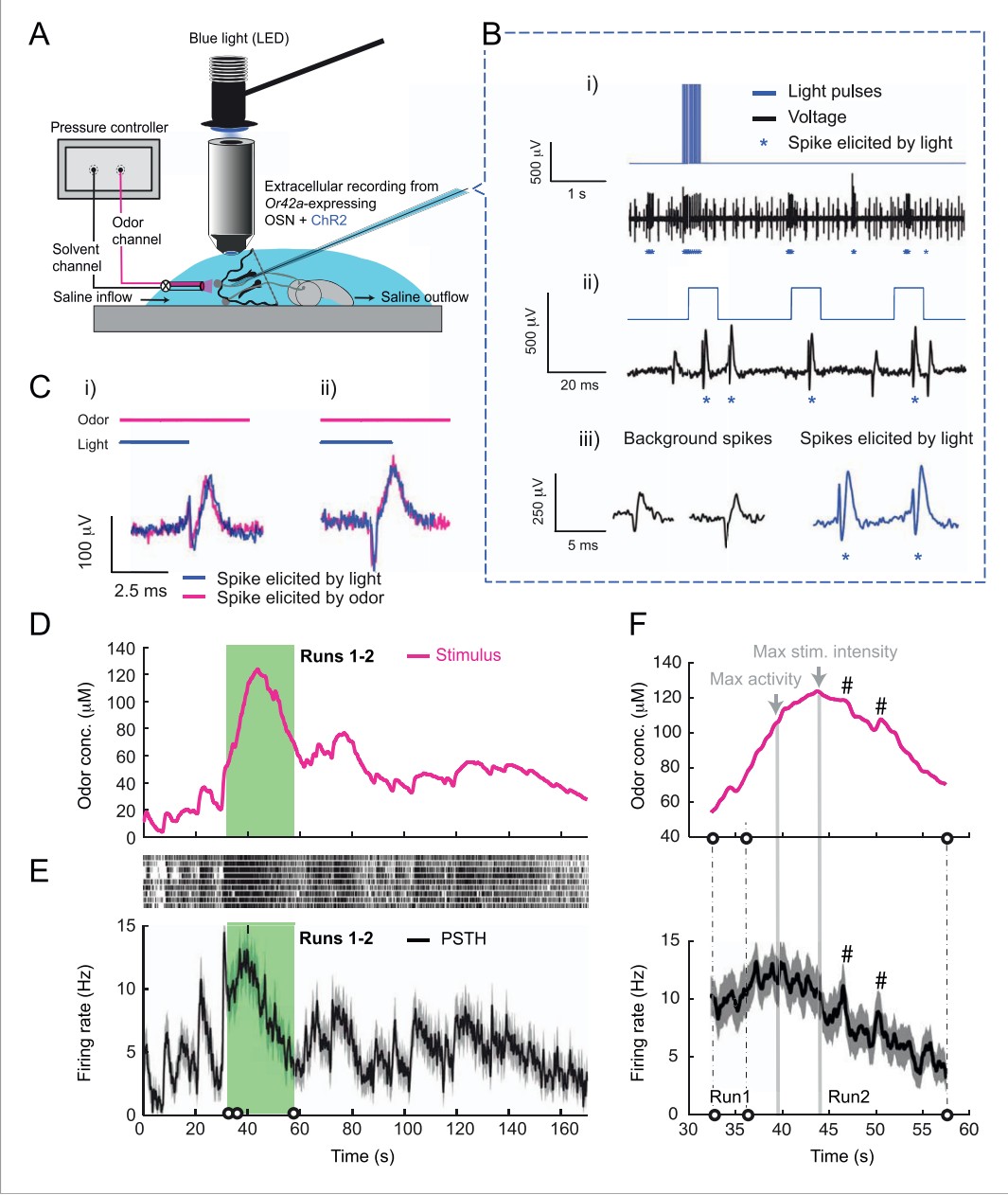

**Figure 2**. Response of a single larval olfactory sensory neuron (OSN) to naturalistic odor stimulation. (**A**) Illustration of preparation for suction electrode recordings of single functional OSNs expressing channelrhodopsin (ChR2). The preparation is bathed in saline to prevent the dehydration of the dorsal organ ganglion to which the recording electrode is attached. Controlled odor stimulations are achieved in liquid phase with a customized mass flow controller system. (**B**) Recording from the dorsal organ stimulated by a series of 10-ms light pulses. (**i–ii**) The voltage trace shows spikes with different amplitudes. (**iii**) Close-up view of the voltage trace corresponding to three consecutive light pulses. Action potentials with a stereotyped waveform are observed at a short-latency after the onset of the light pulse (spikes denoted by a blue star *). These spikes are associated with the activity of the *Or42a* OSN expressing ChR2. Each light pulse yielded an average of 1.8 light-evoked spikes. (**C**) Superimposition of light- and odor-evoked spike waveforms observed for the same OSN. The results of two different recordings are shown in (**i**) and (**ii**). Spike waveforms associated with the light stimulation (blue spikes) are superimposed on spike waveforms collected during an episode of odor stimulation (magenta spikes). The high similarity between the light- and odor-evoked spikes serves as a basis to spike sort the recordings arising from the dorsal organ ganglion ('Materials and methods'). (**D**) Dynamic reconstruction of the concentration time course corresponding to the trajectory of the head position depicted in *Figure 1*. (**E**) Results of 9 suction electrode recordings for the *Or42a>*ChR2 OSN stimulated by *Figure 2. continued on next page*

*Figure 2. Continued*

the concentration shown in panel **D** (5 preparations). (Top) Raster plot. (Bottom) PSTH of the OSN response to the concentration time course shown in panel **D** with shade representing the standard deviation. (**F**) Close-up view of the sensory experience (top) and OSN response (bottom) corresponding to the illustrative runs R1 and R2 shown in panel **D** (green box). Since the maximum firing rate is attained earlier than the stimulus intensity reaches its maximum, the input–output relationship driving the dynamics of the OSN activity is more complex than a proportional detector. Short increases in odor concentration lead to transient bursts in spiking activity (bursts indicated by sharp # signs). Small disks on the abscissa denote turns in the original trajectory presented in *Figure 1*.

features encoded by the OSN were corroborated by responses elicited by other odor ramps (*Figure 4—figure supplement 1*).

We attempted to model the input–output relationship of the OSN by following a linear system-identification approach (*Chichilnisky, 2001*). To this end, we applied an M-sequence (pseudorandom binary sequence with nearly flat frequency spectrum) and reverse-correlation (*Geffen et al., 2009*), but discovered that the resulting linear filter was insufficient to account for the firing patterns elicited by naturalistic stimuli ('Materials and methods'). We thus turned to dynamical systems theory to capture the nonlinear characteristics of the OSN response. We developed a biophysical model that accounts for the slope-sensitivity of the OSN during stimulus upslopes, proportionality response and offset inhibition during downslopes (*Figure 4*).

## Phenomenological model of the olfactory transduction cascade

Negative feedback is known to play an important regulatory function in sensory transduction. Integral feedback control underlies perfect adaption in bacterial chemotaxis (*Yi et al., 2000*). In vertebrates, the olfactory transduction cascade involves a metabotropic pathway downstream from a G-protein coupled receptor (GPCR) that features negative regulatory feedback (*Kaupp, 2010*; *Pifferi et al., 2010*). As with phototransduction, adaptive features of the olfactory transduction cascade in vertebrate can be accounted for by integral feedback (*De Palo et al., 2012*; *De Palo et al., 2013*). Even though invertebrate olfaction does not rely on GPCR signaling (*Kaupp, 2010*), the existence of negative feedback on the odorant receptor has been postulated for olfactory transduction in adult-fly OSNs (*Nagel and Wilson, 2011*). This conclusion was drawn from a biophysical model that combined a linear filter accounting for the OSN spiking dynamics with a kinetic formalism to describe ligand–receptor interactions. Research in the moth has revealed a different regulatory mechanism that constitutes an 'incoherent feed-forward' loop (*Alon, 2007*) in which the activity of the odorant receptor has a dual effect on the OSN spike rate (*Gu et al., 2009*): (1) on a short timescale, the inflow of cations increases the firing rate; (2) on a longer timescale increasing concentration of intracellular calcium ions inhibits the OSN firing rate through a pathway that involves the binding of calcium to calmodulin.

Combining the previous ideas, we hypothesized that the OSN spiking activity is regulated by a negative feedback loop (or integral feedback, IFB) coupled with an incoherent feed-forward loop (IFF) (*Figure 4A* and 'Materials and methods'). In what follows, this composite model will be denoted as IFB+IFF (*Figure 4Bi*). Using a mass-action-kinetics formalism originally developed for genetic networks (*Ackers et al., 1982*; *Bintu et al., 2005*), each of the two regulatory motifs was described by a system of two ordinary differential equations (ODEs) with three variables (*Figure 4Bii*): $x$, the stimulus strength (input: odor concentration or light intensity), $y$ the instantaneous firing rate of the OSN (output), and $u$, a phenomenological variable that might represent the intracellular concentration of calcium. The free parameters of the model were determined through a simplex algorithm which optimized the fit between the experimental spiking activity of the *Or42a*>ChR2 OSN and that produced by the ODE model. The optimization was achieved on a set of 10 linear and nonlinear ramps listed in *Figure 4—figure supplement 1* together with the naturalistic stimulus presented in *Figure 2D*. The parameter set derived from the 10 odor ramps and naturalistic stimulus (*Table 1*) led to a remarkably good fit between the output of the ODE model and the experimentally measured spiking activity (experimental peristimulus time

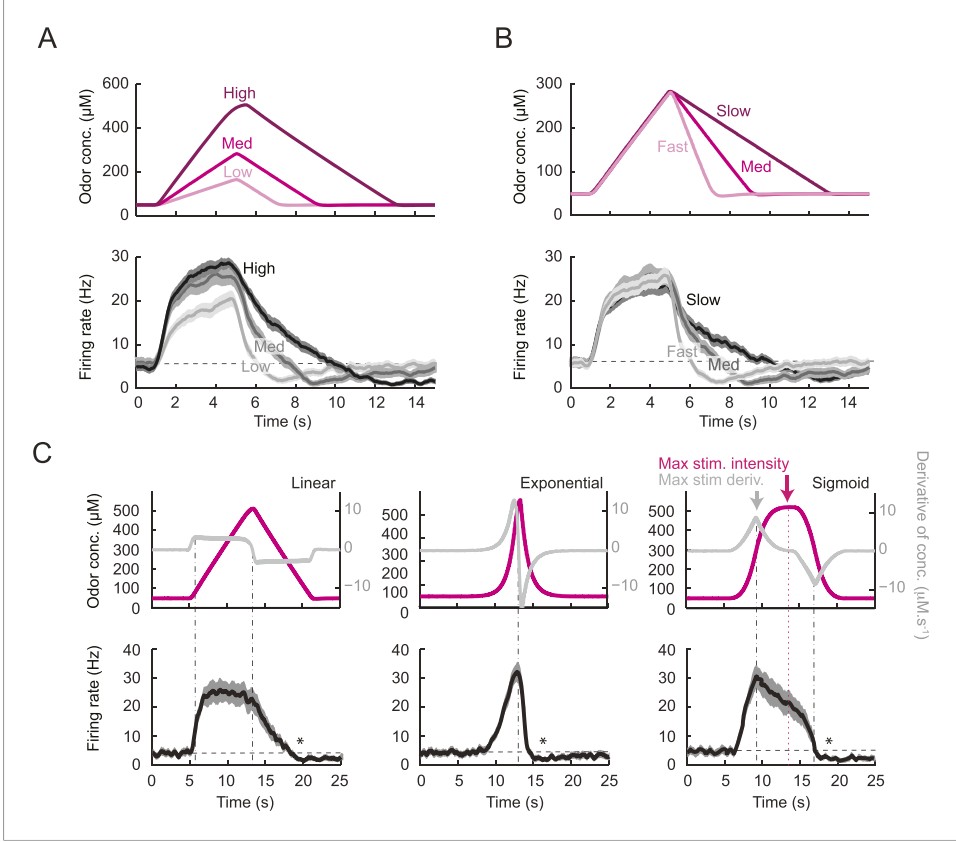

**Figure 3.** Characterization of the dynamical features extracted by the *Or42a*-expressing olfactory sensory neuron. (**A**) Response to three linear ramps with variable slopes during the rising phase and equal slope during the falling phase. The 'low' ('high') ramps have a positive slope that is twice slower (faster) than the medium ramp. PSTH computed on a pool of minimum 24 recordings obtained from minimum 8 preparations. During the rising phase of the ramp, the activity of the OSN reaches a peak value that scales with the slope of the ramp. (**B**) Response to three linear ramps with variable slopes during the falling phase and equal slope during the rising phase. The 'slow' ('fast') ramp has a negative slope that is twice slower (faster) than the medium ramp. During the rising phase of the ramp, the plateau reached by the OSN activity grows with the slope of the ramp. During the falling phase of the ramp, the activity of the OSN is more directly driven by the stimulus intensity. For the three ramps, the OSN activity becomes inhibited when the ramp terminates. PSTH computed on a pool of minimum 24 recordings obtained from minimum 8 preparations. (**B**) Response to nonlinear ramps featuring a symmetrical 8 s-rise and 8 s–fall profiles. From left to right, the ramps tested have the following characteristics: linear ($\propto t$), exponential ($\propto e^{-8}(e^t - 1)$), and sigmoid ($\propto t^3/(t^3 + 4^3)$) with the time given in s. PSTH computed on a pool of minimum 16 recordings obtained from minimum 9 preparations. For panels **C**, the odor concentration (magenta) is computed from the flow ratio measured experimentally based on the flow controller outputs ('Materials and methods'). The time derivative of the concentration time course is represented according to the *y*-scale on the right of the graph (gray). The derivative was computed after mild smoothening of the stimulus input. Asterisks denote inhibitory phases of the OSN response where the activity decreases below its basal level (horizontal dashed line).
The following figure supplement is available for figure 3:

**Figure supplement 1.** Dose-response of the *Or42a*-expressing olfactory sensory neuron stimulated by prolonged odor pulses.

histogram PSTH—black line, results of the IFF+IFB model—blue line; *Figure 4C,E,F*, *Figure 4—figure supplement 1*). Throughout the study, this parameter set was used to reproduce and predict the OSN spiking activity elicited by olfactory stimuli. To rule out over-fitting, we trained the IFF+IFB model on a partial dataset containing the linear ramps alone and validated its response against other stimuli not present in the training dataset (*Figure 4—figure supplement 2*).

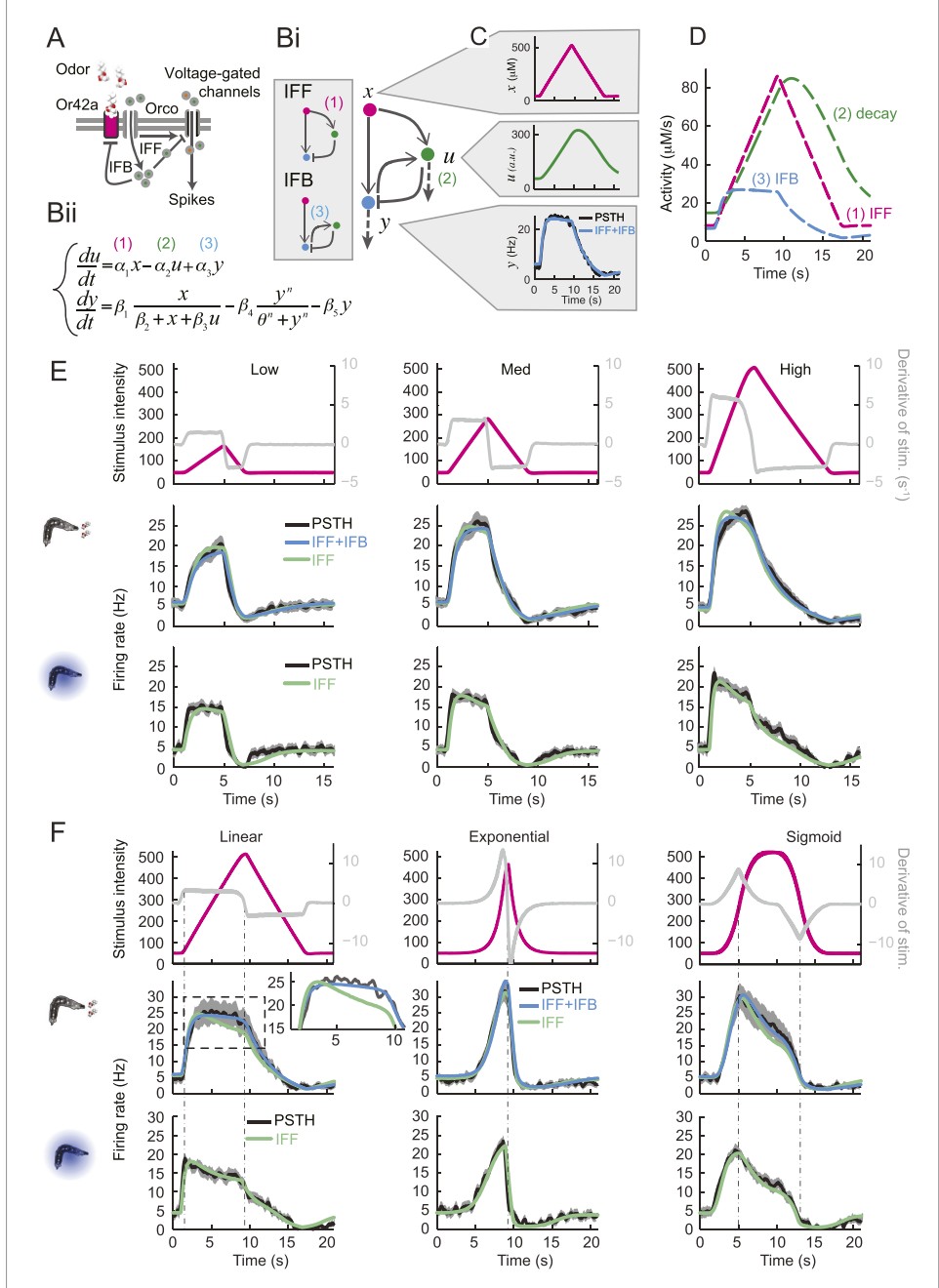

**Figure 4**. Quantitative model for signal processing achieved by a single olfactory sensory neuron. (**A**) Hypothetical physiological processes underlying the olfactory transduction pathway and spike generation. The integral feedback (IFB) motif is built on the assumption that inhibitory feedback modulates the activity of the odorant receptor, as was proposed in the adult fly (**Nagel and Wilson, 2011**). This motif appears to be essential to olfactory transduction in vertebrates (**De Palo et al., 2013**). The incoherent feed-forward (IFF motif) relies on the hypothetical existence of a delayed inhibitory effect, as was proposed for the transduction cascade of C. elegans (**Kato et al., 2014**). (**B**) Biophysical model of the olfactory transduction pathway. (**Bi**) Circuit elements combining the IFF and IFB motifs described in panel **A**. Variable $x$ represents the stimulus intensity, $u$, the activity or concentration of the intermediate variable and $y$, the firing rate of the OSN. Pathway (3) is specific to the IFB motif (light blue). (**Bii**) ODE system providing a phenomenological description of the reaction scheme outlined in panel **A** for the combination of the IFF and IFB regulatory motifs. The three pathways regulating the activity of $u$ are outlined by numbers (1)–(3). Reaction (1) corresponds to a 'production' of $u$ through the IFF branch; (2) corresponds to a first-order 'decay' of $u$; (3) corresponds to a 'production' of $u$ through the IFB branch. (**C**) Simulated activity of $u$ (green, middle) and firing rate $y$

*Figure 4. Continued*

(blue, bottom) in response to an 8-s linear odor ramp (magenta, top). Numerical simulations were achieved by integrating the ODE system described in panel **Bii** with the parameter values listed in *Table 1*. (**D**) Decomposition of the predicted activity of individual pathways contributing to the regulation of *u* for the linear odor ramp displayed in panel **C**. Activity computed from the terms (1)–(3) outlined in panel **B** for the feed-forward activation by the stimulus (IFF, 1), first-order decay (2) and coupling of the firing rate with the intermediate variable through the negative feedback (IFB, 3). Notably, the contribution of the reaction specific to the IFB motif (3) is dominated by the reaction specific to the IFF motif (1). (**E**) Fit of the solution of the ODE model for three linear stimulation ramps introduced in *Figure 3A,B* and produced with odor (middle) and light (bottom). (Top) Stimulus intensity given as odor concentration (μM). The time derivative of the concentration profile (gray lines) is given according to the *y*-axis shown on the right side of the graph. The derivative was computed after mild smoothening of the stimulus time course. The same (idealized) profile was used for the light stimulation with an intensity ranging between 15 W/m$^2$ and 207 W/m$^2$. (Middle) Comparison of the outcome of the model featuring a pure IFF motif (green) and a combination of the IFF and IFB motifs (blue). The parameters of both models were obtained independently through a Simplex optimization procedure ('Materials and methods'). For the pure IFF model, parameter $\alpha_3$ was artificially set to 0. (Bottom) Comparison of the outcome of the experimental PSTH and the model's predictions based on a pure IFF motif (green) for light stimulation. Parameter optimization shows that the IFB motif does not contribute to the light-evoked OSN dynamics. (**F**) Fit of the solution of the ODE model for three nonlinear stimulation ramps generated with odor and light. (Middle) Results of the model compared to the odor-evoked OSN activity. Close-up view of the 8-s linear ramp highlighting the differences between the behavior of the pure IFF (green) and combined IFF+IFB (blue) circuit motifs for odor stimulation. (Bottom) Comparison of the outcome of the experimental PSTH and model based on a pure IFF motif (green) for light stimulation. For all conditions shown in the figure, PSTHs were computed on a pool of a minimum of 10 recordings obtained from a minimum of 10 preparations.

The following figure supplements are available for figure 4:

**Figure supplement 1**. Response dynamics of the *Or42a*>ChR2 OSN to linear and nonlinear stimulation ramps induced with odor and light, and fits of the stimulus-to-neural (ODE) models.

**Figure supplement 2**. Validation of the stimulus-to-neural (ODE) model for the olfactory transduction cascade upon training on partial datasets.

Next, we examined the individual contribution of the IFF and IFB pathways to the response dynamics of the OSN. In *Figure 4D*, the activity of each pathway was separately computed in response to stimulation by a linear odor ramp. The contribution of the IFB motif to the dynamics of variable *u* is approximately 30% that of the IFF (cyan vs magenta curves, *Figure 4D*). This led us to conclude that the IFF pathway dominates the control of OSN spiking activity. The IFB pathway has nonetheless a non-negligible impact on the dynamics. Using the parameter optimization procedure, the pure IFF model was trained on the full set of odor ramps (*Table 1*, middle column). At a qualitative level, both the pure IFF and the composite IFF+IFB models reproduced the OSN spiking dynamics (*Figure 4* and *Figure 4—figure supplement 1*), but a quantification of the goodness of fit established the superiority of the IFF+IFB model (*Table 2* and inset of *Figure 4F*). In addition, none of the other standard 3-element circuit motifs we tested produced a reasonable fit of the OSN spiking activity (data not shown), arguing that the composite IFF+IFB model comprises essential regulatory features of the olfactory transduction cascade.

The response properties of the OSN were then studied for a family of stimulus ramps induced by light instead of odor. The temporal profiles of the light ramps were identical to the odor ramps; the intensity range was fixed to coarsely match the low firing rate of the OSN activity observed for the odor stimulations. In this regime, the temporal pattern of the OSN activity elicited by the light ramps was comparable to that elicited by the odor ramps (experimental PSTH—black lines, *Figure 4E,F* and *Figure 4—figure supplement 1*). This close similarity in the input–output relationships permitted us to substitute the odor stimulus with light. Using the full set of linear and nonlinear light ramps together with a naturalistic pattern of light stimulation (Figure 6B), optimization of the parameters of the ODE system showed that the IFB pathway does not contribute to the light-evoked response dynamics of the OSN, suggesting that the integral feedback motif is specific to the odor-evoked activity (*Table 1*). The results of the pure IFF model are in good agreement with the experimental observations (*Figure 4E,F* and *Figure 4—figure supplement 1*). Notably, the goodness of fit of the

**Table 1**. Parameters of ODE model derived from the Simplex optimization procedure ('Materials and methods') applied on the OSN spiking activity elicited by light stimulation (pure IFF model—left column) and odor stimulation (pure IFF model—middle column; combined IFF+IFB model—right column)

|  | Light: IFF motif | Odor: pure IFF motif | Odor: IFF+IFB motifs |
|---|---|---|---|
| $\alpha_1$ | 0.1 (W m$^{-2}$)$^{-1}$ s$^{-1}$ | 0.1 µM$^{-1}$ s$^{-1}$ | 0.13 µM$^{-1}$ s$^{-1}$ |
| $\alpha_2$ | 0.88 s$^{-1}$ | 0.6 s$^{-1}$ | 0.26 s$^{-1}$ |
| $\alpha_3$ | 10$^{-6}$ Hz$^{-1}$ s$^{-1}$ | 0* Hz$^{-1}$ s$^{-1}$ | 1.1 Hz$^{-1}$ s$^{-1}$ |
| $\beta_1$ | 1731.41 Hz s$^{-1}$ | 1002.25 µM s$^{-1}$ | 2903.36 µM s$^{-1}$ |
| $\beta_2$ | 1.27 W m$^{-2}$ | 8.63 µM | 0.01 µM |
| $\beta_3$ | 2.48 W m$^{-2}$ | 2.39 µM | 2.65 µM |
| $\beta_4$ | 1214.08 Hz s$^{-1}$ | 624.69 µM s$^{-1}$ | 795.62 µM s$^{-1}$ |
| $\beta_5$ | 13.03 s$^{-1}$ | 6.44 s$^{-1}$ | 23.79 s$^{-1}$ |
| $\theta$ | 0.3 Hz | 1.01 Hz | 1.88 Hz |
| $n$ | 2 | 2 | 2 |

Parameters were obtained upon training of the model on 10 stereotyped stimulus ramps (see **Figure 4—figure supplement 1**) together with the naturalistic stimulation patterns shown in **Figure 2D** (odor) or Figure 6B (light). For light stimulation, the parameter of the IFB pathway ($\alpha_3$) was negligible and considered equal to 0 in the rest of the study. For odor stimulation, parameter $\alpha_3$ was artificially set to 0 in the case of the pure IFF motif. Note that the units of the intermediate variable $u$ are undefined. We empirically found that the goodness of fit improved when the value of the offset $\beta_4$ undergoes a small correction over time. In all numerical simulations of this study, we used $\beta_4(t) = (1.023\ t^4/(t^4 + 30^4)) \times \beta_4$. The Hill coefficient $n$ was set equal to 2. In this table, all concentrations are given for odor stimulation in liquid phase. As described in the 'Materials and methods' section, the concentration equivalence in gaseous phase can be approximated by multiplying the liquid phase concentration by a factor $\rho^{\text{liquid} \rightarrow \text{gas}} = 26.73$. The parameters listed in this table are used in all numerical simulations of the study, except the validation controls described in **Figure 4—figure supplement 2**.

*parameter set artificially to 0.

pure IFF motif, when applied to both light and odor stimulations, was comparable (**Table 2**). In conclusion, the nonlinear-dynamical response properties of the OSN stimulated by odor and light ramps can be well approximated by the IFF motif, even though the IFB motif brings a non-negligible contribution to the modeling of odor-evoked response dynamics.

## Peripheral encoding of the olfactory stimulus produces transient normalization

To clarify the sensory computation achieved by the *Or42a* OSN, we sought to derive an analytical solution of the ODE system. We restricted this analysis to the pure IFF motif, which provides a good approximation of the dynamics of the composite IFF+IFB motif. The general solution of the IFF motif required solving the ODE system shown in **Figure 4Bii**. Since the OSN spiking activity evolves on a different timescale than the other two variables, the solution of the ODEs could be simplified through a quasi-steady-state approximation (QSSA, see 'Materials and methods'). The mathematical expression of the QSSA solution reveals that the OSN spiking activity ($y$) is determined by a hyperbolic ratio function of the stimulus intensity $x$:

$$y^{\text{QSSA}} = \delta_1 \frac{x}{x + \delta_2 - S(x, t)} - \delta_4, \qquad (1)$$

where $\delta_1$, $\delta_2$, and $\delta_4$ are constants ('Materials and methods'). The denominator of this hyperbolic relationship contains a scaling term $S(x, t)$ that normalizes the spiking activity by the short-term history of changes in the stimulus intensity $dx/dt$:

$$S(x, t) \propto \int_0^t e^{-\alpha_2(t - t')} \frac{dx}{dt'}\ dt'.$$

**Table 2**. Quantification of the goodness of fit between the stimulus-to-neural (ODE) models and the experimental firing dynamics of the OSN stimulated by the linear and nonlinear ramps

| | Odor, IFF | Odor, IFF+IFB | Light, IFF |
|---|---|---|---|
| A Correlation coefficient ($\rho$) | | | |
| Linear (4 s), low | 0.982 | 0.985 | 0.954 |
| Linear (4 s), med | 0.994 | 0.994 | 0.980 |
| Linear (4 s), high | 0.986 | 0.995 | 0.983 |
| Linear (4 s), slow | 0.993 | 0.994 | 0.980 |
| Linear (4 s), fast | 0.983 | 0.979 | 0.970 |
| Linear (8 s) | 0.983 | 0.994 | 0.982 |
| Quadratic | 0.993 | 0.993 | 0.988 |
| Exponential | 0.990 | 0.984 | 0.965 |
| Sigmoid | 0.985 | 0.994 | 0.990 |
| Asymptotic | 0.990 | 0.994 | 0.972 |
| B CV(RMSE) | | | |
| Linear (4 s), low | 0.143 | 0.121 | 0.225 |
| Linear (4 s), med | 0.112 | 0.097 | 0.174 |
| Linear (4 s), high | 0.159 | 0.093 | 0.184 |
| Linear (4 s), slow | 0.111 | 0.091 | 0.167 |
| Linear (4 s), fast | 0.173 | 0.179 | 0.229 |
| Linear (8 s) | 0.185 | 0.106 | 0.153 |
| Quadratic | 0.131 | 0.111 | 0.150 |
| Exponential | 0.155 | 0.226 | 0.237 |
| Sigmoid | 0.175 | 0.102 | 0.131 |
| Asymptotic | 0.124 | 0.080 | 0.193 |

**(A)** Pearson's correlation coefficient ($\rho$) computed for stimulus ramps listed in **Figure 4—figure supplement 1**.
**(B)** Coefficient of variation (CV) of the root-mean-square error (RMSE).

The integration-differentiation scaling function S($x$, $t$) plays a role similar to the 'input gain control' resulting from lateral inhibition of the local interneuron on the projection neurons in the adult antennal lobe (**Olsen et al., 2010**) with the notable difference that the rescaling takes place within the primary OSN and that it is driven by the temporal integration of changes in the stimulus intensity. In analogy to the divisive normalization reported in the visual system (**Carandini and Heeger, 2012**), we termed the rescaling operation described in **Equation 1** as 'transient normalization'. This operation appears related to the adaptive rescaling of the spike dynamics observed in adult-fly OSNs (**Kim et al., 2011**; **Nagel and Wilson, 2011**).

## Behavioral relevance of the dynamical features encoded in the OSN activity

By examining the analytical solution of the IFF motif under the quasi-steady-state approximation (**Equation 1**), we discovered that the most salient features encoded in the activity pattern of the *Or42a>*ChR2 OSN are: (1) rapid increases in firing rate triggered by abrupt positive changes in the stimulus intensity (accelerations); (2) a relaxation of the firing rate toward stationary activity when the first derivative of the stimulus is null or constant (no acceleration or deceleration); (3) decreases in firing rate in response to stimulus decelerations. In addition, we experimentally observed that (4) the spiking activity of the neuron is strongly inhibited upon abrupt return to the stimulus baseline. We asked whether these features bore any relevance to the control of run-to-turn transitions during odor gradient ascent. We hypothesized that sustained spiking activity of the OSN would suppress turning while inhibition of the OSN would facilitate turning. To test this hypothesis, we built a tracker to

monitor the position and behavioral state of a single larva in real-time at a rate of 30 Hz (*Figure 5A* and detailed description in 'Materials and methods'). Equipped with blue LEDs, the tracker was designed to evoke controlled patterns of spiking activity in the *Or42a*>ChR2 OSN by means of optogenetics. To avoid innate photophobic behavior (*Sawin-McCormack et al., 1995*; *Kane et al., 2013*), experiments were conducted on blind larvae ('Materials and methods').

We took advantage of our ability to use stereotyped light ramps to elicit predictable and reproducible patterns of firing activity in the *Or42a*>ChR2 OSN (*Figure 4*, light-evoked activity patterns). In a series of experiments, we associated individual runs with a predefined light ramp and correlated the simulated OSN firing rate with the onset of run-to-turn transitions. In the example shown in *Figure 5B*, we began each run with either an exponential ramp or a constant basal light intensity (internal control). When an exponential ramp was played to the larva, the pattern of light stimulation was executed as long as the larva remained in a run state. Upon interruption of the run, the light intensity was reset to baseline. As the motion of the larva had no influence on the stimulation pattern it experienced, this experimental protocol featured a sensorimotor loop that is essentially 'open'.

When the behavior was modulated by changes in light intensity, the majority of the runs associated with an exponential ramp did not terminate before the falling phase of the ramp. This trend was quantified through the probability of turning (or turn rate) defined as the relative number of runs that switched to a turn during a given time window of 1 s ('Materials and methods'). The turn probability was estimated at every time point by using a sliding window. Upon constant light stimulation, we found that the instantaneous turn probability was largely independent of the duration of the ongoing run (light gray line, *Figure 5C*). In contrast, the turn probability was strongly modulated by the exponential light ramp (black line, *Figure 5C*). During the rising phase of the ramp (0–8 s), turning was suppressed below the value corresponding to basal stimulation. Conversely, a sharp increase in turn probability was observed during the falling phase of the ramp. The modulation of the turn probability by the light-evoked spiking activity corroborated the idea that strong activation of the OSN efficiently suppresses turning, whereas inhibition promotes turning.

We set out to develop a quantitative model for the control of run-to-turn transitions by the neural activity. As the probability of turning remained approximately constant when the OSN activity was stationary, we hypothesized that the relationship between the OSN spiking activity and the control of run-to-turn transitions could be captured by a simple model where the time-varying probability of turning was described by the combination of a constant term ($\lambda_0$) and a term proportional to the current OSN firing rate: $\lambda_0 + \lambda_1 y(t)$. To map this linear combination (which can be positive or negative) onto the definition domain of a probability (which varies between 0 and 1), we applied a standard logit transformation and described the turn probability as a generalized linear model (GLM) (*Myers et al., 2002*):

$$\lambda(t) = \frac{1}{1 + e^{-(\gamma_0 + \gamma_1 y(t))}}.$$

To define the parameters of the GLM ($\lambda_0$ and $\lambda_1$), we transformed the previous relationship as shown in *Figure 5D*, and we carried out a linear regression on the open-loop behavior elicited by 10 light ramps identical to those used to characterize the OSN response dynamics (*Figure 5—figure supplement 1*). The parameter set obtained through this procedure is reported in *Table 3*. It was used to reproduce or predict behavioral transitions throughout the study. From here on, the GLM (*Figure 5D*) was fed with the OSN firing rate predicted from the neural model (*Figure 4B*). This model will be referred to as the integrated *stimulus-to-behavior GLM*.

For the linear and nonlinear ramps, the stimulus-to-behavior GLM accurately reproduces the time courses of the experimental turn probability (blue lines, *Figures 5E,F*). The performance of the test model was compared to a control GLM in which the turn probability was directly predicted from the stimulus intensity without any sensory processing from the OSN (dashed magenta lines, *Figure 5E,F*). For this control model, we independently fitted the same GLM with the simulated OSN firing rate replaced by the stimulus intensity ('Materials and methods'). The values of the parameters of the control model are reported in *Table 3*. The goodness of fit of the GLM was clearly contingent on the nonlinear transformation achieved by the OSN (*Table 4*). To rule out that the test GLM was overfitted, we trained the model on a subset of linear light ramps and validated the model on a set of nonlinear light ramps (*Figure 5—figure supplement 2*).

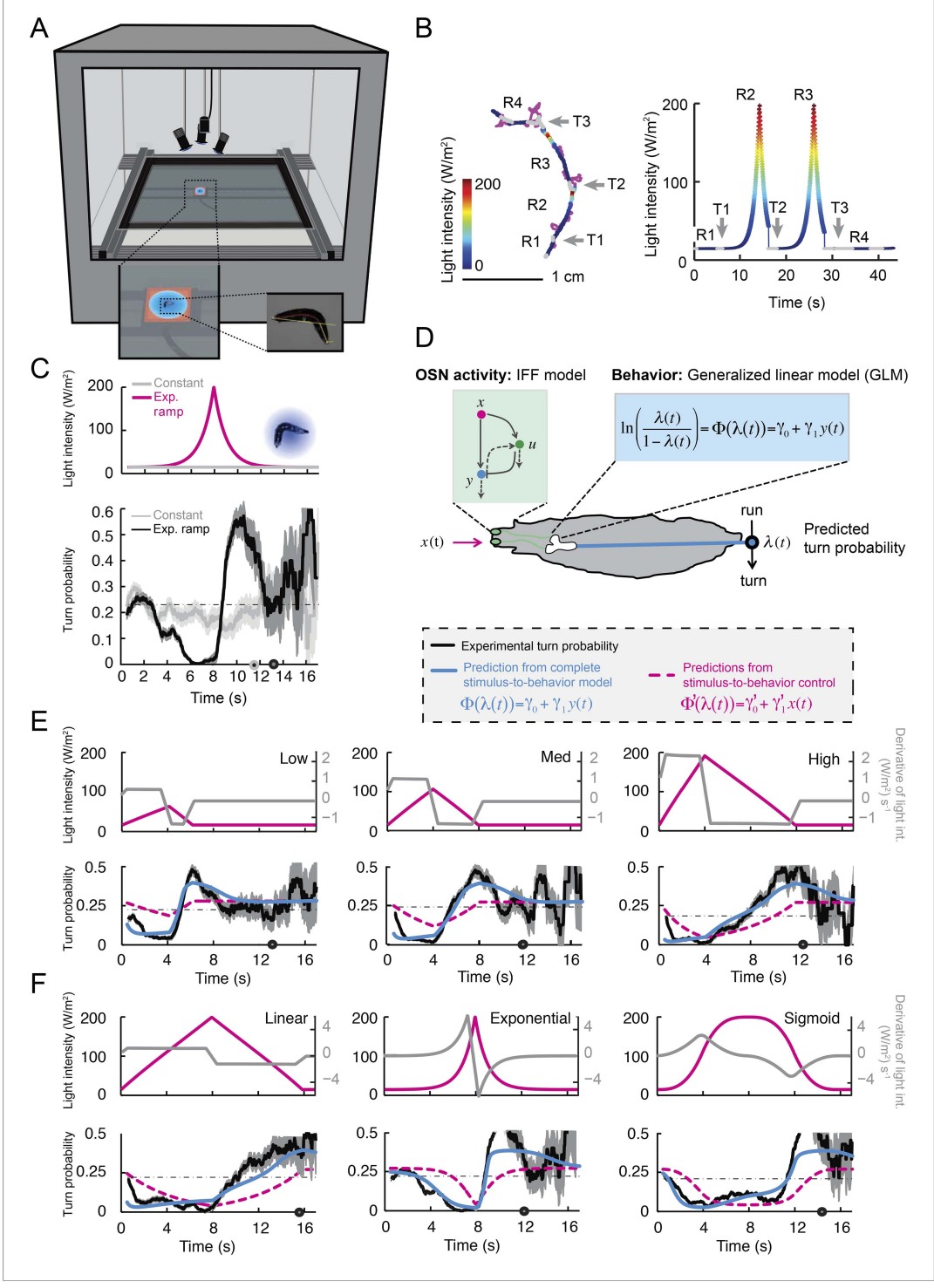

**Figure 5**. Modulation of run-to-turn transitions by light-evoked activity in the *Or42a*-expressing OSN. (**A**) Illustration of the closed-loop tracker used to synthesize virtual olfactory realities with light stimulation coupled to optogenetics. Close-up view of the larva illuminated by a red light pad fixed to a moving stage below the agarose slab. The camera and LEDs are mounted on a second moving stage whose position is updated synchronously with the bottom stage to remain locked on the position of the larva. The platform on which the larva behaves is fixed. (**B**) Presentation of the run-based light stimulation paradigm where runs are randomly assigned to constant stimulation (control) or to a test ramp with an exponential profile similar to that introduced in *Figure 3C*. (Right) Midpoint position of the larva during a trajectory with the light intensity color-coded in accordance with the color bar on the left. Illustrative runs

*Figure 5. continued on next page*

*Figure 5. Continued*

denoted as R1-4 are interspaced by turns T1-3 denoted by arrows. (**C**) Turn probability estimated from a set of runs associated with constant stimulation (light gray) or stimulation by an exponential light ramp (black). The turn probabilities are reported as the fraction of turns occurring during a 1-s window centered on the time point of interest ('Materials and methods'). Error bars are estimated from resampled sets of runs (shaded areas denote standard deviation, see 'Materials and methods'). The dashed line depicts the mean turning rate computed for constant light stimulation. The turning rate is in first approximation independent of the run duration. Small disks on the *x*-axis indicate time points after which fewer than 10% of the total number of runs are left for the constant stimulation (light gray) and exponential ramp (dark gray). Beyond these time points, the estimates of the turn probability should be considered as unreliable. (**D**) Generalized linear model (GLM) for the modulation of turn probability as a function of the sensory experience (integrated stimulus-to-behavior model). The turn probability is predicted from a linear combination of the predicted neural activity ($\gamma_1 \, y \, (t)$) and a constant term ($\gamma_0$). This linear combination is then fed into a logit transformation to convert the domain of definition of the neural activity into a probability. The two parameters of the model, $\gamma_0$ and $\gamma_1$, are determined from a linear regression on the experimental profiles of turn probability. The OSN activity driven by light, *y*, is predicted from the pure IFF (ODE) model described in *Figure 4B*. As a control, we consider the same model where the input is the stimulus intensity. The parameters of this control model are denoted as $\gamma'_0$ and $\gamma'_1$. Upon training of the test and control models on the full set of linear and nonlinear ramps (*Figure 5—figure supplement 1*), we derived the parameter values reported in *Table 3*. (**E**) Predictions of the integrated stimulus-to-behavior GLM for linear ramps of different rising and falling slopes. (Top) For the individual ramps, the time course of light intensity is shown in magenta. The time derivative of the light ramp (gray line) is computed after mild smoothening of the stimulus input. (Bottom) Behavioral predictions based on the neural activity predicted by the IFF (plain blue line) and the control model that is purely based on the stimulus (dashed magenta line). The integrated stimulus-to-behavior model clearly outperforms the predictions of the control model, which highlights the importance of the signal processing achieved by the OSN. (**F**) Predictions of the integrated stimulus-to-behavior model for 8-s light ramps. For all conditions tested, the experimental turn probability was estimated on a sample of 490–970 runs. Quantification of the goodness of fit is reported in *Table 4* for the test and control models.

The following figure supplements are available for figure 5:

**Figure supplement 1**. Fit of the integrated stimulus-to-behavior generalized linear model (GLM) for the linear and nonlinear light ramps.

**Figure supplement 2**. Validation of the integrated stimulus-to-behavior generalized linear model (GLM) upon training on a partial set of light ramps.

**Figure supplement 3**. Comparison of the predictions of the integrated and control generalized linear model (GLM) with and without the contribution of the first derivative of the stimulus intensity.

The successful application of the stimulus-to-behavior GLM led us to conclude that: (1) stationary levels of OSN firing rate lead to probabilistic transitions from run to turn. When the OSN spiking activity remains constant, the probability of turning at a given time is largely independent of the duration of the run; (2) excitation of the OSN suppresses turning (evident during rising phase of all ramps); (3) inhibition of the OSN facilitates turning (most evident during falling phase of the exponential ramp). Consistent with our finding that the OSN activity is sensitive to the slope of the ramp, we found that the performance of the control GLM was improved by combining the light intensity (*x*) with its first derivative (d*x*/d*t*) (*Figure 5—figure supplement 3*). For the majority of ramps, this improvement did, however, not match the quality of the fit produced by the integrated stimulus-to-behavior GLM (*Table 4*). We concluded that the nonlinear response characteristics of the OSN have a noticeable influence on the control of orientation behavior.

## Predicting run-to-turn transitions in virtual olfactory gradients

To test the relevance of the integrated stimulus-to-behavior GLM in conditions in which the sensorimotor loop is closed, we synthesized a controlled light gradient with a shape comparable to that of the odor gradient (*Figure 6A*). In this stimulation paradigm, the light intensity was continuously updated based on the position of the larva ('Materials and methods'). *Figure 6A* illustrates the

**Table 3**. Parameters of the stimulus-to-behavior generalized linear model (GLM) obtained upon training of model on the stimulation ramps listed in **Figure 5—figure supplement 1**

| | **Control without first derivative of stimulus** | **Control with first derivative of stimulus** | **Test model with predicted neural activity** |
|---|---|---|---|
| $\gamma_0$ (constant) | −0.8156 | −0.8200 | −0.3534 |
| $\gamma_1$ (input variable) | −0.0114 $(W/m^2)^{-1}$ | −0.0013 $(W/m^2)^{-1}$ | −0.1523 $Hz^{-1}$ |
| $\gamma_2$ (derivative of input variable) | – | −0.0214 $(W/m^2)^{-1}$ s | – |

The first two columns of the table report the value of the stimulus-to-behavior control model without (left) and with (center) the contribution of the first derivative of the stimulus. The last column reports the value of the integrated stimulus-to-behavior model fed with the predicted firing rate of the OSN. The parameters listed in this table are used in all numerical simulations of the study, except for the validation controls described in **Figure 5—figure supplement 2**.

behavior of an *Or42a*>ChR2 larva in a light gradient. As observed for the odor-evoked behavior (*Gomez-Marin et al., 2011*), the larva ascended the light gradient and remained in the vicinity of its peak by implementing a series of runs and directed turns. In *Figure 6B*, we examined how the *Or42a*>ChR2 OSN responds to a replay of the light intensity changes experienced during the trajectory shown in *Figure 6A*. The spiking activity of the OSN displayed considerable processing of the stimulation pattern. This transformation of the stimulus was well captured by the IFF model. To predict the temporal evolution of the turn probability associated with individual runs, we fed the predicted spiking activity of the OSN into the GLM trained on the open-loop light ramps (*Figure 5* and *Table 3*). Correlating the predictions of the model with the termination of the actual runs revealed that the initiation of a turn was typically preceded by a steady increase in the predicted probability of turning (*Figure 6B,C* and *Video 1*). To quantify this trend, we analyzed a large set of runs included in 25 trajectories, each trajectory corresponding to a different animal (representation of a subset of 10 trajectories in *Figure 6D*). Since every run corresponded to a unique sensory experience, the predictions of the stimulus-to-behavior GLM could only pertain to the average behavior observed over multiple runs. We therefore analyzed the averaged trend of the turn probability preceding individual turns.

As reported in previous work (*Gomez-Marin et al., 2011*), we found that the stimulus intensity decreases steadily for several seconds prior to a turn (data not shown). Accordingly, the stimulus-to-behavior GLM predicted that the stimulus downslope was transformed into a monotonic increase in turn probability (light blue line, *Figure 6E*). To establish the sensorimotor control underlying this trend, we computed the turn-triggered averages of the turn probability by using two control models ('Materials and methods'): (1) behavioral predictions based on the assumption that the OSN spiking activity remained constant throughout the trajectory and (2) behavioral predictions upon uncoupling of the stimulus and the behavior by temporally inverting the reconstructed time course of the stimulus. In contrast to the test GLM, neither control displayed a substantial increase in turn probability prior to turning (red and back dashed lines, *Figure 6E*). The significance of the improvement in the predictive power of the test model relative to the controls was established by comparing the log-likelihood computed over the entire set of runs (*Figures 6F*, bottom panel and 'Materials and methods'). This allowed us to conclude that the integrated neural-to-behavior model built on controlled conditions of stimulation (open-loop paradigm) was sufficient to predict run-to-turn transitions arising from free behavior in a virtual odor gradient (closed-loop paradigm).

## Inhibition of OSN spiking activity facilitates turning during free behavior

Next, we tested the idea that inhibition of the OSN activity at the stimulus offset is sufficient to trigger a nearly deterministic release of turning during free behavior. To this end, we designed radially symmetrical light landscapes with geometrical features producing inhibition or maintenance of OSN activity during free motion. As a control, we considered a landscape with an exponential rise interrupted by an exponential fall at a fixed distance of 8 mm from the center (*Figure 7A*, top panel, 'rim' indicated by a dashed line). The shape of this landscape is reminiscent of a 'volcano'. The geometry of the landscape was chosen such that a larva moving at a speed of 1 mm/s from the foot of

**Table 4.** Quantification of the goodness of fit of the control and the test generalized linear model (GLM)

| | Control GLM <u>without</u> derivative | Control GLM <u>with</u> derivative | Test GLM |
|---|---|---|---|
| **A Correlation coefficient (ρ)** | | | |
| Linear (4 s), low | 0.69 | 0.75 | 0.89 |
| Linear (4 s), med | 0.62 | 0.90 | 0.90 |
| Linear (4 s), high | 0.67 | 0.96 | 0.95 |
| Linear (4 s), slow | 0.54 | 0.92 | 0.91 |
| Linear (4 s), fast | 0.65 | 0.76 | 0.90 |
| Linear (8 s) | −0.05 | 0.66 | 0.78 |
| Quadratic | 0.11 | 0.70 | 0.88 |
| Exponential | 0.29 | 0.43 | 0.92 |
| Sigmoid | 0.13 | 0.61 | 0.60 |
| Asymptotic | −0.10 | 0.25 | 0.03 |
| *All conditions (all time points included)* | *0.53* | *0.74* | *0.86* |
| **B CV(RMSE)** | | | |
| Linear (4 s), low | 0.59 | 0.54 | 0.33 |
| Linear (4 s), med | 0.60 | 0.41 | 0.32 |
| Linear (4 s), high | 0.75 | 0.41 | 0.33 |
| Linear (4 s), slow | 0.56 | 0.34 | 0.27 |
| Linear (4 s), fast | 0.53 | 0.45 | 0.31 |
| Linear (8 s) | 1.06 | 0.71 | 0.59 |
| Quadratic | 1.06 | 0.73 | 0.55 |
| Exponential | 0.84 | 0.85 | 0.39 |
| Sigmoid | 0.95 | 0.65 | 0.58 |
| Asymptotic | 0.85 | 0.52 | 0.97 |
| *All conditions (all time points included)* | *0.65* | *0.51* | *0.39* |

Comparison of the performances of the integrated stimulus-to-behavior GLM and the control model bypassing the OSN processing. The outputs of the test and control GLMs are obtained based on the parameter sets listed in **Table 3**. (A) Application of Pearson's correlation coefficient (ρ) on the time course of the experimental turn probability and the simulated turn probability. Restriction of the quantification to the first 12 s of the ramp where the experimental estimate of the turn probability is reliable. (B) Same as panel **A** for the coefficient of variation of the RMSE. The goodness of fit computed for the entire set of ramps is reported at the bottom of the table for both metrics.

the gradient toward its center would experience a light pattern similar to the 8-s exponential ramp (*Figure 5F* and *Figure 7Ai*). In response to an exponential ramp, the spiking activity of the OSN featured a steady increase followed by a rapid decrease. We therefore expected to observe turn suppression during the rising phase of the ramp and turn facilitation during the falling phase of the ramp. The tendency of larvae to initiate turning upon crossing of the volcano's rim was evident from the set of runs that moved from the outer to the inner edge of the volcano (*Figure 7Aii*, bottom panel). The alternation between turn suppression and turn facilitation resulted in a zigzagging of trajectories across the rim of the volcano. The integrated stimulus-to-behavior GLM predicted a rise in the turn probability prior to the interruption of a run (*Figure 7E* and *Video 2*).

We then considered an extreme version of the volcano: a well (*Figure 7B*). In this landscape, the light intensity experienced by a larva moving toward the center of the well corresponded to an exponential rise followed by a near-instantaneous drop. The IFF model correctly reproduced the strong inhibitory phase experimentally observed in the OSN spiking activity (*Figure 7Bi*, bottom panel). This inhibition was expected to generate a nearly deterministic release of turns. Consistently, larvae avoided the well region (*Figure 7Bii*). Such a behavior was correctly described by the GLM,

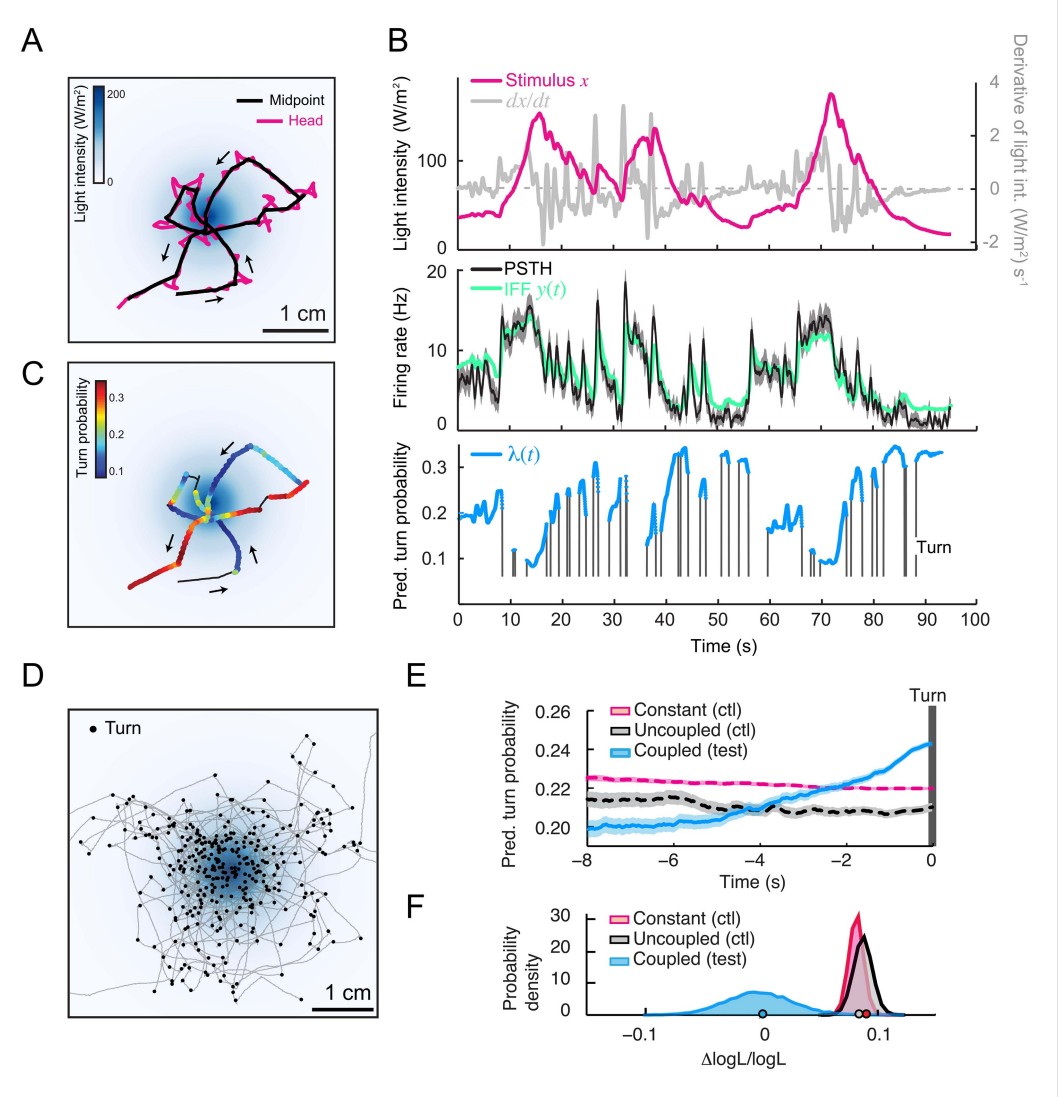

**Figure 6**. Predictions of the integrated stimulus-to-behavior generalized linear model (GLM) for run-to-turn transitions observed in a virtual olfactory gradient. (**A**) Synthetic chemotaxis in a virtual odor gradient produced by light stimulation. The larva experiences a light intensity determined by a predefined stimulus landscape. The landscape displayed in the background of panel **A** is an exponential gradient centered on a point 'source'. Larvae responding to this light gradient accumulate at the gradient's peak as observed for odor gradients. Illustrative trajectory of the midpoint (black) and head (magenta). Black arrows indicate the direction of motion. (**B**) Sensorimotor analysis of a representative trajectory. (Top) Time course of the light intensity associated with the trajectory displayed in panel **A**. (Middle) PSTH of the OSN activity measured experimentally upon a replay of the intensity time course at the electrophysiology rig (black line). Numerical simulations of the neural activity carried out by the IFF motif (green line) presented in *Figure 4B*. (Bottom) Turn probability (blue line) predicted from the integrated stimulus-to-behavior GLM presented in *Figure 5D* (parameter set listed in *Table 3*). The neural activity simulated in the middle panel is fed into the GLM to predict the turn probability shown in the bottom panel. Behavioral predictions are only shown for the sequences associated with runs. (**C**) Overlay of the trajectory of the midpoint with the predicted turn probability color-coded in accordance with the color bar on the left. We observe that the turn probability tends to increase (red color range) when the larva is moving away from the gradient's peak, whereas it decreases (blue color range) when the larva is moving toward the peak. (**D**) Overlay of 10 trajectories recorded in the exponential light gradient shown in panel **A**. For each trajectory, the position of the midpoint is shown in gray. Turns are indicated as small black circles. (**E**) Turn-triggered average of the predicted turn probability for the exponential light gradient. A comparison is made between predictions based on the simulated OSN activity driven by the stimulus intensity (test model, blue line), predictions based on the simulated OSN activity driven by the time-reversed stimulus time course (uncoupled control, black line), and predictions based on the assumption that

*Figure 6. continued on next page*

*Figure 6. Continued*

the neural activity stays constant over the course of each trajectory (constant control, magenta line). The parameters of the stimulus-to-behavior GLM are listed in *Table 3*. We observe that the turn probability steeply increases 4 s before the turn, which coincides with the median duration (3.8 s) of the entire set of runs. Analysis conduced over 750 runs with a duration of minimum 1 s. Shaded areas represent SEM. (**F**) Log-likelihood of the predictions of the stimulus-to-behavior test GLM compared to the controls. Bootstrap analysis of the difference in log-likelihood (logL) between the test model and the controls normalized by the log-likelihood of the test model ($\Delta$logL/logL$_{test}$). Distribution of the relative difference in logL is shown for the test model against the constant neural activity control (red), and the uncoupled stimulus control (black). The median of the distribution is equal to the value obtained from the original full set of runs; the median of the entire distribution is indicated by a dot in the x-axis. As an internal control, the test model was compared to itself (blue). Out of 10,000 resampled subsets of runs, none of the controls was found to be more likely than the test model ($p < 0.0001$). The analysis included all observed runs with a duration of minimum 1 s (750 runs originating from 25 trajectories).

which predicted a dramatic increase in turn probability following the crossing of the rim (*Figure 7E* and *Video 3*). To probe the idea that sustained OSN spiking activity suppresses turning, we synthesized a landscape complementary to the well: a 'mesa' in which the light intensity at the rim was extended to the central area of the landscape (*Figure 7C*). During the transition from an exponential rise in light intensity to a plateau value, the OSN underwent a mild drop in spiking activity before a stationary value was reached—a feature accurately reproduced by the IFF motif (*Figure 7Ci*). The stimulus-to-behavior GLM predicted a modest increase in turn probability upon crossing of the rim without significant avoidance of the central area of the mesa (*Figure 7E* and *Video 4*). This prediction was corroborated by our experimental results (*Figure 7Cii*, bottom panel).

Finally, we considered an intermediate landscape consisting of a linear 'hat' (a cone) in which runs moving toward the center underwent a deceleration in stimulus intensity during the transition from the exponential rise to the linear rise ('linear', *Figure 7D*). Due to the sensitivity of the OSN activity to deceleration in the stimulus intensity, the linear hat landscape led to a modest drop in firing rate similar to that observed for the mesa (*Figure 7Di*). The IFF model faithfully reproduced this counterintuitive observation. At a behavioral level, the stimulus-to-behavior GLM predicted no significant difference between the behavior evoked by the mesa and the linear hat landscapes (*Figure 7E* and *Video 5*). These predictions were in good agreement with the free behavior of larvae (*Figure 7Dii*, bottom panel).

To assess the predictive power of the integrated stimulus-to-behavior GLM, we compared the average turn probability preceding a turn (*Figure 7E*) with the observed latency to turning upon crossing of the rim (*Figure 7F*). The well was associated with the prediction of the steepest increase in turn probability, leading to the expectation that most runs stopped within 2 s of the rim crossing. The volcano led to a milder increase in the predicted turn probability for about 3 s, while the mesa and linear hat were predicted to generate an even weaker increase. As shown in *Figure 7F*, we found that the average latency to turn was shortest for the well landscape (0.93 s) followed by the volcano (3.48 s). We observed significantly longer turn latencies for the mesa and linear hat with no difference

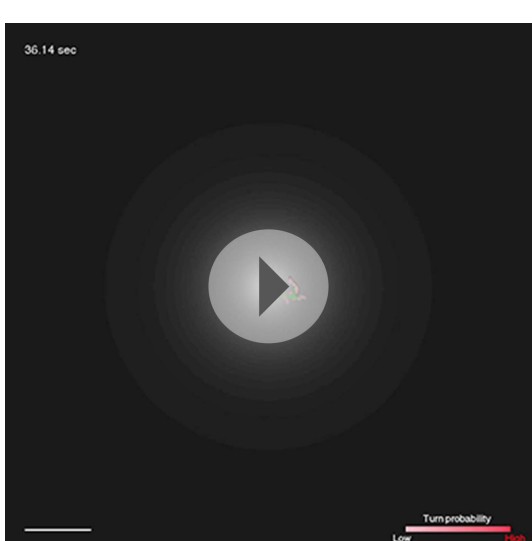

**Video 1.** Illustrative trajectory sequence in an exponential light gradient with predicted turn probability colored in red. The scale bar at the bottom left of the Video represents 1 cm. The green trace displays a 20 s segment of the past positions of the centroid. White spots represent the position of the head every 20 frames (or 0.66 s). The behavioral sequence is accelerated by a factor 3.

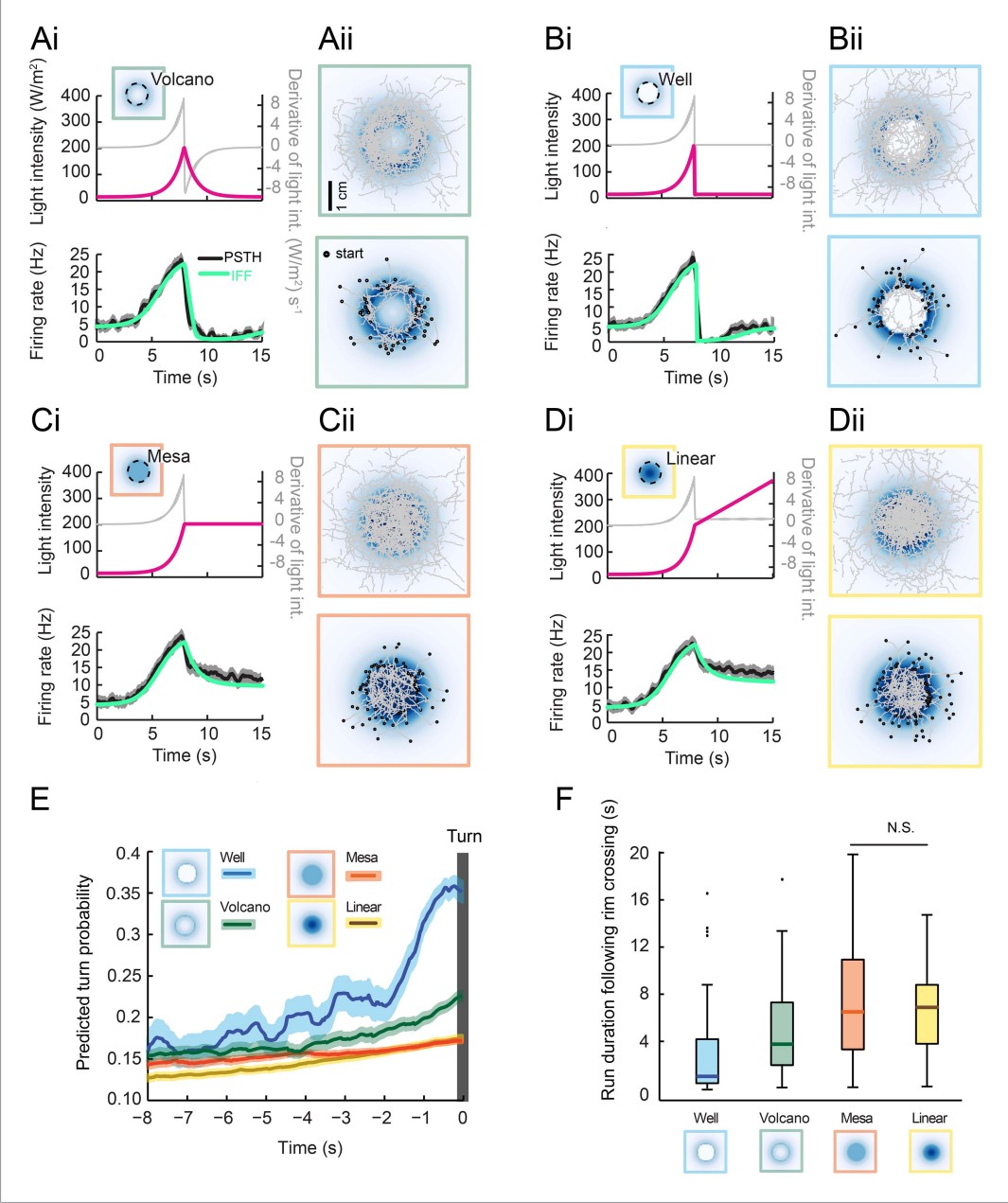

**Figure 7**. Predictions of run-to-turn transitions elicited by a family of radially symmetrical light landscapes. (**Ai–Di**) Stereotyped OSN responses to light ramps starting with a common 8-s exponential rise: ramp with a smooth exponential fall ('volcano', see panel **A**), ramp with an abrupt fall to basal intensity ('well', see panel **B**), ramp with a prolongation of the maximum intensity ('mesa', see panel **C**), and ramp with a linear increase ('linear', see panel **D**). The spiking activity of the OSN is computed from the pure IFF (ODE) model presented in *Figure 4B* (parameter set listed in *Table 1*). Time course of the light intensity shown in magenta; first derivative of the light intensity shown in gray; simulated OSN activity shown in green. (**Ai**) Experimental and predicted OSN activity elicited by an exponential rise followed by an exponential fall. (**Aii**) Symmetrical two-dimensional light landscapes corresponding to the exponential 'volcano' profile described in panel **Ai**. (Top) Set of 54 trajectories superimposed onto the stimulus landscape. (Bottom) Set of 48 runs starting from the external edge of the 'volcano' and heading toward its center (minimum run duration: 1 s). (**Bi**) Same as **Ai** for the 'well' profile. Strong inhibition of the OSN activity follows the abrupt fall in light intensity. (**Bii**) Crossing of the rim leads to an aversive response. As a consequence, larvae avoid the well at the center of the landscape. Set of 42 trajectories represented in the diagram at the top; set of 63 entering runs represented in the diagram at the bottom. (**Ci**) Same as **Ai** for the 'mesa' profile. The transition from an exponential rise in light intensity to constant intensity leads to a transient drop in neural activity before a steady state

*Figure 7. continued on next page*

*Figure 7. Continued*

value is reached. (**Cii**) For the mesa landscape, crossing of the rim does not lead to an aversive response: larvae tend to maintain their ongoing run. The center of the landscape is therefore visited. Set of 36 trajectories represented in the diagram at the top; set of 79 entering runs represented in the diagram at the bottom. (**Di**) Same as **Ai** for a 'linear' hat profile. The deceleration in stimulus from an exponential rise to a linear rise leads to a transient drop in neural activity before a steady state value is reached. The corresponding OSN dynamics is similar to that elicited by the mesa (panel **Ci**). (**Dii**) Upon crossing of the rim of the linear hat landscape, larvae undergo a deceleration in light intensity that is expected to modulate behavior in a way similar to the mesa landscape. Set of 47 trajectories represented in the diagram at the top; set of 72 entering runs represented in the diagram at the top. (**E**) Turn-triggered averages of the predicted turn probability for the subset of runs entering the landscape's central area (bottom graphs of **Aii–Dii**). Each run included in the analysis crosses the rim of the landscape (minimum run duration: 1 s). Shaded areas represent SEM. (**F**) Distribution of run durations following the crossing of the landscape's rim. Analysis restricted to the subset of runs described in the diagram at the tops of panels **Aii–Dii** (runs entering the central area of the landscape). The experience of an abrupt fall in light intensity promotes rapid turning ('well' condition), whereas runs are elongated by constant light stimulation or by a linear rise in light intensity ('mesa' and 'linear' landscapes). Differences between the median of the run durations associated with each of the four landscapes is assessed through a Kruskal–Wallis test ($p < 10^{-10}$) followed by pair-wise Wilcoxon tests with a Bonferroni correction (for the non-significant difference $p > 0.05/6$; for all other pairwise comparisons $p < 0.05/6$). The behavior predicted from turn probability (panel **E**) is in good agreement with the shortening or elongation of the runs observed for each of the four landscapes.

between the two conditions (6.6 s and 6.7 s, respectively). In conclusion, the use of synthetic light landscapes permitted us to experimentally demonstrate that sustained OSN spiking activity suppresses turning during free behavior, whereas inhibition of the OSN activity promotes run-to-turn transitions in a nearly deterministic manner. This approach established that the relatively simple linear control underlying the GLM trained on the behavior of larvae experiencing stereotyped open-loop light stimulations is sufficient to account for the control of behavior elicited under conditions of closed-loop light stimulation.

**Video 2.** Illustrative trajectory sequence in 'volcano' light gradient with predicted turn probability colored in red. The scale bar at the bottom left of the Video represents 1 cm. The green trace displays a 20 s segment of the past positions of the centroid. White spots represent the position of the head every 20 frames (or 0.66 s). The behavioral sequence is accelerated by a factor 3.

## Sensorimotor control can be predicted for free behavior in odor gradients

Our ultimate goal was to test the ability of the integrated stimulus-to-behavior model to predict the duration of runs in an odor gradient. In a real odor landscape (*Figure 8A*), larvae accumulated at the peak of the gradient with a dispersal notably larger than that observed in a light gradient (*Figure 6D*). This apparent decrease in orientation performances can be partly explained by the shallower geometry of the odor gradient (*Figure 8—figure supplement 1*). In *Figure 8B*, the goodness of fit between the spiking activity of the OSN and the output of the IFF+IFB model can be appreciated for the representative trajectory highlighted in *Figure 8A* (magenta trace). To predict run-to-turn transitions during free motion in a real odor gradient, we replaced the pure IFF model devised for light-evoked spiking activity by the composite IFF+IFB model as input for the stimulus-to-behavior GLM trained on the open-loop light ramps (*Figure 5D* and *Table 3*). The model predicted that runs were on average associated with a monotonic increase in the turn

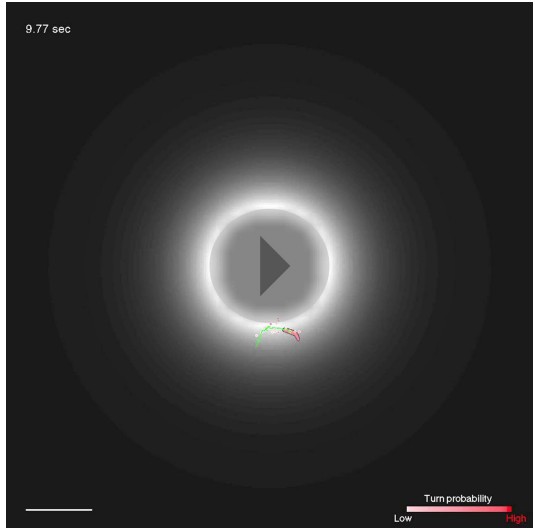

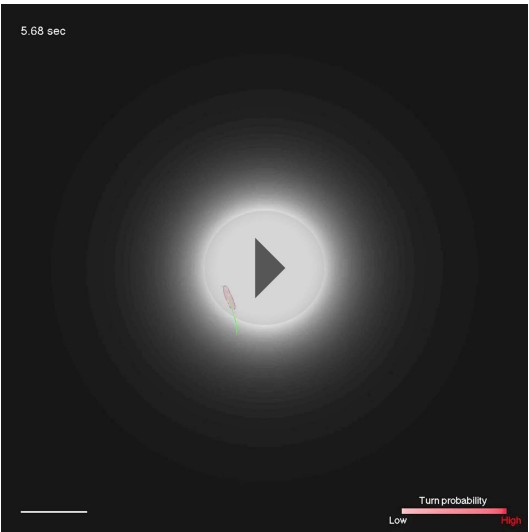

**Video 3.** Illustrative trajectory sequence in the 'well' light gradient with predicted turn probability colored in red. The scale bar at the bottom left of the Video represents 1 cm. The green trace displays a 20 s segment of the past positions of the centroid. White spots represent the position of the head every 20 frames (or 0.66 s). The behavioral sequence is accelerated by a factor 3.

**Video 4.** Illustrative trajectory sequence in the plateau light gradient with predicted turn probability colored in red. The scale bar at the bottom left of the Video represents 1 cm. The green trace displays a 20 s segment of the past positions of the centroid. White spots represent the position of the head every 20 frames (or 0.66 s). The behavioral sequence is accelerated by a factor 3.

probability during several seconds before a turn (*Figure 8C*). Based on these predictions, we computed the likelihood of the ensemble of runs observed in the odor gradient. This likelihood was significantly larger than that computed for two control models (*Figure 8D*). Together, these results establish that the structure and parameters of the integrated stimulus-to-behavior GLM form a solid conceptual basis to describe how the sensory dynamics of single OSNs influence run-to-turn transitions during naturalistic behavior (*Video 6*).

## Discussion

Most primary sensory neurons operate differently from proportional counters (*Rieke, 1997*; *Song et al., 2012*). Individual OSNs of *C. elegans* and cockroaches function as bipolar detectors that selectively respond to either increases or decreases in stimulus intensity (*Tichy et al., 2005*; *Chalasani et al., 2007*). A similar specialization into ON-OFF detection pathways has been observed for thermotaxis in *C. elegans* (*Suzuki et al., 2008*) and motion perception in adult flies (*Joesch et al., 2010*). In contrast with these binary sensory responses, we discovered that a single larval OSN is sensitive to both the stimulus intensity and its first derivative. The enhanced information-processing capacity of primary olfactory neurons in the larva is consistent with the response characteristics of OSNs in adult flies, which encode complex dynamical features of airborne odorant stimuli (*Kim et al., 2011*; *Martelli et al., 2013*).

To describe the input–output response properties of single larval OSNs, we set out to build a biophysical model of the olfactory transduction pathway. IFB motifs constitute the core mechanism of chemoreception in bacteria, olfactory transduction, and phototransduction (*Yi et al., 2000*; *De Palo et al., 2013*). In adult flies, *Nagel and Wilson (2011)* investigated how the potential involvement of negative feedback on the olfactory transduction cascade could account for dynamical and adaptive features of OSN response. On the other hand, IFF motifs are implicated in the regulation of numerous cellular and developmental processes (*Goentoro and Kirschner, 2009*; *Lim et al., 2013*), and their contribution to sensory processing has been documented in recent work (*Kato et al., 2014*; *Liu et al., 2015*). These results led us to conjecture that two regulatory motifs might be involved in larval olfactory transduction: an IFB and an IFF featuring direct excitation and indirect inhibition (*Figure 4A*,

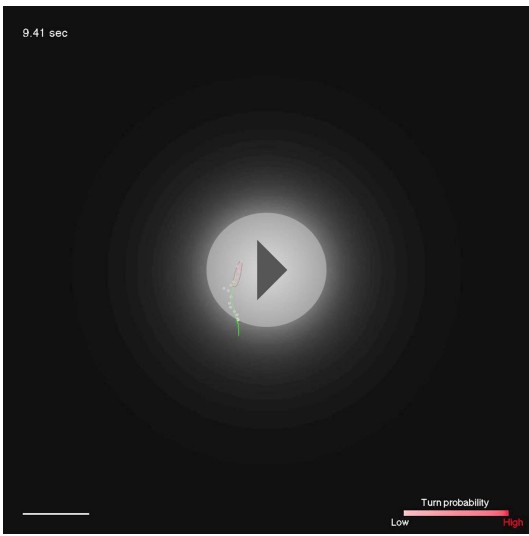

**Video 5.** Illustrative trajectory sequence in the linear 'hat' light gradient with predicted turn probability colored in red. The scale bar at the bottom left of the Video represents 1 cm. The green trace displays a 20 s segment of the past positions of the centroid. White spots represent the position of the head every 20 frames (or 0.66 s). The behavioral sequence is accelerated by a factor 3.

*B*). Using a parameter optimization approach, we found that a pure IFF motif is sufficient to approximate the response properties of the OSN. Combining the IFF and IFB motifs was nonetheless necessary to recapitulate the richness of OSN dynamics elicited by naturalistic olfactory stimuli (*Figure 4* and *Figure 4—figure supplement 1*). Consistent with the model proposed by *Nagel and Wilson (2011)*, our numerical simulations indicate that the integral feedback applies to the signaling pathway specific to the odorant receptor (OR). Nagel and Wilson have suggested that a diffusible effector—potentially intracellular calcium—inhibits the activity of the OR, thereby affecting the onset and offset kinetics of the OSN response. By contrast, the IFF motif would describe a regulatory mechanism acting on components of the transduction pathway downstream from the OR (*Gu et al., 2009*). It is plausible that the IFF regulation is also mediated by intracellular calcium.

What features of the olfactory stimulus are encoded in the spiking dynamics of single larval OSNs? Our biophysical model of the olfactory transduction cascade shows that the spiking activity of the OSN follows a standard hyperbolic dose-response when stimulated by prolonged pulses of odor ('Materials and methods'). In this regime, the maximum OSN firing rate we observed for IAA is modest (*Figure 3—figure supplement 1*). Changes in odor concentration occurring on a timescale relevant to the behavior—a second or shorter—can produce significantly higher (or lower) firing rates. This sensitivity to positive and negative changes in stimulus intensity can be explained by the mathematical solution we derived for the OSN dynamics (*Equation 1*). Upon changes in odor concentration, the dose–response function describing the OSN spiking activity is transiently rescaled (or 'normalized') by the short-term history of the stimulus derivative ('memory' on characteristic time scale of 1 s, see 'Materials and methods'). As a result, positive derivatives in stimulus intensity excite the OSN. Negative derivatives can inhibit the OSN firing rate in a manner consistent with the stimulus-offset inhibitions observed in adult-fly OSNs (*Hallem et al., 2004*; *Nagel and Wilson, 2011*). Our model indicates that a single *Or42a* OSN combines the function of a slope (ON) detector in response to positive gradients and an OFF detector in response to negative gradients. When larvae ascend Gaussian odor gradients originating from single odor sources, we thus expect high OSN firing rates. Robust inhibition of OSN spiking activity would result from motion that takes larvae down the odor gradient.

How relevant are the features encoded by the spiking activity of the *Or42a* OSN to the behavioral dynamics directing chemotaxis? To address this question, we substituted the odor stimulation with optogenetics-based light stimulation and gained unprecedented control over the spiking activity evoked in a genetically targeted OSN. Under the conditions of open-loop light stimulation, we found that OFF responses (offset inhibition of the OSN firing) promote turning, whereas ON responses (sustained high firing) suppress turning (*Figure 5*). We applied a GLM to describe the link between the OSN spiking dynamics and the probability of switching from a run to a turn (*Figure 5D*). The accuracy of the model's output showed a striking dependence on the nonlinear transformation achieved by the olfactory transduction cascade (*Figure 5—figure supplement 3*). Ultimately, we combined the biophysical model for the OSN spiking dynamics with the GLM to make robust predictions about closed-loop behavior in virtual and in real odor gradients (*Figures 6–8*).

The integrated stimulus-to-behavior GLM clarifies how features encoded in the activity pattern of individual primary olfactory neurons influence behavioral dynamics. The information transmitted by

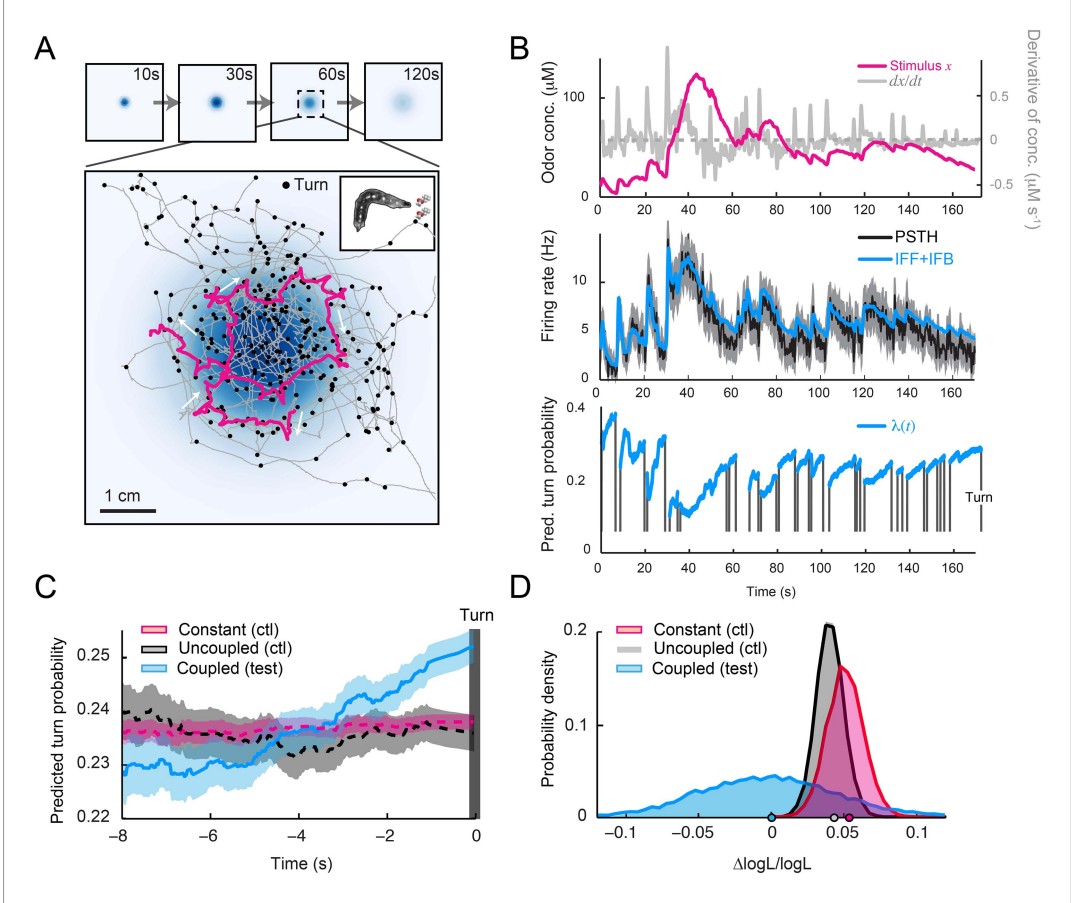

**Figure 8**. Predictions of the integrated stimulus-to-behavior model for run-to-turn transitions observed in a real odor gradient. (**A**) Superimposition of 10 consecutive trajectories observed in an odor gradient of isoamyl acetate (same experimental conditions as *Figure 1*). For every trajectory, the position of the midpoint is shown in gray. Small black circles indicate turns. The head position of the trajectory presented in *Figure 1A* is highlighted in magenta. Arrows indicate the direction of motion. The odor gradient shown in the background corresponds to the reconstructed snapshot 60 s after the onset of the diffusion process ('Materials and methods'). (**B**) Sensorimotor analysis of a representative trajectory. (Top) Time course of the reconstructed odor concentration associated with the trajectory displayed in panel **A**. (Center) PSTH of the OSN measured experimentally in response to a replay of the odor concentration course (black line). Neural activity simulated by the composite IFF+IFB ODE model (blue line) presented in *Figure 4B* (parameter set listed in *Table 1*). (Bottom) Turn probability (blue line) predicted by the stimulus-to-behavior GLM trained on the light-evoked open-loop behavior reported in *Figure 5—figure supplement 1* (parameter set listed in *Table 3*). The neural activity simulated in the middle panel is fed into the GLM to predict the turn probability shown in the bottom panel. Behavioral predictions are only shown for the sequences associated with runs. The predicted turn probability is only shown for the behavioral sequences associated with runs. (**C**) Turn-triggered average of the predicted probability of turning for behavior observed in the odor gradient. A comparison is made between predictions based on the simulated OSN activity driven by the stimulus intensity (coupled test, blue), predictions based on the simulated OSN activity driven by the time-reversed stimulus time courses (uncoupled control, black), and predictions based on the assumption that the neural activity stays constant over the course of each trajectory (constant control, red). As for the light gradient, we observe that the predicted turn probability increases 5 s before the turn, which coincides with the median duration (5.4 s) of the entire set of runs. Since the geometry of the odor gradient is shallower than the light gradient (*Figure 8—figure supplement 1*), the increase in turn probability has a reduced amplitude compared to the behavior elicited by the light gradient (*Figure 6E*). Shaded areas denote SEM. (**D**) Log-likelihood of the predictions of the integrated stimulus-to-behavior GLM compared to the controls. Bootstrap analysis of the difference in log-likelihood (logL) of the test model and the controls normalized by the log-likelihood of the test model ($\Delta$logL/logL$_{test}$). Distribution of the relative difference in logL computed for the test model against itself (blue), against the constant neural activity control (red), and against the uncoupled stimulus control (gray). The median of the distribution is equal to the value obtained from the original full set of runs; the median of the entire distribution is indicated by a dot in the x-axis. Based on 10,000 resampled

*Figure 8. continued on next page*

*Figure 8. Continued*

subsets of runs, we conclude that the test model is significantly larger than both controls (p = 0.0001 for the constant neural activity control and the uncoupled stimulus control). For panels **C** and **D**, the analysis includes all runs with a duration of minimum 1 s (304 runs originating from 20 trajectories).

The following figure supplement is available for figure 8:

**Figure supplement 1**. Comparison of the geometry of the exponential light gradient and the odor gradient.

a single larval OSN is sufficient to represent positive and negative odor gradients through the excitation and inhibition of spiking activity. Unlike for chemotaxis and thermotaxis in *C. elegans* where the ON and OFF pathways are associated with different cellular substrates (*Chalasani et al., 2007*; *Suzuki et al., 2008*), the same larval OSN is capable of controlling up-gradient and down-gradient sensorimotor programs. This observation echoes findings recently made for thermotaxis in the *Drosophila* larva (*Klein et al., 2015*). Furthermore, it corroborates the idea that sensory representations are rapidly transformed into motor representations in the circuit controlling chemotaxis (*Luo et al., 2014*).

In the future, it will be important to define whether the sensorimotor principles proposed for the *Or42a* OSN can be generalized to OSNs expressing other odorant receptors (*Fishilevich et al., 2005*; *Kreher et al., 2008*; *Mathew et al., 2013*). In addition, the network of interneurons located in the larval antennal lobe (*Das et al., 2013*) is expected to participate in the processing of olfactory information arising from the OSNs (*Asahina et al., 2009*; *Larkin et al., 2010*). Although our work suggests that the computations achieved by the antennal lobe are not strictly necessary to guide robust chemotaxis (see also 'Materials and methods'), the function of the transformation carried out by the synapse between the *Or42a* OSN and its cognate projection neuron (PN) remains to be elucidated in the larva (*Ramaekers et al., 2005*; *Asahina et al., 2009*; *Masuda-Nakagawa et al., 2009*). As adult-fly PNs encode the second derivative of olfactory stimuli (*Kim et al., 2015*) including circuit elements downstream of the OSNs in the present multilevel model are expected to improve the accuracy of the behavioral predictions of the model.

The aim of this study was to clarify the relationships between the peripheral encoding of naturalistic olfactory stimuli and gradient ascent toward an odor source. By exploiting the sufficiency of a single OSN to direct larval chemotaxis (*Fishilevich et al., 2005*; *Louis et al., 2008*), we developed a mathematical model accounting for the transformation of time-varying stimuli into the firing rate of an OSN and the conversion of dynamical patterns of OSN activity into the selection between two basic types of action—running and turning. It will be interesting to examine the validity of the present model for the sensorimotor control of other aspects of larval chemotaxis such as turn orientation through lateral head casts (casting-to-turn transitions). In adult flies, turn orientation is determined by the crossing of the boundaries of odor plumes: upon encountering of an odor plume, flies veer upwind whereas exiting the plume initiates lateral and vertical casting (*van Breugel and Dickinson, 2014*)—an orientation strategy related to the surge-and-cast response of moths (*Carde and Willis, 2008*). To orient in

**Video 6.** Illustrative trajectory sequence in odor gradient with turn probability colored in red. Same trajectory as that shown in *Figure 8A*. Superimposition of the behavior on the dynamical reconstruction of the odor gradient based on the PDE simulations. The scale bar at the bottom left of the Video represents 5 mm. The green trace displays a 20 s segment of the past positions of the centroid. White spots represent the position of the head every 20 frames (or 0.66 s).

a rapidly changing olfactory landscape, the OSNs of various flying insects are capable of tracking rapid odor pulses on sub-second timescales and differentiating these signals (*Kim et al., 2011*; *Fujiwara et al., 2014*; *Szyszka et al., 2014*). Whether the processing of turbulent olfactory inputs involves more temporal integration than that described by the sensorimotor model proposed here remains to be elucidated. Finally, the *Drosophila* larva offers a unique opportunity to delineate the neural circuit basis of behavior (*Ohyama et al., 2013*, *2015*). Interdisciplinary approaches combining behavioral screens, functional imaging, and circuit reconstruction on the one hand (*Yao et al., 2012*), and computational modeling and robotics on the other hand (*Grasso et al., 2000*; *Webb, 2002*; *Izquierdo and Lockery, 2010*; *Ando et al., 2013*), should improve our understanding of how brains with reduced numerical complexity exploit streams of sensory information to direct action selection.

## Materials and methods

### Fly stocks

All behavioral experiments shown in the main figures were achieved with third instar larvae expressing the co-receptor Orco in only one OSN (*Fishilevich et al., 2005*) (*Or42a*-Gal4>UAS-*Orco*,UAS-ChR2-H134R; *Orco*$^{-/-}$) in a double blind background (*GMR-hid*/+;*dTrpA1*$^1$) (*Kwon et al., 2008*; *Xiang et al., 2010*). For the control experiments shown Figure 18, the double blind background was achieved with the null alleles *glass*$^{60j}$ and *dTrpA1*$^1$ (*Moses et al., 1989*; *Busto et al., 1999*). The UAS-ChR2-H134R transgene was donated by Stefan Pulver and Leslie C Griffith (*Pulver et al., 2009*). Flies were raised on standard fly food containing 0.5 mM all-*trans*-retinal in an incubator in complete darkness (food vials wrapped in aluminum foil). Exposure to ambient light was minimized until the experimental test. Approximately 96 hr after egg laying, third instar larvae were taken out of the food and immersed in a 15% (wt/V) glucose solution.

### Experimental arena with controlled odor gradient

A controlled odorant environment was created in a 120 × 120 × 12 mm arena consisting of a polystyrene dish (the lid of a Greiner square dish ref. number: 688102, Sigma–Aldrich, St. Louis, MO) standing on a 2% wt/V agarose surface inside the closed-loop tracker. A 3-µl odor droplet of IAA (0.25 M) was placed inside a plastic reinforcement ring at the center of the dish (internal diameter of disk occupied by the odor droplet: 5 mm). Inside the arena, an odor gradient emerged as a result of the diffusion from the source for 30 s prior to the introduction of a single larva. This step required a brief opening of the arena. The tracking was carried out for a minimum duration of 3 min. A minority of trajectories associated with no chemotactic response or with larvae idly dwelling under the odor source was excluded from the dataset.

### Behavioral quantification

#### Run/non-run classifiers

To detect turning events, we adopted a geometrical approach based on the physical trajectory described by the larva. First, we parsed the trajectory into segments of equal sizes. Next, we calculated the angle between successive segments of the trajectory. Finally, we computed the distribution over these angles. This distribution had the characteristics of a long-tailed exponential (data not shown). We defined a threshold at the location of the 'kink' of the distribution. Turning events were associated with positions with a turning angle larger than the threshold. We empirically found that good results were obtained for trajectory segments of 5 mm and an angular threshold of 20°. The results of the behavioral classification were insensitive to the precise length of the trajectory segments and the angular threshold.

Throughout the present study, the identification of turns was based on the trajectory of the midpoint (point located on the skeleton at a third of the distance from the head). Due to the high sampling rate of the tracker and the inherent noise of the stepper motors moving the stage holding the camera, the positions of the points of interest were subject to minute jittering. Unless stated otherwise, the trajectories of every point of interest were smoothened with a Savitzky-Golay filter to remove fluctuations on a small spatial scale irrelevant to the motion of the larva. This geometric approach was used to classify the data obtained from the closed-loop experiments (*Figures 1, 6–8*). For the open-loop experiments described in *Figure 5*, turns were identified by the online classification carried out by the tracker software (see section Tracker, Figure 16).

## Physical model for odor diffusion

For the experimental conditions used in previous work (*Louis et al., 2008*; *Asahina et al., 2009*; *Gomez-Marin et al., 2011*; *Gomez-Marin and Louis, 2014*), we obtained evidence that the odor gradients could be approximated as static. Due to the use of an odor source with reduced volume, this approximation did not hold in the present study. To correlate the behavior of the larva with a more accurate reconstruction of the odor gradient, we developed a physical model for the diffusion of the odor inside the behavioral arena (*Figure 9*). We used model-based estimation techniques for parameters underlying this physical model. We considered 3D diffusion with separate diffusion constants for air and the droplet. Exposed plastic surfaces of the chamber were treated as adsorptive boundaries. Since the odor gradient was initially established in the arena for 30 s prior to the introduction of a larva, our model also included non-zero initial concentration of the odor in the air, agarose, and plastic chamber. COMSOL Multiphysics v4.3 (COMSOL, Burlington, MA) was used to solve the diffusion equation with these boundary conditions. Parameter estimation was performed using the MATLAB/Optimization toolbox (MathWorks, Natick, MA) by solving a nonlinear least squares problem that matched the simulated odor concentration to measurements at the same time points.

The geometry of the experimental arena is described in *Figure 9*. The radius $r_{ring}$ of the odor ring confines the liquid droplet so that the radius of the flat face is equal to $r_{ring}$. The volume $V_{drop}$ of the droplet, made up of odor and solvent, is fixed to be 3 µl. According to the formula of a sphere, $V_{drop} = \frac{\pi}{6} h_{drop}(h_{drop}^2 + 3r_{ring}^2)$ where $h_{drop}$ is the droplet height. The agarose layer at the bottom of the chamber was modeled as a two-dimensional sheet with an independent diffusion constant.

## Boundary conditions

The top flat face of the droplet that contacts the plastic cap was treated as a no-flux boundary. Flux continuity was imposed on the spherical interface with air. The remaining boundaries, air-agarose, and air-plastic, were modeled as Robin boundary conditions to accommodate the possibility of adsorption-desorption reactions at the boundary. Although the standard way of treating adsorption reactions would be to use a reactive boundary condition where the odorant is treated as free in the air or bound to the boundary, we modeled these boundaries as a Robin boundary condition, which reduces the number of parameters to be considered (*Singer et al., 2008*). This simplification of the boundary conditions was necessary for us to estimate the associated parameters with a single experiment.

Considering a diffusing chemical species with concentration $x(t, \vec{r})$ that varies with time $t$ and location $\vec{r}$ within the chamber, the flux vector $\vec{J}$ of this chemical is given by $\vec{J} = -D\vec{\nabla}x$, where $D$ is the diffusion constant in air. If the normal direction to the boundary under consideration denoted as $n$, the Robin boundary condition relates to the normally incident flux to the boundary reaction by $-n.\vec{J} = k_i(x_{0,i} - x)$ where $k_i$ is related to the reaction rate at the boundary $i$ (agar or plastic), and $x_{0,i}$ is the saturation concentration of the odorant on this boundary. This reaction drives the flux toward the saturation concentration of the boundary. For example, if the concentration of the odorant in air is lower than the saturation concentration, the boundary would become an odor source by undergoing desorption, with a rate governed by $k_i$. Conversely, a higher concentration of the odorant in air would lead to adsorption at the boundary.

## Diffusion equation and coupling of the evaporation-diffusion process

There were two simultaneous diffusion processes, both of which were modeled using a PDE:

$$\frac{\partial x}{\partial t} = D_i \nabla^2 x.$$

The diffusion constant $D_i$ depended on whether the medium $i$ is air or droplet. We used a flux continuity condition at the droplet-air boundary, that is, the odorant could not accumulate at this boundary. As a result of the flux continuity at the droplet-air boundary, and because diffusion through air was substantially faster than through the droplet, the odor developed a radial profile within the droplet despite starting with a homogenous initial concentration. This process is our approximation of the coupled evaporation-diffusion process. Modeling the diffusion limited evaporation process (*Kelly-Zion et al., 2011*) would render this problem intractable in the context of our model-based parameter estimation. We, therefore, used an approximation for the evaporation process. Although inaccuracies could arise from this approximation because we ignored natural convection and concentration-dependent

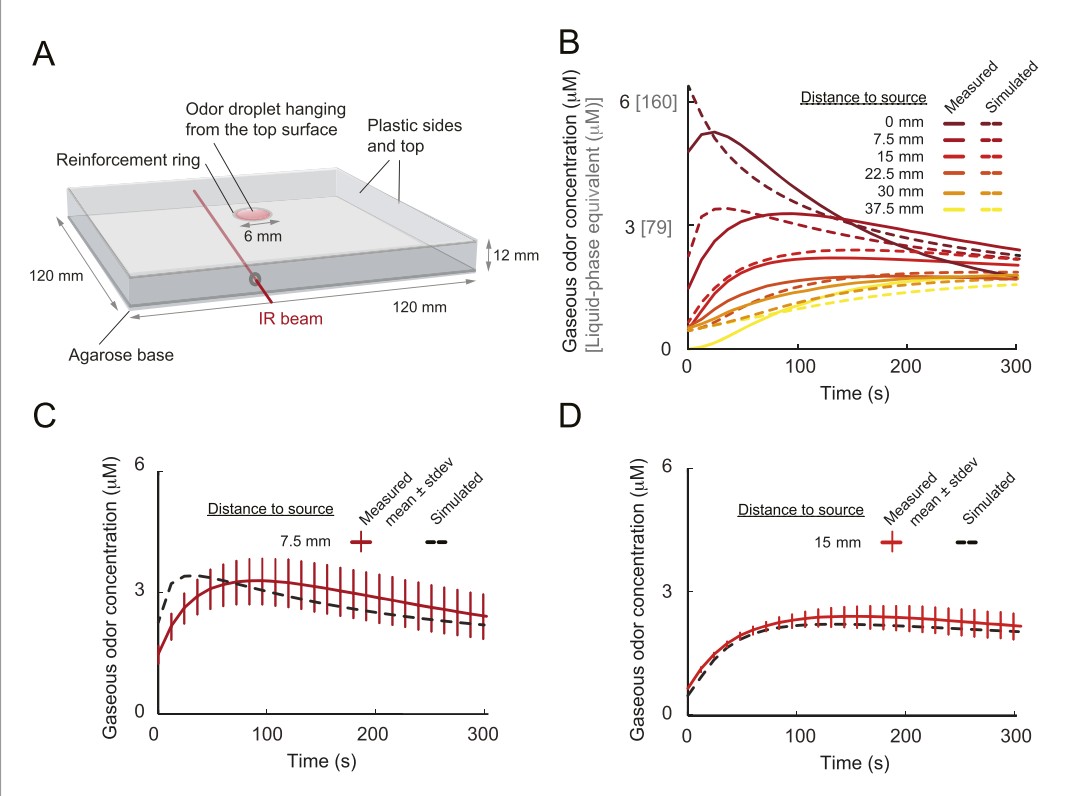

**Figure 9**. Physical model of odor diffusion in behavioral arena. (**A**) Configuration of behavioral arena on which the PDE model is based. The arena consists of a square shaped transparent plastic box with a side length of 120 mm and a height of 12 mm. The lid is inverted on a surface of agarose. The odor source consists of a solution of isoamyl acetate mixed with paraffin oil. A droplet of 3 µl of odor is placed inside a transparent reinforcement ring of a radius $r_{ring}$. This volume fills the ring evenly and, upon inversion of the lid on an agarose slab, the droplet remains suspended due to surface tension. The droplet shape is modeled as a spherical cap. The flat face of the droplet is in contact with the top plastic lid. The volume $v_{drop}$ of the droplet is related to the radius of the flat face $r_{ring}$ and the droplet height $h_{drop}$ according to the formula of a sphere. The agarose layer at the bottom of the chamber is modeled as a two-dimensional sheet with an independent diffusion constant. The top flat face of the droplet that contacts the plastic cap is treated as a no-flux boundary, and flux continuity is imposed on the spherical interface with air. The remaining boundaries, air-agarose and air-plastic, are modeled as Robin boundary conditions to accommodate the possibility of adsorption-desorption at these boundaries. The establishment of the odor gradient in the arena is modeled by two simultaneous diffusion processes, both of which are described by partial differential equation (PDE): $\partial x(\vec{r}, t)/\partial t = D \nabla^2 x(\vec{r}, t)$ where $x(\vec{r}, t)$ denotes the odor concentration at position $\vec{r}$ and time $t$. The diffusion constant D depends on whether the medium is air or the odor droplet. We used a flux continuity condition at the droplet-air boundary. For additional details about the model, see 'Materials and methods'. (**B**) As described in *Louis et al. (2008)*, infrared spectroscopy was used to estimate the absorbance and thereby the average concentration along sections of the arena (IR beam depicted in panel **A**). The time course of the cumulated concentration was determined for 7 sections at a distance from the source ranging from 0 to 45 mm (only first 6 are shown in the graph). Each concentration profile results from an average over 2 to 4 independent measurements. The absorbance was measured for a source concentration of 1.0 M. As discussed in 'Materials and methods', the parameters of the model are estimated by optimizing the fit of the model with the average concentration profiles along the 7 sections of the arena. The parameters of the model are reported in *Table 5*. The PDE model leads to a good fit of the temporal profiles of the average concentrations after an initial transient phase of 30 s. (**C**) Assessment of variability in the concentration estimates at a fixed position of the center of the 1.0 M odor source (7.5 mm). Mean concentration obtained from four independent infrared measurements. Error bars denote standard deviation. The time course of the simulated concentration is shown as a dashed line. (**D**) Assessment of variability in the concentration estimates at a fixed position of the center of the 1.0 M odor source (15 mm). Mean concentration obtained from three independent infrared measurements. Error bars denote standard deviation. The time course of the simulated concentration is shown as a dashed line.

changes in volatility of the odorant arising from chemical interactions with the solvent, our model was able to match experimental measurements with good fidelity as seen from the results of our model-based estimation (*Figure 9B–D*).

## Calibration of FT-IR measurements

Following the protocol described in (*Louis et al., 2008*), we used a gas-flow cell and Fourier Transform-Infrared Spectroscopy (FT-IR) (Bruker, USA) to assess the molar extinction coefficient of the odor in gaseous phase. The odor (isoamyl acetate, IAA) was mixed with the solvent n-hexane in different proportions. The solution was then injected directly into the gas flow cell by using a 10 µl Hamilton syringe (Hamilton Company, Reno, NV). The quantities of odor tested were 0, 0.01, 0.1, 0.2, 0.25, 0.5, and 1 µl. In all cases, the absorbance was measured by calculating the height of the absorption peak at the wave number 1765 cm$^{-1}$—a wavelength specific to IAA. Between trials, the gas flow cell was disassembled and all parts were rinsed with n-hexane. The absorption coefficient was estimated to be 479.87 M$^{-1}$cm$^{-1}$ at 25˚C. The concentration of saturated odor was estimated by injecting 10 µl of pure IAA into the gas-flow cell. The absorbance saturated at a value close to 1.36. Using the absorption coefficient, the concentration of the saturated vapor was estimated to be 278.45 µM.

## Model-based estimation of PDE parameters

Absorbance of infrared light was measured through the air along sections of the behavioral arena. As described in (*Louis et al., 2008*), the average odor concentration along the light path was estimated using the Beer–Lambert law. Based on this, we defined the parameters of the PDE model by characterizing the geometry of the gradient produced by a single odor source of 3 µl of IAA at a concentration of 1.0 M. A higher source concentration was used for the infrared spectroscopic measurements than for the behavioral experiments (0.25 M) to ensure the accuracy of the concentration estimates. We measured the absorbance along sections located at distances 0 mm, 7.5 mm, 15 mm, 22.5 mm, 30 mm, 37.5 mm, and 45 mm from the center of the droplet (*Figure 9B–D*). The measurements were taken after the odorant was placed in the chamber following the exact same protocol as for the behavioral experiments. From these, we inferred the average concentration at these 7 sections over a time interval of 360 s. For a given set of parameters based on an initial guess about the order of magnitude, we simulated the diffusion process. To this end, we used trapezoidal integration to estimate the average of the concentration along the sections that corresponded to the experimental measurements. The objective function to be minimized for estimating the parameters was the root-mean-squared error of the average concentration at each of the seven locations and at all times.

To minimize the objective function, we used sequential quadratic programming as implemented by the function *fmincon* in MATLAB v8.2. The Jacobian of the cost function with respect to the parameters being estimated was computed using finite differences. We constrained all physical parameters to be greater than 0. None of these inequality constraints were active for the converged solution. We tried 20 random initial guesses, and one where all parameters were set to 0. Once the optimization converged, we perturbed the converged estimates using random numbers. The perturbed estimates were fed back to the estimator, and the optimization ran again until convergence. We carried out 20 such restarts, and all of them converged to the same estimates, which are reported in *Table 5*. As described in *Figure 10*, the reconstruction of the odor gradient experienced by freely moving larvae (3-µl source of IAA at a concentration of 0.25 M) was then achieved by scaling down the gradient obtained at a source concentration of 1.0 M. We noted that the temporal evolution of the odor gradient is non-negligible (*Figure 10B,C*), which justified its integration in the entire analysis.

## Electrophysiology

Third instar larvae were transferred from the food vial into 15% (wt/V) glucose solution. Dissection of tissues was carried out in cold extracellular saline solution (*Singleton and Woodruff, 1994*) where the head was separated from the rest of the body while the brain was left intact. Using tissue glue (Histoacryl B, Braun, Germany), the dissected head was then glued in the middle of a glass slide at the bottom of a flow chamber. The cuticle covering the mouth hook was removed using a 3 mm Vanna spring scissor (Fine Science Tools, Germany) to make the dorsal organ ganglion accessible to the recording electrode. Throughout the experiment, the head was immersed in extracellular saline. The flow chamber was connected to two syringe pumps (Aladdin2-220, World Precision Instruments,

**Table 5.** Parameters of PDE model for odor diffusion

| Parameter | Physical description | Converged value |
|---|---|---|
| $D_{air}$ | Diffusion constant in air | $8.9377 \times 10^{-7}$ m$^2$s$^{-1}$ |
| $D_{drop}$ | Diffusion constant in droplet | $8.7859 \times 10^{-11}$ m$^2$s$^{-1}$ |
| $c_{0,air}$ | Initial odorant concentration in air | $4.0492 \times 10^{-7}$ mol l$^{-1}$ |
| $c_{0,drop}$ | Initial odorant concentration in droplet | $0.0450$ mol l$^{-1}$ |
| $k_{agar}$ | Robin rate for air-agar boundary | $1.5762 \times 10^{-6}$ ms$^{-1}$ |
| $k_{plastic}$ | Robin rate for air-plastic boundary | $5.8025 \times 10^{-5}$ ms$^{-1}$ |
| $c_{0,agar}$ | Saturation concentration of agar | $3.6817 \times 10^{-5}$ mol l$^{-1}$ |
| $c_{0,plastic}$ | Saturation concentration of plastic | $5.7921 \times 10^{-7}$ mol l$^{-1}$ |

The parameters are optimized for an odor gradient of isoamyl acetate under the experimental condition outlined in *Figure 9*.

Sarasota, FL) to perfuse the preparation with fresh saline and to ensure the continuous evacuation of the odor out of the chamber. The chamber volume was approximately 500 µl. The flow in the chamber was 28.4 µl/s, leading to a turnover of the chamber volume in 17.6 s.

Recording electrodes were pulled (P97, Sutter Instruments) out of borosilicate glass capillaries (1.5 mm/1.12 mm outer/inner diameters (OD/ID), World Precision Instruments, Novato, CA) with a 10 µm open tip. Electrodes were then back-filled with 3 µl of extracellular saline. A chlorinated silver wire (0.38 mm in diameter) was used to connect the electrode to the head stage of a microelectrode amplifier (Axon MultiClamp 700B, Molecular Devices, Sunnyvale, CA). The electrode was mounted on an automated micromanipulator (ROE-200 & MPC-200, Sutter Instruments). The antennal nerve in close vicinity of the dorsal ganglion was sucked into the recording pipette by applying a negative pressure (−20 kpa) created through vacuum. The extracellular signal was amplified 100 times at the microelectrode amplifier; it was visualized on an oscilloscope (Tektronix, Beaverton, OR), and recorded at a sample rate of 20 kHz by a personal computer (PC) equipped with the free data acquisition software SpikeHound (*Lott et al., 2009*). The conception of this preparation and recording technique benefited from pioneering recordings from the larval olfactory organ (*Oppliger et al., 2000*; *Kreher et al., 2005*; *Hoare et al., 2008*).

## Light stimulation at electrophysiology rig
For the light stimulation, a blue light emitting diode (LED) (LCS-0470-03-22 LED, Mightex, Canada) was mounted in a lighthouse (U-DULHA, Olympus, Japan) and integrated into the light path of the microscope (BX51, Olympus) allowing for localized stimulation of the larval head through a 40× immersion objective. The light intensity arriving at the larval head was estimated by measuring the photocurrent under the objective with a photodiode (SM05PD7A, Thorlabs, Newton, NJ) connected to a bench top photodiode amplifier (PDA200C, Thorlabs). The LED was controlled by a custom Labview (National Instruments, Austin, TX) interface available from the following link: https://github.com/LabLouis/eLife_2015/tree/master/Electrophysiology. The current controlling the LED was fed into the data acquisition software, where it was recorded along with the signal from the suction electrode. Pulses of blue light elicited spikes exclusively in OSNs expressing ChR2 (*Figure 2B,C* and *Figure 11A*).

## Spike-sorting algorithm
Spike-sorting and PSTH analysis was performed with Matlab using custom scripts available from the following link: https://github.com/pahammad/OpSIN. In the *Orco* null background, spontaneous activity of OSNs is drastically reduced but not abolished (*Hoare et al., 2008*). Furthermore, non-OSNs contribute to the activity monitored from the DO ganglion. We devised an optogenetic spike-sorting strategy to distinguish spikes of the *Or42a* OSN from other spikes (*Figure 11B,C*) (*Lima et al., 2009*). Channelrhodopsin-2 (ChR2-H134R) was expressed in the single functional *Or42a* OSN (*Or42a>*ChR2). The spike-sorting algorithm (called OpSIN) parsed the extracellular recording data and collected candidate ChR2-evoked spikes specifically during light-activation time windows based on amplitude

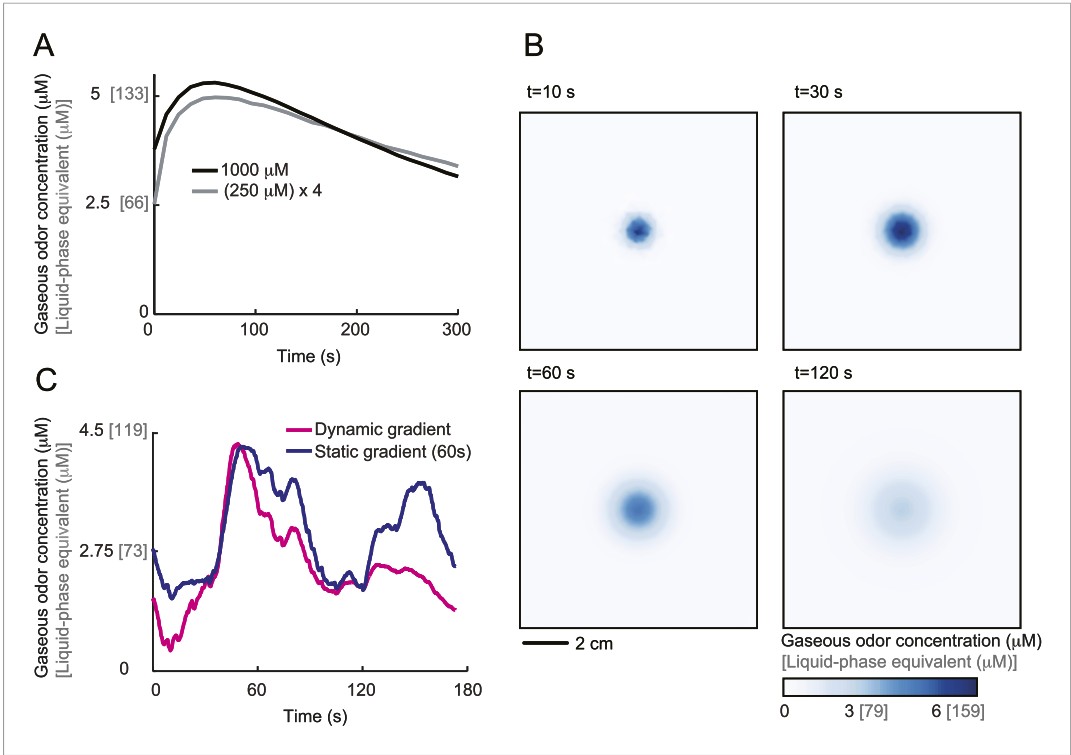

**Figure 10**. Dynamical reconstruction of the odor gradient experienced by the larva during free behavior. (**A**) Behavioral experiments were conducted with a 3-μl odor droplet at a concentration of 0.25 M. At this low concentration and due to the small volume of the source, accurate temporal profiles of the average odor concentration could not be obtained through infrared spectroscopy for all sections of the plate. Under the source, we find that the average concentration of the odor scales as a function of the source concentration. We therefore assume that the gradient obtained for a 0.25 M source can be approximated by the gradient reconstructed at 1.0 M scaled by a factor 0.25. (**B**) Numerical integration of the PDE model (parameter set listed in **Table 5**) permits us to reconstruct the temporal evolution of the odor gradient experienced by the larva at arbitrary spatiotemporal precision. Due to practical considerations, the reconstruction was saved at time steps of 1 s. As on the time scale of 1 s the geometry of the gradient does not evolve considerably, a linear interpolation is applied between defined sections of the gradient. Over time, the gradient tends to flatten out. This effect is due to the gradual depletion of the source. It is worth noting that in previous studies (**Louis et al., 2008**; **Asahina et al., 2009**; **Gomez-Marin et al., 2011**), the gradients generated by a single odor source could be approximated as roughly constant. In these configurations, the enhanced stability of the gradient was due to the larger volume of the odor source (10 μl) and the reduced dimension of the behavioral arenas. (**C**) Reconstruction of the concentration time course experienced during the trajectory displayed in **Figure 1A**. The magenta trace was obtained after mapping the behavior onto the dynamical reconstruction of the gradient shown in panel **B**. In contrast, the blue trace was obtained after mapping the behavior onto a static gradient computed 60 s after the onset of the odor diffusion. The differences between the two temporal profiles highlight the importance of the dynamical reconstruction for the experimental conditions in the present study.

thresholding and local non-maximum suppression. Template candidates were separated into clusters using an affinity propagation algorithm (**Frey and Dueck, 2007**). Spike selection was accomplished by comparing candidate waveforms identified throughout the recording to ChR2 derived waveform templates. Candidate waveforms were then transformed to appear as similar as possible to the template waveform via dynamic time warping (**Berndt and Clifford, 1994**). The probability of spike occurrence at every candidate location was estimated by warping the residual distance between the candidate waveform $X$ and the chosen set of spike templates $T$.

$$d(T, X) = \sqrt{(X - DTW(T \rightarrow X))^2}.$$

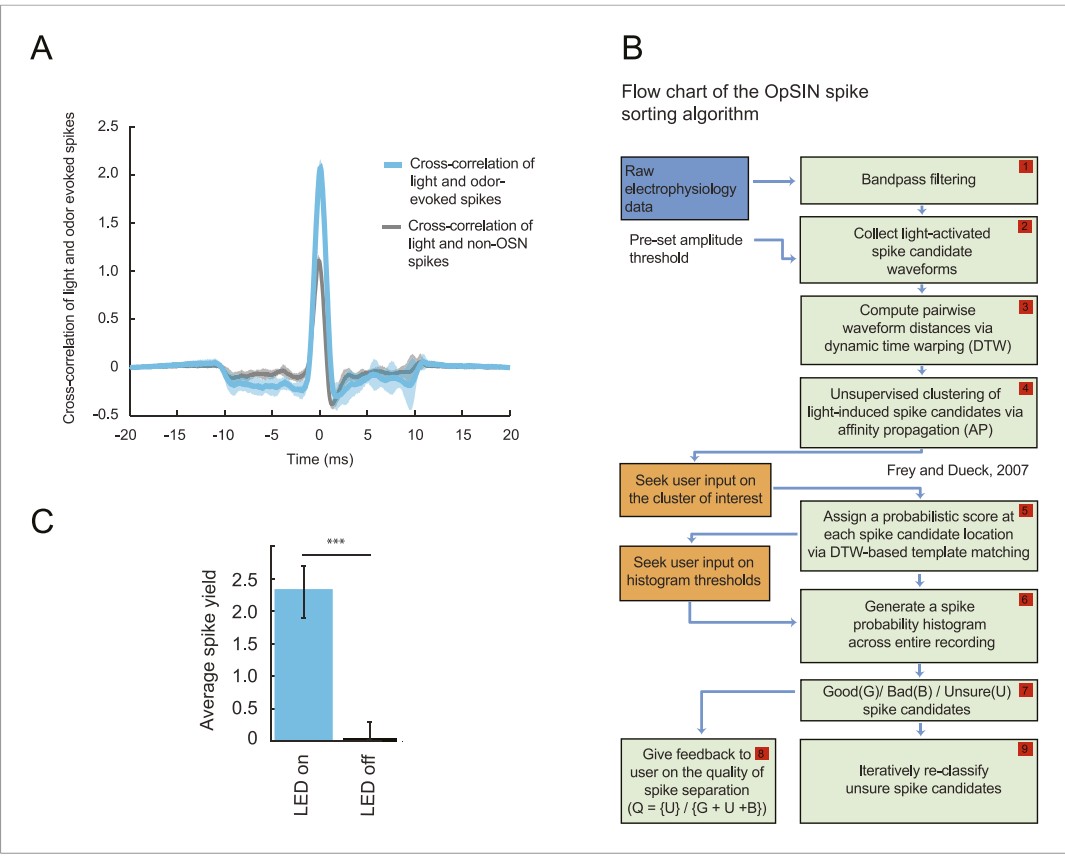

**Figure 11**. Semi-automated channelrhodopsin assisted spike sorting (OpSIN). (**A**) Cross-correlation of light- and odor-evoked spikes (blue) recorded from the *Or42a*>ChR2 OSN, and cross-correlation of light-evoked and spontaneous background spikes (gray). The similarity in the shape of the waveform is larger between the light- and odor-evoked spikes compared to light-evoked and background spikes. Light-evoked spikes were collected from time windows during a light stimulation interval; odor-evoked spikes were collected during the odor stimulation interval; background spikes were collected from intervals devoid of light and odor stimulations. Although light- and odor-evoked spike waveforms are very similar to each other during the same recording, the overall shape of the spike waveforms can vary across recordings. (**B**) Flow chart of the functions underlying the spike-sorting algorithm (OpSIN). Spike candidates were selected during light-activation episodes based on simple amplitude thresholding and local non-maximum suppression. Spike selection was accomplished by comparing candidate waveforms identified throughout the recording to the ChR2-derived waveform template by transforming candidate waveforms to appear as similar as possible to the template via dynamic time warping. Orange boxes denote steps where inputs are sought from the user. For more details, see 'Materials and methods'. (**C**) Average spike yield of the *Or42a*>ChR2 OSN measured in a 58-ms time window with and without light stimulation. The data were obtained by analyzing 35 recordings. The spike yield during the light stimulation is 2.30 ± 0.40 spikes. The corresponding spike latency is 6.96 ± 1.02 ms (t-test, p < 0.001).

A spike-probability-histogram-based cut-off was applied across the entire recording to select the correct spikes and assign identities.

$$p(T, X_i) = e^{\left(\frac{d(T,X_i)}{K}\right)},$$

where *K* is the median of all the pair wise distances computed across the set of spike-candidates.

$$p(T) = \max(p(T, X_i)),$$

where the spike candidate's probability is based on its best matched template. Every PSTH was mildly low-pass filtered using a Savitzky-Golay filter.

## Liquid phase odor stimulation

For the odor stimulation two-barrel pipettes (1.5 mm/0.84 mm OD/ID, World Precision Instruments) were pulled using a PMP-107 Multipipette Puller (MicroData Instrument, South Plainfield, NJ) resulting in a two-barrel tip with a 5 µm OD. One barrel was back-filled with the odor solution (IAA dissolved in extracellular saline), and the other with extracellular saline alone. An injection needle (0.51 mm/0.26 mm OD/ID, Becton, Dickinson and Company, Franklin Lakes, NJ) was inserted into the back of each barrel and airtight sealed using hot-melt adhesive. Each barrel was subsequently connected to separate channels of a pressure-driven flow controller (Fluigent, France). The tip of the odor stimulation pipette was placed and maintained at a distance of 10 µm in front of the larval dorsal organ with a micromanipulator (ROE-200 & MPC-200, Sutter Instruments). The output pressure and thereby the flow of each individual channel of the odor delivery pipette was controlled at a temporal resolution of 10 Hz via a custom Labview interface available from the following link: https://github.com/LabLouis/eLife_2015/tree/master/Electrophysiology. Rapid concentration changes of the odorous stream were achieved by varying the flow between the empty channel and the odor channel while keeping the overall flow constant at a set value of 0.32 nl/s (measured flow rate corresponding to a 320 mbar output pressure). The experimental setup is outlined in *Figure 12A*.

The liquid phase odor stimulation system was calibrated using a dilution series of the fluorescent dye fluorescein, which was expelled at a constant pressure (320 mbar) into the perfusion chamber (Bioscience Tools, San Diego, CA) while imaging the tip of the pipette with a high-speed camera (Andor, United Kingdom). We determined that the relationship between the fluorescein concentration and the measured fluorescence was best fitted by a linear function (*Figure 12E*, left panel). When varying the output pressure while keeping the concentration of fluorescein constant, the observed fluorescence could be mapped to the output pressure using a linear function (*Figure 12E*, right panel). The concentration arriving at the dorsal organ was quantified by comparing the fluorescence measured at the tip of the odor delivery pipette (region of interest $ROI_{tip}$) to the fluorescence measured right at the dorsal organ ($ROI_{dorsal}$). Assuming that the fluorescence measured at the tip of the odor delivery pipette represents the undiluted concentration, the dilution factor ($d$) was calculated according to:

$$d = \frac{\Delta F(ROI_{tip})}{\Delta F(ROI_{dorsal})}.$$

The time lag between the pressure output and the odor arriving at the dorsal organ was quantified by computing the cross correlation between the recorded output pressure of the mass-flow controller system (MFCS) and the measured fluorescence at the larval dorsal organ. By calculating the shift of the peak of the cross correlation with respect to the peak of the autocorrelation of the pressure output signal, the final lag for each experiment was determined. The lag was estimated to be 500 ms.

## Phase conversion for olfactory stimulation

Gas-phase odor stimulation was achieved by means of a custom olfactometer delivering a continuous stream of air (510 ml/min). The air stream, regulated by two mass flow controllers (Cole–Parmer, Vernon Hills, IL), was humidified and subsequently passed through the odor solution and delivered to the tip of the larval head. Rapid concentration changes of the odorized stream were achieved by varying the flow between the empty channel and the odor channel while keeping the overall airflow constant. A custom Labview protocol was used to control the flow rate of the individual mass flow controllers at a rate of 30 Hz. This script is available from the following link: https://github.com/LabLouis/eLife_2015/tree/master/Electrophysiology. The experimental setup is outlined in *Figure 12B*: it is similar in design to past olfactometers (*Borst and Heisenberg, 1982*; *Kim et al., 2011*) and it was adapted to produce a stimulus time course on the timescale of a typical run.

Odor concentrations were estimated in gaseous phase by using a mini photoionization detector (200B miniPID, Aurora Scientific, Canada). Calibration of the miniPID (photoionization detector) was achieved by passing an air stream through pure IAA to obtain saturated vapor. Since the PID signal saturated at high concentrations of airborne odor, it was necessary to establish the calibration curve in an odor range that was significantly lower than saturated vapor. The saturated airflow was therefore diluted into clear air in ratios of 1:10.0, 1:19.2, 1:38.5, and 1:83.0. Gaseous concentrations and PID voltage readings followed a linear relationship (*Figure 12F*), which allowed us to convert the PID readout into absolute concentration.

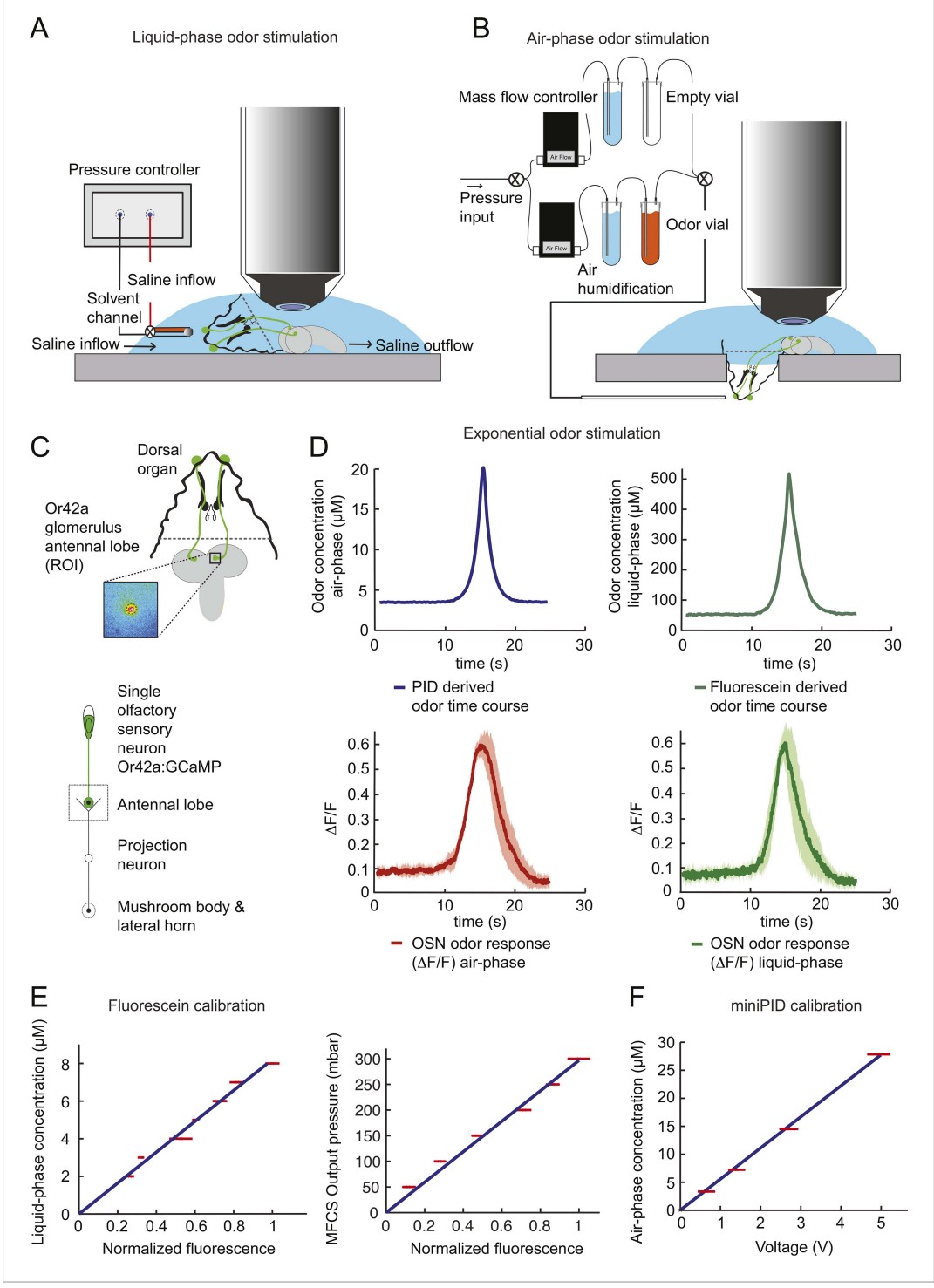

**Figure 12**. Phase conversion of air and liquid phase odor stimulation. (**A**) Schematic illustration of the single-OSN imaging setup: liquid phase odor stimulation of the larval dorsal organ with a pressure controller connected to a two-barrel pipette. The odor solution is placed in one channel of a two-barrel pipette while the other channel contains pure saline. The fixed larval head is perfused by a constant flow of extracellular saline while the odorous solution is delivered via the glass pipette. As for the olfactometer in gaseous phase (panel **B**), rapid changes in concentration of the odorous stream are achieved by varying the flow of the empty channel and the odorous channel while keeping the overall flow constant. As described in 'Materials and methods', the flow rates of individual channels are controlled by a system of mass flow controllers. (**B**) Air-phase odor stimulation of the larval dorsal organ

*Figure 12. continued on next page*

*Figure 12. Continued*

with a custom-built olfactometer. A continuous stream of air (510 ml/min) is regulated by combining the outputs of two mass flow controllers. The air stream is humidified, and subsequently passed through the odor solution after which the odorous air stream is delivered to the larval dorsal organ. Rapid concentration changes of the odorous stream are achieved by varying the flow between the empty channel and the odor channel while keeping the overall flow constant. As described in the 'Materials and methods', individual flow rates are controlled in real-time. The OSN activity was recorded by imaging the GCaMP activity elicited in the axon terminal of the *Or42a* OSN. (**C**) Schematic illustration of the site of imaging of single OSN glomeruli at the level of the antennal lobe (inset: false-color-coded activity in the axon terminal of the *Or42a* OSN). (**D**) Odor response profile (ΔF/F) of the *Or42a* OSN in response to an exponential odor stimulation in the air-phase (left) and liquid-phase (right). The response amplitude and overall dynamics are conserved between the liquid and gaseous phases. Shades denote standard deviation. Recordings were made on a total of six experiments conducted on three different preparations. (**E**) Calibration of fluorescein in liquid phase showing a linear relationship of the fluorescein concentration and the measured change in fluorescence (left panel). Linear relationship between the observed change in fluorescence and the output pressure of the odorous stream (right panel). (**F**) Calibration of the photoionization detector (PID) with airborne odorant stimuli showing a linear relationship between the odor concentration and the measured voltage change. Error bars denote standard deviation.

As illustrated in *Figure 12C*, the OSN activity was monitored by means of calcium imaging at the axon terminal of the OSNs at the larval antennal lobe (*Or42a*>GCaMP3) using a high-speed camera (Andor, United Kingdom) and a fluorescence microscope (BX51 mounted with 40× water immersion objective, Olympus). The OSN activity was measured during odor stimulation in gaseous and liquid phase, respectively. The phase correlation between liquid and gaseous stimulations was established for an 8-s exponential ramp. In liquid phase, both background and maximum concentration were fixed to the pre-existing experimental conditions used for the electrophysiology: they corresponded to a range spanning between 50 µM and 530 µM. The corresponding ΔF/F of the OSN response ranged between 10% and 60% in liquid phase. The background and maximum concentration of the airborne ramp were adjusted to obtain an activity profile with a ΔF/F matching the liquid phase stimulations.

Using the miniPID, the estimate of the final air-phase exponential odor time course ranged from 2.20 to 20.13 µM and led to an OSN response ranging between 10% and 59%. From the original ΔF/F, it was found that the ratio of the max ΔF/F in gaseous and liquid phases was 0.98. The ratio of the integral of ΔF/F over the full time course of the stimulation was 0.87. We established the equivalence of the OSN calcium responses elicited by an exponential ramp with a concentration range of 50–530 µM in liquid phase and with a concentration range of 2.20–20.13 µM in gaseous phase, with comparable OSN response dynamics for both phases (*Figure 12D*). To map the concentration range of 2.20–20.13 µM in gaseous phase onto the concentration range of 50–530 µM in liquid phase, we used the ratio between the maximum concentrations of the ramp in liquid and gaseous phases: $\rho^{\text{liquid} \to \text{gas}} = 26.73$. This conversion was applied to predict the behavior of larvae in an airborne odor gradient on the basis of the model for the OSN activity developed for liquid phase stimulation (*Figure 4*).

$$y(t) = F_{\text{liquid}}\left(x_{\text{gas}}(t) \times \rho^{\text{liquid} \to \text{gas}}\right),$$

where $y$ denotes the activity of the OSN, $x_{gas}$ the concentration in gaseous phase, and $F_{liquid}$ the predicted neural activity from the IFF+IFB (ODE) model introduced in *Figure 4B*.

## Reverse-correlation analysis of OSN response dynamics

Standard systems identification approaches have shown that the important aspects of the response of invertebrate photoreceptors can be approximated as linear (*Marmarelis and McCann, 1977*), even though the modeling of nonlinear features of the response requires a more sophisticated treatment (*French et al., 1993*). More recently, it has been suggested that the response dynamics of OSNs in *C. elegans* and primary thermosensory neurons in the *Drosophila* larva are largely linear (*Kato et al., 2012*; *Klein et al., 2015*). We therefore examined whether a linear-nonlinear model could be used to describe the OSN response of the larva. Following a reverse-correlation approach, we stimulated the

*Or42a*>ChR2 OSN by a M-sequence induced with light (*Figure 13A*). Reproducible patterns of neural activity were observed (*Figure 13C*), from which a biphasic filter was computed (*Figure 13B*). This filter had a shape similar to those found in retinal ganglion cells (*Chichilnisky, 2001*) and insect OSNs (*Geffen et al., 2009*; *Martelli et al., 2013*). To test the predictive power of this filter, we reconstructed the activity elicited by an exponential and a sigmoid ramp (*Figure 13D,E*). Whereas the linear filter led to a reasonable reconstruction of the firing pattern elicited by the M-sequence (*Figure 13C*), it produced unsatisfactory results for the graded ramps with a mismatch so pronounced that it could not be rectified by a nonlinear function. This conclusion is consistent with the nonlinear response dynamics observed in adult-fly OSNs upon stimulation by graded odor ramps (*Kim et al., 2011*).

## IFF and IFB motifs

The dynamics of the pure IFF motif (*Figure 4Bi*) is described by a 3-variable ODE system (*Figure 4Bii*). We hypothesized that the firing rate of the OSN (*y*) results from the combined effects of direct excitation and indirect inhibition of the OSN activity. The excitation is mediated by the gating of the OR by the binding of odorant molecules or the absorption of photons by channelrhodopsin-2 (ChR2). By analogy to the olfactory transduction cascade in the moth (*Gu et al., 2009*), we speculated that the indirect inhibition is mediated by an intermediate variable (*u*) that might represent the concentration of calcium bound to calmodulin. For the pure IFF motif, the dynamics of variable *u* results from a production term proportional to the stimulus *x* and a first-order decay term.

To model the direct excitation and indirect inhibition of the OSN activity, we used a control function (d(*x*,*u*) where d stands for depolarization) inspired by the *cis*-regulatory logic of gene transcription (*Goentoro and Kirschner, 2009*):

$$d(x, u) = \beta_1 \frac{x}{\beta_2 + x + \beta_3 u}.$$

This expression was built from thermodynamic considerations about the state of a promoter occupied by transcription factors (*Ackers et al., 1982*; *Bintu et al., 2005*). Here, we hypothesized that a similar function is suitable to describe the depolarizing effects of the opening of the OR (or ChR2), and the indirect hyperpolarizing effects that calcium bound to calmodulin might have on the OSN membrane. The contribution of each trend is described by *x* and $\beta_3 u$, respectively.

In addition, we assumed that the intermediate variable (*u*) and the OSN spiking activity (*y*) undergo a first-order decay. For the OSN activity, the introduction of such a decay can be justified by speculating about the existence of ion pumps that restore the membrane to resting potential after an initial increase of cations following the gating of the OR (or ChR2) (*Gu et al., 2009*). By trial and error, we also discovered the necessity of including a constitutive decay (offset) term that vanishes at a low firing rate. Although the molecular correlate of this offset remains undefined, it could be explained by the homeostatic function of ion pumps. The combined effects of the two decays are mathematically described as:

$$h(y) = -\beta_4 \frac{y^2}{y^2 + \theta^2} - \beta_5 y,$$

where *h* stands for hyperpolarization. To keep the model as simple as possible, the membrane potential was not modeled explicitly. Instead, we assumed that depolarizing d(*x*,*u*) and hyperpolarizing h(*x*) effects on the OSN membrane can be translated into excitatory and inhibitory effects on the OSN firing rate. While our knowledge about the olfactory transduction cascade in *Drosophila* was insufficient to justify these assumptions, the goodness of fit resulting from the integration of the ODE model demonstrated that the OSN dynamics could be captured by the combination of d(*x*,*u*) and h(*x*). By combining the previous relationships, we obtained the following systems of ODEs:

$$\frac{du}{dt} = \alpha_1 x - \alpha_2 u,$$

$$\frac{dy}{dt} = \beta_1 \frac{x}{\beta_2 + x + \beta_3 u} - \beta_4 \frac{y^2}{y^2 + \theta^2} - \beta_5 y. \tag{2}$$

The second regulatory motif we considered is a negative IFB. This motif has been implicated in the process of olfactory transduction and adaptation in adult flies (*Nagel and Wilson, 2011*). It also forms the

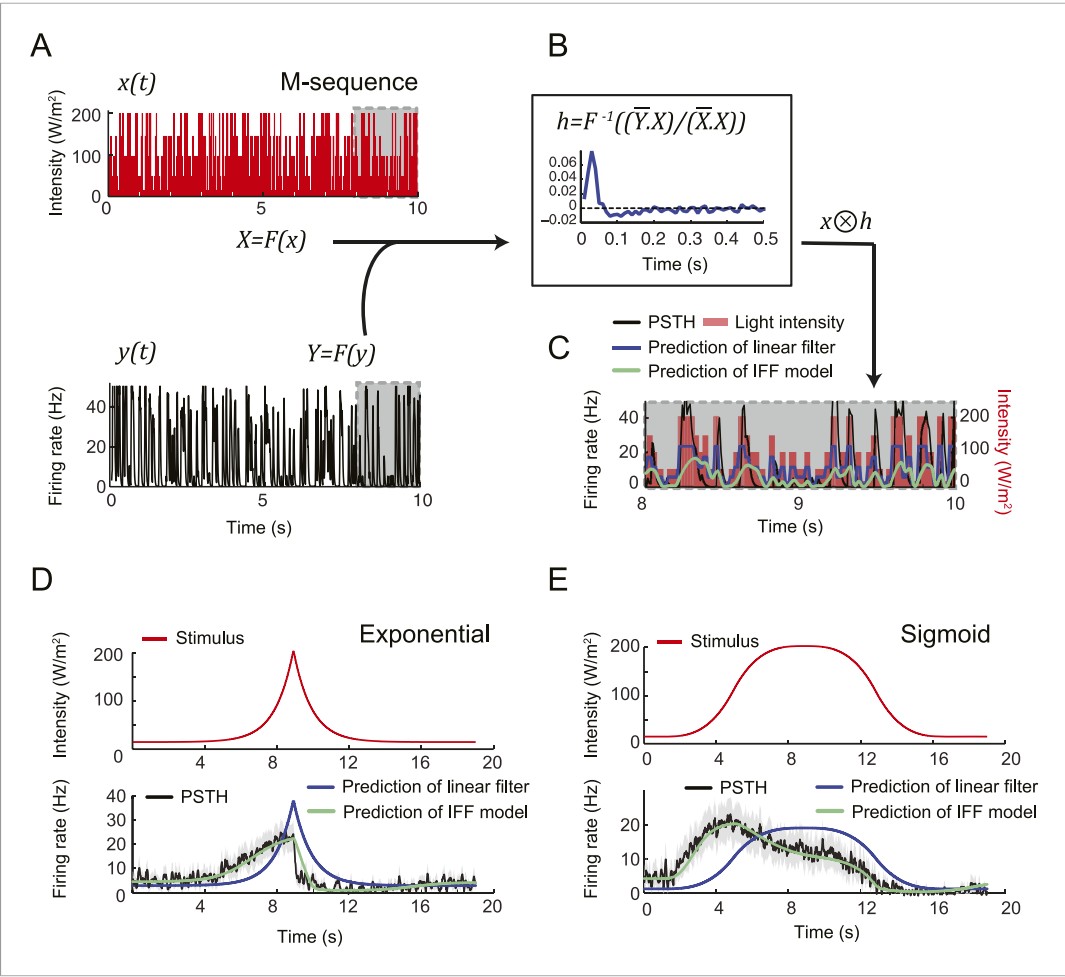

**Figure 13**. A linear filter alone is insufficient to account for the transfer function of the *Or42a>*ChR2 OSN.
(**A**) Stimulation of the *Or42a>*ChR2 OSN by a maximum-length (M) sequence generated with light. The M-sequence was based on a discretization of the light intensity range 15–207 W/m² into the following 5 values: 15, 50, 100, 150, and 207 W/m². The M-sequence featured all possible 4-element combinations of these 5 intensities. In the experiments, changes in light intensity occurred with time steps of 33 ms. (Bottom, left) PSTH of the neural response computed over 10 trials (10 preparations) and for a bin size of 10 ms. The gray boxes outline a 2-s time window over which the predictions of the linear filter are reported in panel **C**. (**B**) Computation of the linear filter, *h*, through the operations described in panel **B** (*Chichilnisky, 2001*; *Nagel and Wilson, 2011*). Function F represents the Fourier transform from the time domain to the frequency domain; $F^{-1}$ represents the inverse transformation from the frequency domain to the time domain. The bar above the variables in frequency space denotes the complex conjugate transformation. To cancel any DC drifts in the OSN response, the filter was computed on windows of 5-s slid over the entire duration of stimulus (20 s). An average filter was computed from this series (dark blue line). The linear filter was used to make predictions about particular stimulus time courses. (**C**) Neural activity predicted from the linear filter in response to the M-sequence. The prediction was obtained by convolving the filter with the time course of the stimulus. The resulting activity was normalized to have the same mean as the experimental activity. The result of the prediction is shown for a 2-s window of the complete stimulus (dashed gray box in panel **A**). The activity predicted from the linear filter (dark blue line) is compared to the output of the IFF (ODE) model introduced in *Figure 4B* (green line). Pearson's correlation coefficient ($\rho$) of the predicted and the experimental activities is 0.37 for the linear filter reconstruction compared to 0.57 for the IFF model. (**D**) Application of the linear filter derived in panel **C** for the M-sequence to stimulation by an exponential ramp (red line). The linear filter (dark blue line) fails to predict the PSTH that was observed experimentally (black line; gray error bars represent the standard deviation) ($\rho = 0.56$). Prediction of the IFF model shown in green ($\rho = 0.93$). (**E**) Application of the linear filter derived in panel **C** to a sigmoid ramp (red line). As in panel **D**, the linear filter (dark blue line) fails to reproduce the experimental PSTH ($\rho = 0.54$) while the IFF model leads to a good fit (green line, $\rho = 0.98$).

regulatory basis of the transduction pathway underlying adaptive chemoreception in bacterial chemotaxis (*Yi et al., 2000*; *Tu et al., 2008*). For this motif, we assumed that the activity of the neuron had an excitatory effect on the intermediate variable $u$, which in turn has an inhibitory effect on the OSN activity. In a first approximation, the negative feedback was assumed to be linear. The difference between the IFF and IFB motifs lies in the production of the intermediate variable ($u$), which in the case of the IFB is proportional to the firing rate ($y$) and not the stimulus intensity ($x$). These considerations yielded the following system of ODEs:

$$\frac{du}{dt} = \alpha_3 y - \alpha_2 u,$$

$$\frac{dy}{dt} = \beta_1 \frac{x}{\beta_2 + x + \beta_3 u} - \beta_4 \frac{y^2}{y^2 + \theta^2} - \beta_5 y. \tag{3}$$

Using numerical simulations, we found that the IFB motif alone cannot account for the dynamics of the OSN activity. In contrast, we discovered that the combination of the IFF and IFB motifs leads to substantial improvements in the quality of the fit (*Figure 4*). For all simulations achieved in this work, the ODE equations were numerically integrated by the solver *ode23s* built in Matlab.

## Parameter fitting of the composite IFF+IFB and pure IFF models

To optimize the parameters of the model to the experimental data, a standard fitting procedure was applied. As outlined in *Figure 4Bi*, we considered three possible models: the motif IFF, the motif IFB, and a combination of the two motifs. The joint probability of the observations was maximized as a function of the internal parameters for the neural activity patterns elicited by a set of 6 linear ramps, 5 nonlinear ramps, (*Figure 4—figure supplement 1*) and a naturalistic stimulus (*Figure 2D,E*). For each stimulation protocol, the confidence interval of the OSN activity (PSTH) was used to achieve a robust fit of the free parameters of the model. This procedure was applied to each of the three models independently.

In the ODE systems presented in *Figure 4Bii* and *Equations 2, 3*, we observe that the time derivative of $u$ can be multiplied by an arbitrary scaling factor reabsorbed by the fitting parameter $\beta_3$ in the time derivative of $y$. For this reason one is forced to fix one of the eight parameters to a constant value. For numerical convenience we chose to fix $\alpha_1 = 0.1$. To infer the actual value of this parameter, one would need to experimentally access the value of the intermediate variable $u$, whose molecular identity remains unknown. For the pure IFF model, the number of free parameters is therefore seven. In addition to these parameters, we considered the scaling of the firing rate $y$ elicited by individual stimulation protocols via a multiplicative factor accounting for variability across experimental conditions (e.g., minute differences in the positions of the stimulation pipette).

The maximization of the likelihood function was achieved by means of the Nelder-Mead (NM) method (*Nelder and Mead, 1965*), which proved to be fast and reliable. The result of the NM optimization was then refined through a gradient search algorithm (*Brun and Rademakers, 1997*). For the dataset corresponding to the light stimulation, the fitting procedure led us to rule out the relevance of the IFB model alone with a probability of $\chi^2$ very close to zero. In contrast, the IFF was able to reproduce the experimental observation with good accuracy. On the other hand, in the case of the odor stimulation, we obtained a significant improvement of the model fit by adding an IFB component to the IFF motif (addition of the term $\alpha_3 y$ to the dynamics of $u$).

The fitted value of the composite IFF+IFB motif indicated that the IFB component was not negligible during the stimulation and accounted for about 30% of the final firing rate (*Figure 4D*). In contrast to the pure IFF model, variables $u$ and $y$ of the IFF+IFB model were entangled in the structure of the ODE resulting in a coupling that allowed us to fit the value of parameter $\alpha_1$. We also examined the effect of introducing additional terms in the denominator of the function defining $y$, such as the product $u \times y$. Besides the test of other circuit motifs, the introduction of additional free parameters represented a qualitative test against the possibility of over-fitting. The improvements in the fitting obtained in these cases were very marginal.

With regards to both light and odor stimulation protocols, the data comprised stimulation patterns on diverse timescales and with varying stimulus durations: 10 linear and nonlinear ramps lasting less than 25 s and one 'naturalistic' stimulus lasting more than 200 s. We found that the parameter set leading to a good fit during the first 30 s of the light or odor stimulation did not yield an accurate fit for longer durations. By fitting the activity at the beginning and the end of the naturalistic stimulation, we discovered that the discrepancy between both time ranges was mainly due to a change in the

threshold $\theta$ of the Hill term in the time derivative of $y$ (**Equation 2**). We therefore allowed the threshold $\theta$ to change smoothly between the two different time ranges with the functional expression: $\theta' = \theta \times (\tau/t)^2$ for $t > \tau$ with $\tau = 30$ s. A third set of measurements of the firing rate at steady state (time interval 20–24 s in **Figure 3—figure supplement 1**) was used as an independent control of the parameter fit obtained from the fitting of the other stimulation protocols.

## Derivation of the mathematical solution of the IFF motif

As observed in **Figure 4E,F**, the pure IFF motif not only accounts for the response of the OSN stimulated by light, but it also represents a good approximation of the OSN dynamics stimulated by an odor (**Figure 4E,F** and **Table 2**). The general solution of **Equation 2** is:

$$u(t) = \alpha_1 e^{-\alpha_2 t} \int_o^t e^{\alpha_2 t'} x(t') dt' + C^{ste} e^{-\alpha_2 t}.$$

For times $t$ larger than $\alpha_2$, the second term of the solution converges to zero, and we obtain the more compact form:

$$u(t) = \alpha_1 \int_o^t e^{-\alpha_2(t-t')} x(t') dt'. \tag{4}$$

By sequentially integrating (4) by parts, we obtain the following identities:

$$u(t) = \alpha_1 \langle x(t) \rangle_{\alpha_2} = \frac{\alpha_1}{\alpha_2} x(t) - \frac{\alpha_1}{\alpha_2} \langle \frac{dx}{dt}(t) \rangle_{\alpha_2}. \tag{5}$$

where the brackets $< >$ denote the convolution introduced in relationship (4). While timescale of the dynamics of variable $u$ is given by $\alpha_2$, the time scale of the stimulus can be approximated as

$$\tau_x \simeq \frac{x_{max} - x_{min}}{\left(\frac{dx}{dt}\right)_{max}}.$$

For the linear and nonlinear ramps tested in **Figure 3**, $\tau_x$ is typically 10 s. As the value of $\alpha_2$ is 0.88 s$^{-1}$ (**Table 1**), the variable $u$ evolves on a timescale approximately 10 times faster than the stimulus. Using relationship (4), **Equation 2** can be rewritten as:

$$\frac{dy}{dt} = \beta_1 \frac{x}{\beta_2 + x + \beta_3 \alpha_1 \int e^{-\alpha_2(t-t')} x(t') dt'} - \beta_4 \frac{y^2}{y^2 + \theta^2} - \beta_5 y. \tag{6}$$

The function multiplying parameter $\beta_4$ is a steep sigmoid (or Hill function) whose value is close to 1 when $y$ is reasonably larger than 0. More formally, this approximation is valid for values of $y$ larger than the threshold $\tilde{y}$:

$$\frac{\tilde{y}^2}{\tilde{y}^2 + \theta^2} = (1 - \varepsilon) \rightarrow \tilde{y} = \sqrt{\frac{(1-\varepsilon)}{\varepsilon}}\theta. \tag{7}$$

Given that the value of $\theta$ of 0.3, we see that the Hill term will be larger than 0.95 for values of $y$ larger than 1.3 Hz. For this range of values, **Equation 6** can be rewritten as:

$$\frac{dy}{dt} = f(x(t), t) - \beta_5 y, \tag{8}$$

where $f(t)$ is a function independent of $y$. This function evolves on a timescale slower than $\alpha_2$. If we assume that the firing rate $y(0)$ is initially 0, the solution of (8) is:

$$y(t) = \int_o^t e^{-\beta_5(t-t')} f(x(t'), t') dt'. \tag{9}$$

**Equation 9** shows that the dynamics of $y(t)$ obeys a characteristic time given by $\beta_5$. Since $\beta_5 = 13.03$ s$^{-1}$ (**Table 1**), variable $y$ evolves on a timescale more than 10 times faster than the stimulus. In view of this separation of the timescales, it is justified to assume that $y$ is at quasi-steady-state (QSSA, $dy/dt \approx 0$) during the evolution of the stimulus $x$ and variable $u$. By combining this assumption with (6), we find that:

$$y^{QSSA}(t) = \frac{\beta_1}{\beta_5}\left(\frac{x(t)}{\beta_2 + x(t) + \alpha_1\beta_3\int e^{-\alpha_2(t-t')}x(t')dt'}\right) - \frac{\beta_4}{\beta_5}, \tag{10}$$

Based on the values of the parameters of the original ODE system (*Table 1*), we obtain $\beta_1/\beta_5 = 132.9$ Hz, $\beta_2 = 1.27$ W/m², $\alpha_1\beta_3 = 0.25$ s⁻¹, and $\beta_4/\beta_5 = 93.17$ Hz. It is interesting to note that the convolution of $x$ is necessarily smaller when $x$ takes on larger values. Using the identities (5), we can rewrite (10) as:

$$y^{QSSA}(t) = \delta_1\left(\frac{x(t)}{\delta_2 + x(t) - \delta_3\int e^{-\alpha_2(t-t')}\frac{dx}{dt'}(t')dt'}\right) - \delta_4, \tag{11}$$

where

$$\delta_1 = \frac{\beta_1}{\beta_5}\frac{\alpha_2}{(\alpha_2 + \alpha_1\beta_3)} = 103.66 \; Hz.$$

$$\delta_2 = \beta_2\frac{\alpha_2}{(\alpha_2 + \alpha_1\beta_3)} = 0.99 \; W/m^2.$$

$$\delta_3 = \frac{\alpha_1\beta_3}{(\alpha_2 + \alpha_1\beta_3)} = 0.22.$$

$$\delta_4 = \frac{\beta_4}{\beta_5} = 93.17 \; Hz.$$

As expected, the QSSA solution is in excellent agreement with the results of the integration of the full ODE system (*Figure 14*). Given the values of the parameters (*Table 1*), the denominator of relationship (11) is mostly driven by the stimulus intensity for slowly evolving stimuli. The contribution of the convolution over the first derivative is significant for rapid and large changes of the stimulus intensity. In the limit where the stimulus is constant over time ($dx/dt = 0$), the scaling term $S(x,t)$ is equal to zero and the QSSA predicts the dose–response function displayed in *Figure 3—figure supplement 1C*:

$$y^{QSSA} \propto \frac{x}{x + \delta_2} - C_1 \qquad dx/dt = 0,$$

where $C_1$ is a constant. For stimulation patterns in which the intensity changes at a constant rate (linear ramps, *Figure 4E* and *Figure 4—figure supplement 1A*), the first derivative is constant with a positive or a negative sign.

$$y^{QSSA} \propto \frac{x}{x + \delta_2 - C_2} - C_1 \qquad \text{constant } dx/dt,$$

with $C_1$ and $C_2$ being constants. The sign of $C_2$ is determined by that of $dx/dt$. For constant stimulus gradients (linear increases in concentration), we predict that the firing rate also followed a hyperbolic function (dose–response) of the stimulus intensity (*Figure 3—figure supplement 1*). This time, however, the dose–response is expected to saturate at lower values of the stimulus intensity when the gradient is positive (rising phase of the linear ramps) and higher values of the stimulus intensity when the gradient is negative (falling phase of the linear ramps). These predictions are consistent with the firing patterns observed in *Figure 4*.

## Tracker

The tracker outlined in *Figure 5* is presented in more detail in *Figure 15*. It was conceived and built at the Instrumentation Design and Fabrication Facility at the Janelia Research Campus. Part of the construction was carried out by KeyTech (Baltimore, USA). Unlike other tracking systems (*Faumont et al., 2011*; *Leifer et al., 2011*; *Kocabas et al., 2012*), our setup was designed to keep the stage on which the larva evolves fixed by mounting the camera and stimulation LEDs on a moving stage whose position was continuously updated to stay locked with the animal's position (*Figure 15*). The upper and lower moving stage were powered by a pair of stepper motors (T-LSR450B, Zaber Technologies, Canada). The blueprint of the tracker and list of parts are available from the following

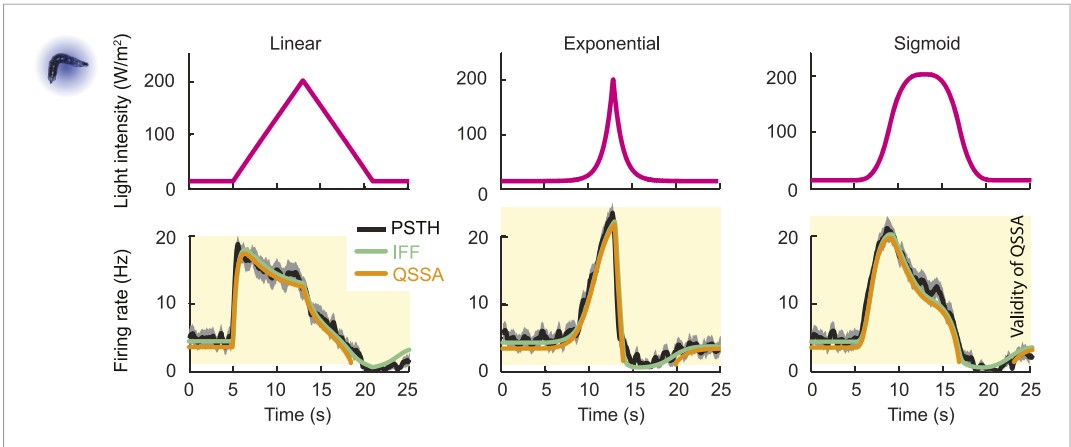

**Figure 14.** Quasi-steady state approximation of the IFF model describing the spiking dynamics of a single OSN stimulated by light ramps. Comparisons of the analytical solution of the OSN activity upon numerical integration of the full ODE system (green line) with the solution obtained under the quasi-steady state approximation (QSSA, orange line) and the experimental PSTH (black line, shades denote standard deviation). The goodness of fit of the QSSA is remarkable for the linear and nonlinear ramps. As discussed in the 'Materials and methods', the QSSA holds for values of y such that the Hill term can be linearized ($y > 1.3$ Hz). This domain of validity is depicted by the orange background.

link: https://github.com/LabLouis/eLife_2015/tree/master/Tracker%20Hardware. The light stimulation module consisted of three LEDs, connected in series (LCS-0470-03-22, Mightex Systems) to an LED controller (SLA-1200-2, Mightex Systems) whose output current limit was set to 750 mA. The angle and position of each LED was fixed to cover the camera's field of view with maximum light intensity. The controller's output current scaled proportionally to the analog voltage fed into the controller board. The light intensity reaching the arena was estimated by measuring the current emitted by a photodiode (SM05PD7A, Thorlabs) connected to a benchtop amplifier (PDA200C, Thorlabs). The tracker's video camera (A622f, Basler, Germany) was placed at the center of the three blue LEDs; it delivered images at a resolution of 800 × 800 pixels at a frame time interval of 23 ms. Combined with the time required to process the image and actuate the position of the stage, the effective frame rate was 30 Hz.

As summarized in *Figure 15C*, the larval tracker control unit (LTCU) formed the main hardware interface controlling the LEDs and acquiring images from the video camera. While the camera was controlled via a transistor–transistor logic pulse signal, a 12-bit digital-to-analog converter output was used to send the control signals (0–5 V analog signal) to the LED controller. An in-circuit debugging (ICD) Port was used to connect the LTCU to an ICD3 programmable interface controller via a registered jack (RJ11) connector interface, while an USB Port enabled the LTCU's communication to the PC. A customized program written in C was used to direct the function of the LTCU. This C program enabled the LTCU to respond to the commands issued by the PC. The software interface of the tracking and image analysis software interfacing the LTCU was written in JAVA. Both the C program and JAVA interface are available from the following links: https://github.com/LabLouis/eLife_2015/tree/master/Wormsign (tracker) and https://github.com/LabLouis/eLife_2015/tree/master/Venkman (JAVA controller interface).

## Update of light intensity

For closed-loop experiments described in *Figures 6, 7*, the light intensity was updated based on the position of the head of the larva mapped on a predefined light landscape. The light landscape was loaded and interpolated by the software controller environment from a matrix with a spatial resolution of 1 × 1 mm. Since the position of the larva was monitored at a higher resolution, the intensity of the spatial landscape was redefined by using a bi-linear interpolation along the x- and y-axis. The position of the center of the light gradient was automatically adjusted at the beginning of the experiment in such a way that every larva started in a direction facing the center of the gradient at a fixed distance of

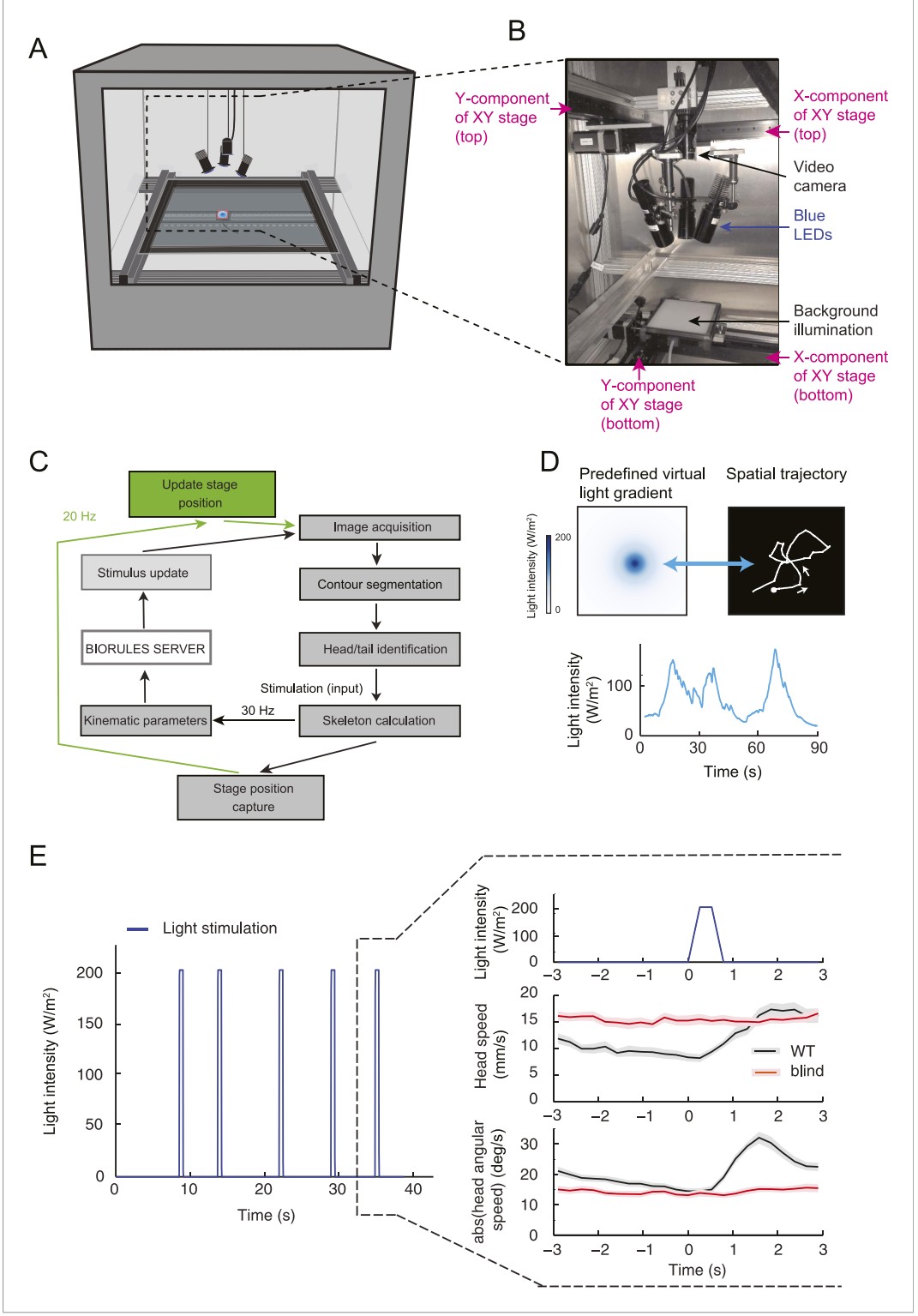

**Figure 15**. Technical description of the closed-loop tracker for virtual olfactory realities. (**A**) Schematic drawing of the closed-loop tracker. The blue LEDs and the camera are mounted on a moving stage that follows the larva while it crawls on an agarose slab (40 × 40 cm or approximately 120 × 120 body lengths of the larva). (**B**) Depiction and description of the moving camera stage equipped with three LEDs. (**C**) Flow chart outlining the interaction of the core modules of the tracking software ('Materials and methods'). (**D**) Illustration of the spatial trajectory generated by an *Or42a*>ChR2 larva undergoing closed-loop light stimulation in a virtual odor gradient. (Top-left) Predefined light
*Figure 15. continued on next page*

*Figure 15. Continued*

landscape with a geometry approximating the odor distribution produced by a point source. During the behavioral tests, the full gradient is not projected onto the arena: the larva is illuminated by the LEDs at an intensity determined by its position in the virtual light gradient. (Bottom) The light intensity is updated based on the motion of the larva, which forms the temporal evolution shown in the graph. (Top-right) The spatial trajectory described by an *Or42a>*ChR2 larva in the virtual light gradient. The orientation response faithfully reproduces chemotactic behavior. In closed-loop experiments, the LED intensity was updated according to the position of the head with respect to a predefined spatial landscape. In open-loop experiments, the LED intensity was determined by a predefined temporal profile implemented only when a larva was in a run mode (*Figure 5B*). During a run, the motion of the larva had no influence on the intensity of the stimulus. As soon as the larva interrupted a run, the light intensity returned to a baseline value (15 W/m²). (**E**) Abolishment of photophobic behavior in blind larvae. Stimulation of larvae with light flashes of 0.5 s (intensity: 207 W/m²). The flashes were interspaced by a time interval picked from a Poisson distribution with mean 7.7 s. To ensure that a larva had sufficient time to react to individual light pulses, the minimum inter-flash interval was set to 5 s. The behavior resulting from the flash was characterized by computing the flash-triggered averages of the amplitude of the absolute head angle and its time derivative. Wild-type larvae (black trace) display an increase in head motion following the flash (release of head cast). Blind *GMR-hid;dTrpA1¹* larvae (red trace) were not affected by the light flashes. The graph represents the means of the kinematic variables computed across trajectories; error bars denote SEM.

9.6 mm from the center for the exponential light gradient (*Figure 6*) and a distance of 17.1 mm for the family of light landscapes (*Figure 7*). The minority of trajectories from which larvae failed to detect the presence of the light gradient was discarded from the dataset.

For open-loop experiments described in *Figure 5*, the LED intensity was updated based on a predefined temporal function. This function was only implemented when the larva was classified in a run mode according to the rules described in *Figure 16*. Throughout the study, odor and light gradient landscapes were represented using brewermap (S. Cobeldick, MathWorks file exchange) with color schemes from http://colorbrewer2.org/.

## Computation and predictions of the turn probability

In the behavioral experiments of *Figure 5*, the probability of turning ('turn rate') was estimated from the relative number of turns that took place during a 1-s time window centered on the time point of interest. If we denote the number of runs observed at the beginning and the end of the ith time window as $Nb_i$ and $Ne_i$, respectively, $Nb1$ represents the total number of runs contained in the dataset. Over time, $Nb_i$ decreases monotonically to 0. The turn probability associated with the ith time point was estimated as the fraction of runs that ended during the corresponding time window: $(Nb_i - Ne_i)/Nb_i$. A sliding time window was then applied to estimate the turn probability corresponding to every time point of the experiment. The turn probability was thus defined as the likelihood of implementing a turn within a 1 s time window, resulting in values smaller than 1 (in the case of 1 all runs entering a given time window switch to a turn) and larger or equal to 0 (in the case of 0 no turn takes place during a given time window). While a short time window led to noisy turn probability estimates, long time windows led to undesirable averaging effects. We empirically found that a time window of 1 s (or 30 time points) offered a good tradeoff.

### Generalized linear model for turn predictions

As described in *Figure 5*, the predicted probability of turning was computed from a GLM based on a linear function of the firing rate of the OSN, *y*. To map the domain of definition of the firing rate ([0, ∞]) onto the domain of definition of the turn probability ([0, 1]), we applied a logit link function (*Myers et al., 2002*). This led to the following relationship:

$$\ln\left(\frac{\lambda(t)}{1-\lambda(t)}\right) = \gamma_0 + \gamma_1 y(t). \tag{13}$$

The two parameters of the model, $\gamma_0$ and $\gamma_1$, were trained on a set of 10 light ramps listed in *Figure 5—figure supplement 1*. Parameter $\gamma_0$ corresponds to a term related to the basal turn rate. The values of the parameters of the model are reported in *Table 3*.

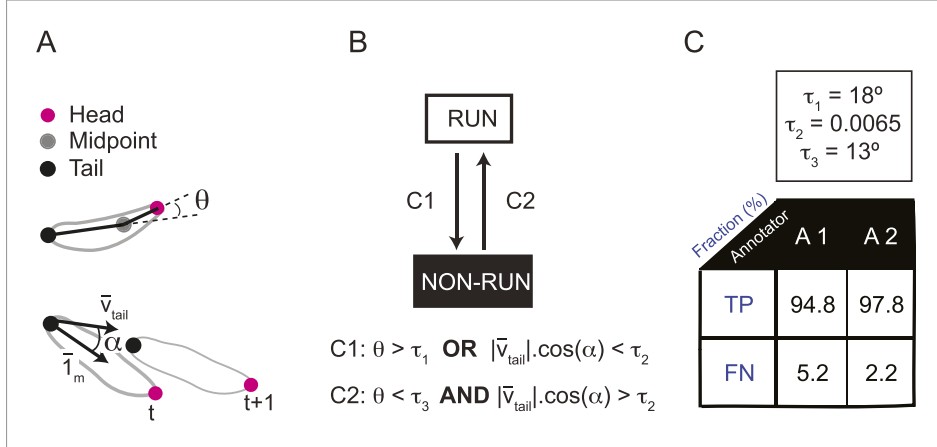

**Figure 16**. Real-time classification of the locomotor behavior of the larva. (**A**) In the present study, the locomotor behavior of the larva is classified into two basic types of action: forward runs and non-runs. Non-run is a meta-state that includes stops, turns, casts, and backward runs. No distinction is made between non-run behaviors. To distinguish runs from non-runs, we defined empirical filters based on the following three sensorimotor features: (1) the head angle between the direction of the body axis (tail-midpoint) and the neck axis (head-midpoint); (2) the instantaneous velocity vector measured at the tail position (vector vtail) and (3) the angle α formed by tail velocity vector and the direction of the body axis (unitary vector 1$_m$). (**B**) Transitions from a run to a non-run state take place as soon as (i) the head angle is larger than a threshold $\tau_1$ or (ii) the dot product of the tail velocity vector and unitary vector corresponding to the direction of the body axis is smaller than $\tau_2$ (condition C1). The first condition on the head angle identifies head casts while the second condition on the dot product identifies sequences of behavior associated with a stop or a backward run. Transitions from a non-run to a run take place when the following two conditions are verified: (iii) the head angle must be lower than the threshold value $\tau_3$ and (iv) the dot product of the tail velocity and unitary vector along the body axis must be larger than $\tau_2$ (condition C2). (**C**) The thresholds of conditions (i–iv) were set at a value that maximizes the difference in the cumulative distribution of the run and non-run states along the relevant sensorimotor feature (head angle or dot product) for the two possible types of behavioral state transitions (run → non-run or non-run → run). Distributions of the frame-by-frame sensorimotor features were constructed by pooling the annotated frames from four representative trajectories. Manual annotation was achieved by two trained experimenters. (Top) The values of the thresholds used in the analysis were: $\tau_1 = 18°$, $\tau_2 = 0.0065$, and $\tau_3 = 13°$. (Bottom) Comparison of frame-by-frame manual classification of run/non-run behavior achieved by the two trained annotators (A1 and A2) vs the computational classification obtained by filters described in panel **B**. The table reports the percentage of true positives (frames classified as a run by both the annotator and the algorithm) and false negatives (frames classified as a run by the annotator and a non-run by the algorithm). This good match validates the use of the computational classifiers in real-time experiments.

As shown in *Figure 17*, the logit transform included in the GLM improved the accuracy of the fit without being essential. The predictive value of the model trained on open-loop behavior was then evaluated when applied to closed-loop experiments such as those described in *Figures 6–8*. The performance of the test model was compared to a control model devoid of sensory processing of the OSN. For the control model, relationship (13) became:

$$\ln\left(\frac{\lambda(t)}{1-\lambda(t)}\right) = \gamma'_0 + \gamma'_1 x(t), \tag{14}$$

where the parameters $\gamma'_0$ and $\gamma'_1$ were computed from a linear regression on the open-loop experiments listed in *Figure 5—figure supplement 1* (parameter values listed in *Table 3*).

Finally, the contribution of the first derivative of the stimulus intensity was assessed in the enhanced control model:

$$\ln\left(\frac{\lambda(t)}{1-\lambda(t)}\right) = \gamma''_0 + \gamma''_1 x(t) + \gamma''_2 \frac{dx}{dt}(t). \tag{15}$$

The time derivative of the stimulus was defined by the change in stimulus intensity that occurred during the frame (33 ms) preceding the present time point *t*. The goodness of fit of the two control models can be compared to the test model in *Figure 5—figure supplement 3*.

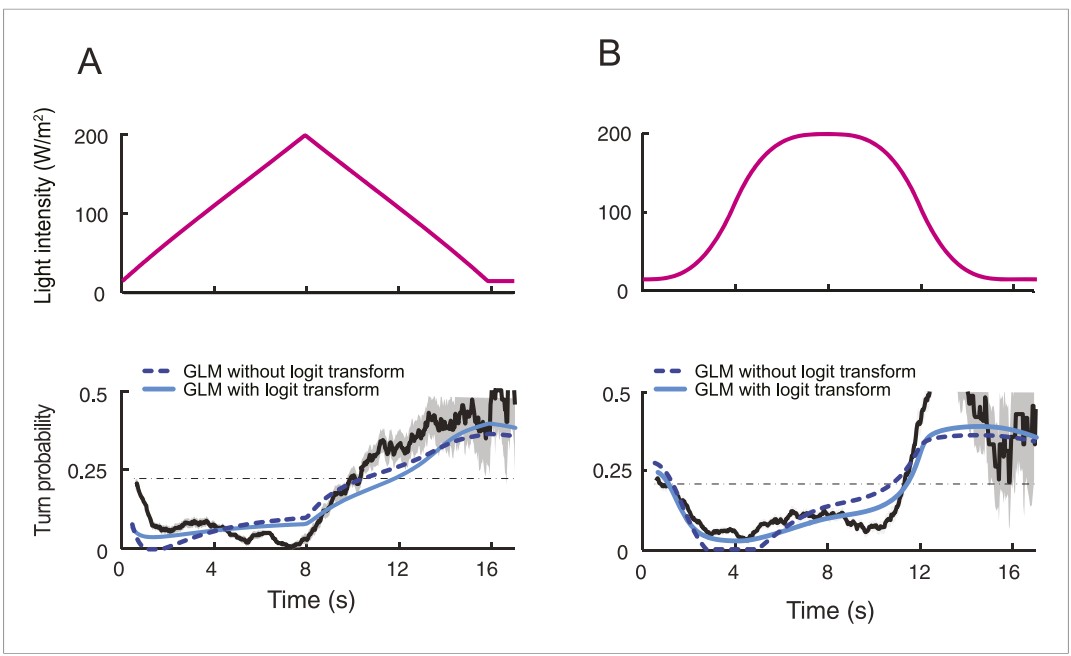

**Figure 17**. Contribution of the logit transform to the predictions of the integrated stimulus-to-behavior generalized linear model (GLM). (**A**) Comparison of the behavioral outputs of the GLM with and without logit transformation for the run-to-turn transitions elicited by a 8-s linear light ramp. In the absence of logit transformation, the GLM is based on a linear combination $\lambda(t) = \gamma_0 + \gamma_1 y(t)$. Negative values of the linear combination were rectified to be equal to 0. The parameters of the model were trained on the full set of ramps presented in *Figure 5—figure supplement 1* (derived parameter set: $\gamma_0 = 0.3762$, $\gamma_1 = -0.0198$ Hz$^{-1}$). The result of the GLM without logit transformation is represented by a dashed line. The GLM model that includes the logit transformation is represented by a plain line (parameter set listed in *Table 3*). The correlation coefficients ($\rho$) and coefficients of variation of the RMSE computed on the outputs of the GLM with and without logit transformation differ by 12% and 10%, respectively. (**B**) Same as panel **A** for a sigmoid light ramp. The correlation coefficients and coefficients of variation of the RMSE computed on the outputs of the GLM with and without logit transformation differ by 5% and 22%, respectively.

## Statistical procedures and data analysis
This section summarizes the methods used for the quantification of physiological and behavioral data.

### Quantification of goodness of fit of the OSN and behavioral models (material pertaining to *Tables 2, 4*, *Figures 13, 18*)
For the firing rate of the OSN and the probability of turning, two metrics were used to quantify the goodness of fit of the models. First, we used Pearson's correlation coefficient. If we denote the experimental observations as $X_t$ and the output from the model as $Y_t$, the correlation coefficient was computed as:

$$r = \frac{\sum_{t=1}^{n}(X_t - \bar{X})(Y_t - \bar{Y})}{s_X s_Y},$$

where $s_X$ is the standard deviation of variable $X$ and $n$ represents the total number of time points in the dataset. The second metric used is the coefficient of variation (CV) of the root-mean-square error (RMSE) defined as:

$$\mathrm{CV(RMSE)} = \frac{\sqrt{\frac{1}{n}\sum_{t=1}^{n}(X_t - Y_t)^2}}{\bar{X}}.$$

### Error bars for the PSTHs (material pertaining to *Figures 2–8*)
Except for *Figure 13*, all PSTH were computed on a sliding time window of 50 ms (*Dayan and Abbott, 2001*). Throughout the manuscript, the error bars (shaded areas) associated with the firing rate of the *Or42a>ChR2* OSN (PSTH) denote the standard deviation.

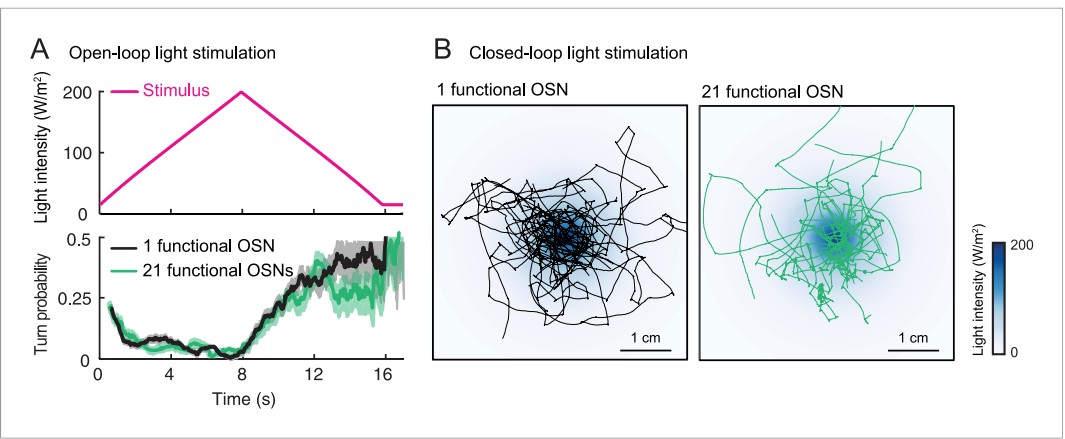

**Figure 18**. Behavior of single functional Or42a>ChR2 OSN in the background of a silenced and an intact peripheral olfactory system (1 vs 21 functional OSNs). (**A**) Open-loop stimulation protocol with a 8-s linear light ramp (magenta line). Comparison of the behavior elicited by ChR2 in the *Or42a* OSN in the background of 20 silenced OSNs (1 functional OSN, black line) and 21 functional OSNs (green line). The genotype of larvae with only one functional OSN is: *w;Or42a*-Gal4,*GMR-hid*;*Orco²,dTrpA1¹* × *w*;UAS-*Orco*,UAS-ChR2-H134R;*Orco²,dTrpA1¹*. This genotype is used throughout the study. The genotype of control larvae with 21 functional OSNs is: UAS-ChR2-H134R/*Or42a*-Gal4;*gl⁶⁰ʲ,dTrpA1¹*. This genotype is only used in the present control. The modulation of the turn probability elicited by the stimulus is similar for both genotypes. Plain lines indicate the mean of the turn probability; shaded areas denote standard deviation (see 'Materials and methods'). (**B**) Closed-loop behavior in an exponential light gradient. The stimulus landscape is the same as that used in *Figure 6A*. Side-by-side comparison of 10 light-driven trajectories superimposed onto the gradient. The overall patterns of the trajectories indicate that light-driven chemotaxis can be mediated by a single OSN in the background of 21 functional OSNs.

## Error bars for the turn probabilities estimated from open-loop behavioral experiments (material pertaining to *Figures 5, 17, 18*)
Error bars for the turn probability were estimated by resampling the initial distribution of runs 100 times without replacement and with a sample containing half of the original set of runs. The error bars reported in the figures (shaded areas) correspond to the standard deviations calculated on the resampled sets of runs.

## Cross-validation of the ODE model for the OSN spiking activity (material pertaining to *Figure 4—figure supplement 2*)
The composite IFF+IFB model developed to reproduce and predict the activity elicited by odor stimulation was trained on the full set of odor ramps listed in *Figure 4—figure supplement 1* together with the natural stimulus shown in *Figure 2D,E*. The parameters derived through this training procedure (*Table 1*) were used throughout the study. To validate the structure of the ODE model, we followed a cross-validation approach in which the IFF+IFB model was trained on the 4-s linear ramps listed in *Figure 4—figure supplement 1Ai–ii* and used to accurately predict the activity elicited by the 8-s ramps listed in *Figure 4—figure supplement 2B* (middle panel) as well as the naturalistic stimulus shown in *Figure 4—figure supplement 2C* (top panel). The same approach was followed to provide a successful validation of the IFF model to describe the spiking activity elicited by light stimulations (*Figure 4—figure supplement 2*, bottom panels).

## Controls for behavioral experiments: open-loop light stimulation (material pertaining to *Figure 5*)
In open-loop stimulation experiments (*Figure 5*), the output of the integrated stimulus-to-behavior GLM described in (13) was computed for each time point of the experiment. The performances of the test model were compared to a control GLM in which the turn probability was predicted from the stimulus intensity without any sensory processing from the OSN. For this control model, we independently fitted a GLM identical to that presented in *Figure 5D* after replacement of the firing rate by the stimulus intensity alone or a combination of the stimulus intensity and its first derivative (*Figure 5—figure supplement 3*). The goodness of fit of the test and control GLMs are reported in *Table 4*.

## Controls for behavioral experiments: closed-loop light and odor stimulation (material pertaining to *Figures 6, 8*)

In closed-loop experiments conducted in light and odor gradients (*Figures 6, 8*), the predictions of the integrated stimulus-to-behavior GLM described in *equation (13)* were analyzed by computing the turn-triggered average of the turn probability. This average time course was compared to two controls. The first control consisted in achieving turn predictions based on a sensory experience whose relation to the corresponding behavior was uncoupled. To this end, we inverted the time course of the light (odor) intensity and associated it with the forward time course of the behavior. This first control preserved the statistics of the stimulus intensity experienced by the larva. The second control was based on the assumption that the OSN firing rate was constant over time and corresponded to the mean activity predicted by a typical trajectory. Given that each larva experienced a different time course of light intensity, the mean OSN firing rate was computed on a trajectory-by-trajectory basis. Note that the control GLM derived in *equation (14)* for the open-loop light ramps was deemed inadequate to predict closed-loop behaviors in odor gradients due to the mismatch between the dynamic range of the stimulus associated with the light ramps and that experienced in the odor gradient.

## Cross-validation of the GLM for the turn probabilities (material pertaining to *Figure 5—figure supplement 2*)

The integrated stimulus-to-behavior GLM was trained on the full set of light ramps listed in *Figure 5—figure supplement 1*. To validate the structure of the GLM, we followed a cross-validation approach in which the GLM was trained on the 4-s linear ramps listed in *Figure 5—figure supplement 2A* and used to accurately predict the activity elicited by the 8-s ramps listed in *Figure 5—figure supplement 2Bi–iii*. Furthermore, we demonstrated that training the GLM on a single light ramp was sufficient to make accurate predictions on a different ramp (*Figure 5—figure supplement 2C*).

## Likelihood analysis for behavioral predictions (material pertaining to *Figures 6, 8*)

To assess the quality of the behavioral predictions obtained from the stimulus-to-behavior GLM described in *equation (13)*, we computed the likelihood associated with the observation of the entire set of runs obtained in the closed-loop light and odor gradients. Let us denote the total number of runs observed in a particular gradient as N. The likelihood of the ith run was computed based on the turn probability predicted on bins of 1 s. The probability of observing a turn between time $t$ and $(t +1$ s) is: $\lambda_i(t)$ where the index $i$ refers to the sensory experience associated with the ith run. The probability of not turning during a time interval $(t, t +1)$ is: $1 - \lambda_i(t)$. Thus, the probability of observing a given run lasting 5.6 s can be estimated as:

$$p_i = \left(1 - \lambda_i(0.6)\right) . \left(1 - \lambda_i(1.6)\right) . \left(1 - \lambda_i(2.6)\right) . \left(1 - \lambda_i(3.6)\right) . \lambda_i(4.6). \tag{16}$$

Using relationship (16), we calculated the likelihood of the entire set of runs as:

$$L = \prod_{i=1}^{N} p_i. \tag{17}$$

The log-likelihoods (logL) of the predictions associated with the test model and the two controls (see previous section) were computed. To evaluate the reliability of differences between the test model and controls, we applied a standard bootstrap approach (*Martinez and Martinez, 2001*). From the original collection of runs, we generated 10,000 independent new samples of runs based on random resampling with replacement. For each sample, we computed the logL of the test model and the controls. Next we considered the relative difference in logL between the model and the control defined as:

$$\frac{\Delta \text{logL}}{\text{logL}_{\text{test}}} = \frac{\left(\text{logL}_{\text{test}} - \text{logL}_{\text{control}}\right)}{\text{logL}_{\text{test}}}.$$

The distribution of this variable is reported in *Figure 6F* for the behavior elicited by the light gradient, and in *Figure 8D* for the behavior elicited by the odor gradient. Finally, we computed the number of instances where $\Delta\text{logL}/\text{logL}_{\text{test}}$ was lower than 0 (control outperforming the test model) and derived a p-value for the hypothesis that the test model yields a larger likelihood than the controls.

## Optogenetics-based chemotaxis in the background of a silenced and intact olfactory system

For optimal results, the spike-sorting method (OpSIN, *Figure 11B*) developed in this work required that the recordings of the odor-stimulated OSN activity be achieved in the background of a silenced olfactory system ($Orco^{-/-}$). For this reason, the genotype of the larvae used throughout the study for physiological and behavioral quantification resulted from the cross: *w;Or42a*-Gal4,*GMR-hid*;*Orco²*, *dTrpA1¹* × *w;*UAS-*Orco*,UAS-ChR2-H134R;*Orco²,dTrpA1¹*. Given that interneurons of the larval antennal lobe clearly contribute to the neural representation of odors (*Asahina et al., 2009*; *Larkin et al., 2010*), it was important to determine whether the sensorimotor principles controlling chemotactic behavior in a silenced and intact olfactory system are the same. In this aim, we examined the behavior of larvae with ChR2 expressed in the *Or42a* OSN in the background of a fully functional olfactory system (21 intact OSNs): UAS-ChR2-H134R/*Or42a*-Gal4;*gl⁶⁰ʲ*,*dTrpA1¹*. Larvae bearing the double mutant alleles *gl⁶⁰ʲ* and *dTrpA1¹* were insensitive to light (data not shown). In closed-loop and open-loop conditions of light stimulation, we found a high similarity of the chemotactic responses observed in the background of 1 or 21 functional OSNs (*Figure 18*). This result corroborates the idea that the basic control principles learned from single functional OSN larvae are relevant to wild-type larvae.

### Scripts

All scripts described in the 'Materials and methods' as well as the blueprint of the larval tracker are available for download from the following website: https://github.com/LabLouis/eLife_2015.

## Acknowledgements

We thank A Leonardo, E Knoche, and the Louis lab for comments on the manuscript. The tracker was developed with the help of L Ramaswamy (ID&F at Janelia Research Campus). We are grateful to the following individuals for insightful discussions at different stages of the project: C Bargmann (molecular interpretation of the IFF motif), H Davidowitz and D Rinberg (design of olfactometer in gaseous phase), A Kim and A Lazar (comparison between transfer function adult and larval OSNs), A Leonardo (linear filter analysis and data interpretation), R Moreno (likelihood analysis), M Phillips (turn classifiers). We thank S Pulver and L Griffith for donating the UAS-ChR2-H134R transgenic stock. ML acknowledges funding from the Spanish Ministry of Science and Innovation (MICINN, BFU2008-00362, BFU2009-07757-E/BMC and BFU2011-26208). We are grateful to KeyTech (Baltimore, USA) for advising about the design of the tracker and for carrying out the assembly of the hardware. AS, AGM, VGR, MM, JS, JR, DJ, and ML were supported by the EMBL/CRG Systems Biology Program. JS was supported by ICREA. AD was funded by the European Commission FP7 Initial Training Network FLiACT (289941). MS was partially funded by the National Centre for Biological Sciences—Tata Institute of Fundamental Research. GL, PA, ETT, CW, SD, and VJ were supported by the Howard Hughes Medical Institute. This project received support from the HHMI Janelia Research Campus visitor program (AS, AGM, VGR, DJ, and ML).

## Additional information

### Funding

| Funder | Grant reference | Author |
| --- | --- | --- |
| Spanish Ministry of Science and Innovation | BFU2008-00362 | Aljoscha Schulze, Alex Gomez-Marin, Vani G Rajendran, Julia Riedl, David Jarriault, Matthieu Louis |
| Centre for Genomic Regulation | EMBL/CRG | Aljoscha Schulze, Alex Gomez-Marin, Vani G Rajendran, Marco Musy, Ajinkya Deogade, James Sharpe, Julia Riedl, David Jarriault, Matthieu Louis |

| Funder | Grant reference | Author |
| --- | --- | --- |
| Howard Hughes Medical Institute (HHMI) | | Gus Lott, Parvez Ahammad, Eric T Trautman, Christopher Werner, Shaul Druckmann, Vivek Jayaraman |
| European Commission (EC) | FP7 Framework (Marie Curie ITN, FLiACT) | Ajinkya Deogade |
| Tata Insitute of Fundamental research (TIFR) | (NCBS) | Madhusudhan Venkadesan |
| Howard Hughes Medical Institute (HHMI) | Janelia visitor programme | Matthieu Louis |
| Spanish Ministry of Science and Innovation | BFU2011-26208 | Aljoscha Schulze, Alex Gomez-Marin, Vani G Rajendran, Julia Riedl, David Jarriault, Matthieu Louis |
| Spanish Ministry of Science and Innovation | BFU2009-07757-E/BMC | Aljoscha Schulze, Alex Gomez-Marin, Vani G Rajendran, Julia Riedl, David Jarriault, Matthieu Louis |

The funders had no role in study design, data collection and interpretation, or the decision to submit the work for publication.

## Author contributions

AS, Conception and design (electrophysiology and part of the behavioral experiments), Acquisition of all data included in the study (electrophysiology, IR spectroscopy and behavior), Development of the OSN recording technique, Analysis and interpretation of data (electrophysiology), Drafting or revising the article; AG-M, Conception and design (models for OSN activity and behavioral control, part of the behavioral experiments); Analysis and interpretation of data (analysis and modeling of the experimental data), Tracker development (machine vision algorithm), Drafting or revising the article; VGR, Conception and design (part of the behavioral experiments), Acquisition of data (early dataset for behavior), Tracker development (behavioral classifiers); GL, Tracker development (machine vision algorithm), Conception and design; MM, Tracker development (machine vision algorithm), Conception and design, Analysis and interpretation of data; PA, Spike-sorting method based on optogenetics (electrophysiology), Conception and design, Analysis and interpretation of data; AD, Conception and design (analysis of model predictions); JS, Design of Simplex parameter optimization (modeling), Drafting or revising the article; JR, Development of the OSN recording technique, Conception and design, Contributed unpublished essential data or reagents; DJ, Development of the OSN recording technique and spike-sorting algorithm, Conception and design, Analysis and interpretation of data, Contributed unpublished essential data or reagents; ETT, Tracker development (controller algorithm), Conception and design; CW, Tracker development (hardware), Conception and design; MV, Conception of model and analysis of data related to the PDE simulations of odor diffusion (modeling), Acquisition of data, Analysis and interpretation of data; SD, Conception of generalized linear model for behavior (modeling), Conception and design, Analysis and interpretation of data, Drafting or revising the article; VJ, Conception and design of the experiments, Spike-sorting method based on optogenetics (electrophysiology), Analysis and interpretation of data (electrophysiology), Drafting or revising the article; ML, Conception and design of the experiments and models, Tracker development; Analysis and interpretation of data (electrophysiology, behavior and modeling), Drafting or revising the article

## Author ORCIDs
Matthieu Louis, http://orcid.org/0000-0002-2267-0262

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
