## [Decision Letter]

Thank you for sending your work entitled “Dynamical feature extraction at the sensory periphery guides chemotaxis” for consideration at *eLife*. Your article has been favorably evaluated by Eve Marder (Senior editor) and three reviewers, one of whom, Ronald L Calabrese, is a member of our Board of Reviewing Editors.

The Reviewing editor and the other reviewers discussed their comments before we reached this decision, and the Reviewing editor has assembled the following comments to help you prepare a revised submission.

The authors present an extraordinarily thorough electrophysiological and behavioral analysis of the chemotaxis in *Drosophila* larvae in natural odor gradients, defined odor gradients, and optogenetically induced fictive olfactory gradients. These analyses relate OSN activity empirically determined to behavior using a model of sensory transduction and a GLM relating neural activity to the probability of turning. They also create closed loop virtual reality environments to test the effectiveness of their models. They conclude that OSN activity accurately predicts turning probability by a simple linear transform and thus OSN spiking dynamics in odor gradients are key for understanding chemotaxis.

The major biological findings are:

1) Real odor landscapes even in a controlled chamber are complex and they can be described quantitatively.

2) OSN activity in real odor and in fictive odor landscapes can be assayed and then modeled (albeit in an ad hoc manner) so that OSN activity can be accurately predicted in both real and fictive odor landscapes.

3) Fictive controlled odor landscapes can be navigated in a chemotatic way and turn probability can be accurately predicted (using the OSN model and a further ad hoc model) based on OSN activity. Fictive odor landscapes are elegantly designed to extract real behavioral principles. For example a ‘well’ landscape shows that a precipitous drop in predicted (and confirmed) OSN activity induces turns.

4) When real odor landscapes are chemotaxed, then turn probability can be accurately predicted based on predicted (and confirmed) OSN activity.

The major technical innovations are:

1) Real odor landscapes can be calculated based on diffusion equations with sufficient accuracy to extract real odor dynamics for a moving maggot. Confirmed by IR spectroscopy.

2) It is possible to move between the liquid phase and the gas phase when experiments demand it by matching liquid phase and gas phase OSN activity.

3) Identical fictive and real odor waveform leads to measureable differences in OSN activity owing to adaptive processes in the OSN during odor stimuli.

4) Ad hoc models of OSNs can be constructed that capture their essential activity to real and fictive odor stimuli.

5) GLM ad hoc models can capture the relationship of the ONS activity to turn probability, and used as a tool to understand real chemotaxis in an odor gradient.

6) Tracker technology can present freely moving maggots with fictive odor gradients to determine turn probability.

At the paragraph level the writing is clear and the paper is thoroughly illustrated, but finding the above major contributions amid the voluminous data (especially supplemental data) is difficult owing to the overall structure of the paper. Moreover the Discussion seems to focus too much on the OSN model and its implications at the expense of the biological insights for chemotaxis. The authors must be at pains to bring out these contributions in the way they structure the paper and in Abstract and Discussion. The logical flow of the techniques (tool development) and experiments and the conclusions must be made more apparent. One thing that will help immensely with flow will be to delete some of the supplemental figures on tool development or to place the call outs in Materials and methods (simply refer—no figure call outs—to Materials and methods in the Results): Figure 1–figure supplement 1, 2, 3 in Materials and methods; Figure 2–figure supplement 1 can be eliminated, as it is a negative result; Figure 2–figure supplement 2 can be eliminated; Figure 4 can be made into a supplemental figure and placed in Materials and methods; Figure 5–figure supplement 1, 2, 5 in Materials and methods; Figure 6–figure supplement 1 eliminated; Figure 8—figure supplement 1 in Materials and methods.

There are some technical concerns that must be addressed:

1) Statistics. In Materials and methods add a statistics section and write a clear description of the statistical tests used. This is somewhat scattered throughout Materials and methods now. Much of the statistics is in tables and in the figure legends and needs to be made more apparent. Figures 6 and 8 present critical statistical tests that are based on a subset of long run trajectory. The rationale and validity of excluding short runs need to be made more apparent.

2) There is some concern about whether parameters for the models were fixed at some point or whether they were changed with each experiment. One possible reading is that the parameters of Table 1 and Table 3 represent fixed parameters based on the ‘training’ of the associated figure and that subsequent uses of the models used these fixed parameters. The authors must make this explicit. If parameters, were changed with each new experiments then this would seriously diminish the value of the modeling. Specifically, the authors' need to address the question of whether models that were trained on a set of input–output pairs generalize to an equivalent set of pairs that were not in the original training set.

3) There is the concern that the sensory neuron model is fitted to events on a time scale of less than 10 sec (Figures 2, 3 and 4, etc.), whereas the dynamics of chemosensory stimuli during behavior in an odor gradient occur on a time scale of about 30 sec. The authors should acknowledge in text any difference in time scale and dynamic range between real odor time courses and the input domain over which the model was fitted.

4) The odor and channelrhodopsin (ChR) experiments are done in *Or83b* mutant background. Thus the dynamics of *Or42a*-ORNs can be more easily related to behavior in an otherwise silent olfactory system. Will the same be the case in the presence of spontaneous spiking of other ORNs? Another important issue with doing behaviors in a completely silent system is that local interaction at higher levels of olfactory circuits would likely be diminished if not completely absent. Therefore OSN activity-to-turn probability transformation determined might not be representative of that in the WT animal. The authors must address this limitation forthrightly in Discussion.

5) The logit transform of the equation in Figure 5 caused some reviewer confusion; here is a case where some explanation in the text is needed and not just a reference to Materials and methods.

The results presented and tools developed have important implications for behavioral analysis of chemotaxis in animals and set a standard for further mechanistic studies in this important model system. Moreover, the approach used here can be a model for analyses of sensory-motor transformations in other systems.

[Editors' note: further revisions were requested prior to acceptance, as described below.]

Thank you for resubmitting your work entitled “Dynamical feature extraction at the sensory periphery guides chemotaxis” for further consideration at *eLife*. Your revised article has been favorably evaluated by Eve Marder (Senior editor), a Reviewing editor, and one of the original reviewers. The manuscript has been improved but there are some remaining issues that need to be addressed before acceptance, as outlined below:

This revision meets the major expectations set down by the previous reviews. The flow of the paper now is greatly improved. All the main technical points have been resolved and the new statistical section adds significantly. There are a few minor points that should still be considered.

1) The authors belabor the description of Figure 3. The figure itself has too much data (3D and 3E are the only panels that are essential.). Emphasize the main point of the figure: the sensitivity of ORNs to concentration slope. Between the subsection “Characterization of the features encoded by a single larval OSN stimulated by controlled olfactory signals”, where you have concluded that linear filters won't work, and the passage, in the subsection “Phenomenlogical model of the olfactory transduction cascade”, where you start with the dynamical system approach, you lose momentum. The authors can do a better job of explaining how the qualitative observations in Figure 3 led them to the model they built in Figure 4.

2) In the Discussion, or when the paper is cited, it might be appropriate to compare the approach of [76].

---

## [Author Response]

*At the paragraph level the writing is clear and the paper is thoroughly illustrated, but finding the above major contributions amid the voluminous data (especially supplemental data) is difficult owing to the overall structure of the paper. Moreover the Discussion seems to focus too much on the OSN model and its implications at the expense of the biological insights for chemotaxis. The authors must be at pains to bring out these contributions in the way they structure the paper and in Abstract and Discussion.* The logical flow of the techniques (tool development) and experiments and the conclusions must be made more apparent. One thing that will help immensely with flow will be to delete some of the supplemental figures on tool development or to place the call outs in Materials and methods (simply refer—no figure call outs—to Materials and methods in the Results).

We implemented this suggestion to reorganize the main text. We only kept the supporting material necessary to back up specific results. The rest of the information has now been moved to Materials and methods. As was proposed by the editor, we are now referring to Materials and methods for technical details without specific figure/section call outs.

*Figure 1–figure supplement 1, 2, 3 in Materials and methods*.

The former Figure 1–figure supplement 1, 2 and 3 have now been moved to Materials and methods as Figures 9, 10, 11 and 12.

*Figure 2–figure supplement 1 can be eliminated, as it is a negative result*.

We believe the material reported in this figure is important since it motivated the development of a biophysical model based on ODEs instead of the use of a more standard linear-nonlinear (LN) model. Showing that a LN cannot adequately reproduce the OSN spiking dynamics is a negative result that many readers may want to see. Taking the editor’s recommendation into account, we therefore moved the former Figure 2–figure supplement 1 to a figure in Materials and methods (new Figure 13).

*Figure 2–figure supplement 2 can be eliminated*.

This figure has now been eliminated.

Figure 4
*can be made into a supplemental figure and placed in Materials and methods*.

This figure has now been moved to Materials and methods (new Figure 14).

*Figure 5–figure supplement 1, 2, 5 in Materials and methods*.

Figure 5–figure supplement 1 and 2 have now been moved to Materials and methods as the new Figures 15 and 16. We propose to keep Figure 5–figure supplement 5 in the main manuscript as it highlights that nonlinear characteristics in the response of the *Or42a* OSN are behaviorally relevant. This result is part of the main conclusions of the work. We therefore kept Figure 5–figure supplement 5 among the main figures of the manuscript (now labeled as Figure 5—figure supplement 3) to permit a direct reference to it.

*Figure 6–figure supplement 1 eliminated*.

This figure has now been eliminated.

Figure 8—figure supplement 1
*in Materials and methods*.

The material included in this figure provides an explanation for the differences observed in the precision of the navigational behavior elicited by the light and odor gradients. Behavioral differences are likely to arise from the spatial characteristics of each gradient rather than the processing of changes in light and odor intensities. As this point is touched on in the main text, making a direct reference to Figure 8—figure supplement 1 appeared to be necessary. This figure was therefore kept in the main text.

*There are some technical concerns that must be addressed*:

*1) Statistics. In Materials and methods add a statistics section and write a clear description of the statistical tests used. This is somewhat scattered throughout Materials and methods now. Much of the statistics is in tables and in the figure legends and needs to be made more apparent.*
Figures 6 and 8
*present critical statistical tests that are based on a subset of long run trajectory. The rationale and validity of excluding short runs need to be made more apparent*.

Following this recommendation, we have created a new section in Materials and methods: Statistical procedures and data analysis. This section summarizes the metrics used to quantify the goodness of fit, the approaches followed to validate the models, as well as non-standard statistical tests involving physiological and behavioral data.

Regarding the previous use of subsets of runs with a duration longer than the median, we have now removed this filtering from the analysis. The rationale behind our initial focus on runs longer than a minimum duration was to reveal a strong trend on a time window preceding a turn where all runs are defined. In Figures 6 and 8, the predictions of the behavioral model are now plotted for the full set of runs (with a minimum duration of 1 sec). Our conclusions still hold in the absence of this filter, even though the average trend is noisier. Runs shorter than 1 sec could not be considered in the analysis since the turn probability is defined on a time bin of 1 sec.

*2) There is some concern about whether parameters for the models were fixed at some point or whether they were changed with each experiment. One possible reading is that the parameters of*
Table 1
*and*
Table 3
*represent fixed parameters based on the ‘training’ of the associated figure and that subsequent uses of the models used these fixed parameters. The authors must make this explicit. If parameters, were changed with each new experiments then this would seriously diminish the value of the modeling. Specifically, the authors' need to address the question of whether models that were trained on a set of input-output pairs generalize to an equivalent set of pairs that were not in the original training set*.

We thank the editor and reviewers for raising this issue. The parameters of the neural and behavioral models are fixed once and for all in the manuscript. The parameters were not modified in an ad-hoc manner for each experimental condition. We have now clarified this point at different occasions in the main text (including the figure captions). In our opinion, it is remarkable that a behavioral model developed for open-loop optogenetic stimulations holds for closed-loop odor-driven behavior in naturalistic conditions. To further establish the quality of the neural and behavioral models, we added cross-validation tests. In Figure 4—figure supplement 2 and Figure 5—figure supplement 2, we trained the model parameters on a subset of ramps (4-s linear) and successfully applied it to predict other ramps (8-s nonlinear ramps and naturalistic stimuli). These controls should address concerns about a potential overfitting of the neural and/or behavioral models.

*3) There is the concern that the sensory neuron model is fitted to events on a time scale of less than 10 sec (*Figures 2, 3 and 4*, etc.), whereas the dynamics of chemosensory stimuli during behavior in an odor gradient occur on a time scale of about 30 sec. The authors should acknowledge in text any difference in time scale and dynamic range between real odor time courses and the input domain over which the model was fitted*.

The parameters of the neural model were trained on a dataset comprising the linear and nonlinear ramps with a characteristic time of 10 sec together with the naturalistic stimulus associated with a real trajectory (see Figure 2 for odor stimulation and Figure 6 for light stimulation). The naturalistic stimulus had a duration longer than 90 sec. The goodness of fit of the neural model appears to be largely insensitive to the duration of the stimulation. We have nonetheless found evidence for slow adaptation that can be (empirically) modeled by a small temporal correction of one parameter of the model (β_4_, see Table 1). This temporal correction improves the fit without being essential. The biophysical model described in Figure 4 is coarse-grained and one should not expect it to account for physiological processes such as long-term adaptation.

Regarding the input domain over which the model was fitted, we designed the light ramps (Figure 4) to match the intensity domain of the naturalistic light stimulation (Figure 6): both ranges between 0 and 207 W/m^2^. Likewise, the intensity domain of the odor ramps (Figures 3 and 4) and the naturalistic odor stimulus (Figure 8) were chosen to be comparable: 0—500 µM for the ramps and 0—120 µM for the naturalistic stimulus. Consequently, the neural models for the light and odor stimulations were trained on a dataset that included the domain over which the naturalistic patterns of stimulation were predicted.

*4) The odor and channelrhodopsin (ChR) experiments are done in* Or83b *mutant background. Thus the dynamics of* Or42a*-ORNs can be more easily related to behavior in an otherwise silent olfactory system. Will the same be the case in the presence of spontaneous spiking of other ORNs? Another important issue with doing behaviors in a completely silent system is that local interaction at higher levels of olfactory circuits would likely be diminished if not completely absent. Therefore OSN activity-to-turn probability transformation determined might not be representative of that in the WT animal. The authors must address this limitation forthrightly in Discussion*.

To address this concern, we ran a new set of control experiments. We tested a fly cross in which channelrhodopsin (ChR2) is exclusively expressed in the *Or42a* olfactory sensory neuron (OSN). In this cross, olfactory function was retained in the 21 OSNs. To avoid photophobic behavior, we used a double-blind background involving the *dTrpA1*^*1*^ and *glass*^*60j*^ mutant alleles. In Figure 18, we report the strong similarity in the sensorimotor modulation observed for a linear ramp whether light was activating the *Or42a* OSN in the background of only 1 or 21 functional OSNs. In addition, we show that larvae with 21 functional OSNs are capable of robust chemotaxis in a light gradient. Although the antennal lobe is expected to contribute to the processing of the information mediated by *Or42a* OSN, this transformation does not appear to significantly influence the types of behavioral responses considered in the present manuscript.

*5) The logit transform of the equation in*
Figure 5
*caused some reviewer confusion; here is a case where some explanation in the text is needed and not just a reference to Materials and methods*.

In the revised manuscript, we have now introduced the logit transformation more extensively in the main text. Furthermore, we have added Figure 17 in the Materials and methods section where the reader can compare the goodness of fit of the GLM with and without the contribution of the logit transformation.

[Editors' note: further revisions were requested prior to acceptance, as described below.]

*1) The authors belabor the description of*
Figure 3*. The figure itself has too much data (3D and 3E are the only panels that are essential.). Emphasize the main point of the figure: the sensitivity of ORNs to concentration slope*.

Following this advice, we have simplified Figure 3 by removing the previous panels A and B. The dose-response shown in panel A is nonetheless important because (1) it further establishes the similarity in response dynamics of larval and adult-fly OSNs; (2) it validates the IFF+IFB model. We have therefore moved the dose-response curve from Figure 3 to Figure 3—figure supplement 1.

*Between the subsection “Characterization of the features encoded by a single larval OSN stimulated by controlled olfactory signals”, where you have concluded that linear filters won't work, and the passage, in the subsection “Phenomenlogical model of the olfactory transduction cascade”, where you start with the dynamical system approach, you lose momentum*.

We have changed the text and inverted the order of presentation of the features encoded by the *Or42a* OSN and the results of the linear filter.

*The authors can do a better job of explaining how the qualitative observations in*
Figure 3
*led them to the model they built in*
Figure 4.

The presentation of the OSN model now begins with the introduction of the integral feedback and incoherent feed-forward motifs. These two motifs were considered because of (1) their involvement in olfactory transduction in other model systems and (2) the nonlinear dynamics they could generate. We refrained from commenting on the specific design of the ODE equations as it would require technical considerations that would break the flow of the main text. A detailed description of the ODE model can be found in the Materials and methods (section: Incoherent feed-forward (IFF) and integral feedback (IFB) motifs).

*2) In the Discussion, or when the paper is cited, it might be appropriate to compare the approach of*
[76].

We have added a (brief) description of the approach used by Nagel and Wilson, together with the main findings reported in their paper. Throughout the text, about 10 references are now made to this paper.